# Polymeric Nanosystems Applied for Metal-Based Drugs and Photosensitizers Delivery: The State of the Art and Recent Advancements

**DOI:** 10.3390/pharmaceutics14071506

**Published:** 2022-07-20

**Authors:** Kele Cristina Ferreira Dantas, Jânia dos Santos Rosário, Priscila Pereira Silva-Caldeira

**Affiliations:** Departamento de Química, Centro Federal de Educação Tecnológica de Minas Gerais, Avenida Amazonas 5253, Belo Horizonte 30421-169, MG, Brazil; kelecfd@gmail.com (K.C.F.D.); jania.santos@hotmail.com (J.d.S.R.)

**Keywords:** metallodrugs, photosensitizers, polymer conjugate, drug delivery systems

## Abstract

Nanotechnology-based approaches for targeting the delivery and controlled release of metal-based therapeutic agents have revealed significant potential as tools for enhancing the therapeutic effect of metal-based agents and minimizing their systemic toxicities. In this context, a series of polymer-based nanosized systems designed to physically load or covalently conjugate metal-based therapeutic agents have been remarkably improving their bioavailability and anticancer efficacy. Initially, the polymeric nanocarriers were applied for platinum-based chemotherapeutic agents resulting in some nanoformulations currently in clinical tests and even in medical applications. At present, these nanoassemblies have been slowly expanding for nonplatinum-containing metal-based chemotherapeutic agents. Interestingly, for metal-based photosensitizers (PS) applied in photodynamic therapy (PDT), especially for cancer treatment, strategies employing polymeric nanocarriers have been investigated for almost 30 years. In this review, we address the polymeric nanocarrier-assisted metal-based therapeutics agent delivery systems with a specific focus on non-platinum systems; we explore some biological and physicochemical aspects of the polymer–metallodrug assembly. Finally, we summarize some recent advances in polymeric nanosystems coupled with metal-based compounds that present potential for successful clinical applications as chemotherapeutic or photosensitizing agents. We hope this review can provide a fertile ground for the innovative design of polymeric nanosystems for targeting the delivery and controlled release of metal-containing therapeutic agents.

## 1. Introduction

Nowadays, polymer-based systems serve as controlled-release drug delivery systems and consist of a suitable approach for improving the existing treatment of various illnesses. This powerful strategy focuses on targeting drugs to a desirable group of cells or tissue, reducing the systemic dose, and increasing the amount of active agent that reaches the aimed site of action while minimizing undesired side effects of the drugs [1]. A selection of polymeric formulations designed as micelles, hydrogels, nanospheres, and nanocapsules have been studied as suitable systems for the delivery of classical organic [2,3] and metal-based therapeutic agents [4,5,6,7,8,9,10,11,12]. The polymeric nanocarrier-assisted drug delivery systems have attracted significant research interest mainly due to their notable properties.

Nanocarrier-based delivery systems designed for metal-based compounds have emerged during the last three decades, especially for platinum chemotherapeutic agents [13,14,15,16] and metal-based photosensitizers (PSs) [17,18,19]. The use of polymeric nanocarriers for platinum-based drugs arose from the necessity to explore alternative strategies to reduce their side effects and improve their pharmacokinetic profiles. For PSs, the poor water solubility, aggregation, and low tumor selectivity were the main issues that prompted research for the development of PS delivery systems.

Thus, the employment of nanotechnology for the delivery of platinum drugs [20,21] is now a recognized field with some nanoformulations in clinical tests and even in medical applications. Preliminary investigations show encouraging results from the nonplatinum-containing metal therapeutic agents associated with polymeric nanosystems [22], stimulating comprehensive research in this field.

In light of the fast developments in nanosized vehicles for drug delivery and the need for regular updates, this review focuses on polymer-based nanocarriers applied for metal-based therapeutic agents. Accordingly, we discuss some biological and physicochemical properties of the metallodrug–polymer assembly (Section 3). We survey the polymeric nanocarriers used for the delivery of metal-based drugs and PSs with particular attention to those used for nonplatinum-containing agents (Section 4). We highlighted some polymeric nanocarriers coupled with metal-based drugs applied for other diseases beyond cancer (Section 5) and the new trends in nanotechnology involving metal complexes with therapeutic properties (Section 6).

## 2. Metal-Based Therapeutic Agents

Metal-based drugs are extensively used in the clinic for the treatment and diagnostics of a wide range of diseases [23]. Metal-based drugs have severe limitations that are proving difficult to overcome, including instability in human plasma, unsuitable lipophilicity for effective cellular uptake, lack of selectivity or efficiency, development of side effects, and resistance mechanisms. These limitations, added to the unique tailorable structures and physicochemical properties of metal complexes motivated a large number of studies into the potential of the use of appropriate polymeric drug delivery systems for metal-based drugs aiming to suppress the main drawbacks mentioned above [24].

Furthermore, beyond acting as classical chemotherapeutic agents, metal-containing compounds have also been investigated as photosensitizers for photodynamic therapy (PDT) [25]. PDT is a promising and less invasive method for treating cancer and other diseases, which is based on the use of a PS and visible light irradiation to produce reactive oxygen species (ROS) [26,27]. It is expected that the localized ROS generation by activation of a PS induces tumor cell necrosis and/or apoptosis with functional preservation of surrounding normal tissues. The ROS photogenerated by PS is singlet oxygen (^1^O_2_), which is reactive to damaged cancer cells but has a short lifetime. Photo-induced in situ ^1^O_2_ generation facilitates its enrichment for PDT; however, the therapeutic effect of PDT is restricted by the light penetration depth and insufficient O_2_ supply in hypoxic tumors. In addition, the limitations of the PSs, such as poor water solubility, aggregation, and lack of target ability, hindered their clinical application in PDT. Thus, many efforts have been made to address these problems, and polymeric nanosized platforms employed for carrying metal-based PSs have been showing promising outcomes.

In this context, the chance of creating nanosized carriers paired with the possibility to conjugate to the metal center of complexes are relevant advantages of polymeric devices for the delivery of metal-based agents. Polymeric nanocarriers generally can circumvent biological barriers and delivery the drug directly to target cells in therapeutic doses while preserving the healthy tissues from the adverse effects of such chemical agents [1]. In addition, the polymeric matrix can be structurally modified to tether to surface ligands to bind to the metal center from metal-based drugs or can encapsulate the unaltered metallodrug. This way, polymeric nanocarriers have attracted substantial attention as a strategy for vectorizing metal-based therapeutic agents [22,28].

## 3. Polymeric Nanocarriers for Metal-Based Therapeutic Agents’ Delivery

A variety of nanotechnologies and coating strategies for metal-based therapeutic agent delivery are currently under clinical application, preclinical investigation, and development [29,30]. The engineering of nanosized delivery systems capable of protecting and transporting specific bioactive coordination complexes to a target site of action is a challenging task. The literature describes numerous options of versatile platforms for being employed as nanosized systems for metal-based agent delivery, which are categorized into the following classes: polymer-, lipid-, ceramic-, and carbon-based nanocarriers, inorganic NPs, nanoscale biological macromolecules, and metal-organic frameworks [22,31,32,33,34,35,36].

Despite the wide range of available nanosized systems for drug delivery, those made from purely organic molecules that include micelles, polymeric NPs, dendrimers, liposomes, and polymersomes are currently the most extensively used materials for carrying drugs [37].

### 3.1. Advantages of Employing Polymeric Nanoparticles as Nanocarriers for Metal-Based Therapeutic Agents

Amongst the organic platforms, polymeric matrices are attracting a great deal of interest as drug delivery carriers because of their good pharmacokinetics, prolonged circulation times in the bloodstream, desirable biocompatible and biodegradable properties, compositional adaptability, and other advantageous features as schematized in Figure 1. On the other hand, metal-based therapeutic agents are generally susceptible to hydrolytic degradation and have high reactivity toward serum proteins—some of them are poorly soluble in aqueous media, and, therefore, they are relevant candidates for drug delivery systems [38]. Seen in these terms, the administration of the polymer-metal complex assembly is an increasing research area, particularly useful to address the uptake and stability issues of the metal-based agents. The collaborative efforts of metal coordination chemistry and polymer science have led to new chemotherapeutic technologies providing metal complex nanoformulations with therapeutic superiority over the free form of metal-based drugs or PSs.

### 3.2. Types of Structures of Polymeric Nanoparticles

Nanostructured polymer-based carriers, depending on their synthesis and reactants, can adopt some types of structures with distinct features that can be typically divided into three categories, namely micelles (or nanomicelles), nanospheres, and nanocapsules, as schematized in Figure 2.

Polymeric micelles are core–shell nanostructures with a small diameter (less than 100 nm) widely used for drug release. These structures are composed of hydrophobic and hydrophilic block copolymers with self-assembly properties [39]. Micelles are attractive as drug delivery systems for poorly water-soluble therapeutic agents since their hydrophobic core accommodates hydrophobic drugs. At the same time, their hydrophilic shell lets the whole structure be water-soluble [39,40]. Other advantages of the incorporated therapeutic agent to nanomicelles are the possibility of the drug achieving good penetration and site-specificity due to the small particle size of nanomicelles [41,42]. Though, a common problem to overcome is that conventional micelles have premature drug release in the blood circulation due to their disintegration into individual polymeric components upon dilution in a physiological environment [40]. However, some researchers have revealed that cross-linking either the core or the shell of the micelles as well as changes in the chemical structure of the inner core may be alternatives to increase the stability of these systems and provide dramatic improvements in both pharmacokinetics and biodistribution of the encapsulated drugs [40,43].

Polymeric NPs, which comprise nanospheres and nanocapsules, are solid particles ranging from 10 to 1000 nm in size. Nanocapsules are vesicular systems with cores where the drug is confined and surrounded by a polymeric membrane, while nanospheres are matrix systems in which the drug is dispersed within the polymer throughout the particle [44,45]. The polymeric NPs have the advantages of enhancing therapeutic efficacy by (i) targeted delivery of drugs in a tissue- or cell-specific manner; (ii) releasing drugs across a range of biological barriers, including intracellular sites of action; (iii) the ability to load therapeutic agents with potentially different physicochemical properties; and, therefore, (iv) the ability to improve the pharmaceutical and pharmacological properties of drugs [37]. Furthermore, some polymeric nanostructured systems can co-delivery metal-based active compounds and other chemotherapeutic agents, which provides an attractive strategy for maximizing therapeutic efficacy [46,47,48,49].

### 3.3. Polymers Commonly Employed in NPs for Metal-Based Agent Delivery

For preparing nanosystems for metal-based drug delivery, the hydrophilic segments poly(ethylene glycol) (PEG, [CH_2_CH_2_O]_n_, also referred to as poly(ethylene oxide), PEO) have been the most commonly employed. One prominent reason for using PEG is because it enhances blood circulation time, improving the therapeutic outcome. Moreover, PEG is inexpensive, has good biocompatibility, and is recognized as safe for internal applications in humans by regulatory agencies, such as the US Food and Drug Administration (FDA) [30]. A specific set of enzyme-susceptible polymers are also employed in conjugation with PEG, which enables further functionalization and site-specific delivery of drugs [45]. Besides being vastly applied for the physical encapsulation of metallodrugs, alone [50] or in combination with other polymers [51,52], PEG has also been employed for preparing metallopolymer prodrug NPs [53,54]. Attaching PEG to metallodrugs by covalent conjugation is a common methodology in drug administration for improving efficacy [55] since this approach permits the self-assembly of metallopolymer in micelles [54], enhances their biocompatibility [56], and reduces the metallodrugs’ interactions with proteins or other biomolecules in the blood circulation [47].

The biocompatible and biodegradable polymers polylactide (PLA, [C_3_H_4_O_2_]_n_) and poly(glycolic acid) (PGA), as well as the corresponding copolymer poly(lactic-co-glycolic acid) (PLGA, [C_5_H_8_O_5_]_n_), are on the list of the most extensively studied synthetic polymers [57]. Under physiological conditions, these aliphatic polyesters are hydrolyzed and decomposed into their non-toxic monomeric components, lactic acid, and glycolic acid, which are biologically inert and are eliminated from the body through common metabolic pathways [58]. That meant these polymers are recognized as being safe by the FDA for drug delivery and other medicinal uses [59]. Although the limitation of these polymers is their imminent clearance from the circulatory system, and to overcome this, often moieties of hydrophilic polymers, such as PEG, are grafted to them, which could result in a copolymer with adequate circulation stability [60]. PLA and PLGA are more commonly utilized as the polymeric matrix in the physical encapsulation of metallodrugs into NPs [46,57,61,62]; however, we find in the literature one example of the employment of PLA to form a metallopolymer prodrug [63].

Notably, copolymer systems are frequently used for nanosized drug delivery systems since they can synergistically improve the therapeutic efficacy of drugs by taking advantage of individual polymer characteristics. The PLGA–PEG and PLA–PEG copolymer systems, for example, are of particular interest in drug delivery systems because it has individual hydrophobic and hydrophilic segments and could self-assemble in an aqueous solution by nanoprecipitation methods [45,59]. Additionally, various synthetic techniques could be explored to improve the functionalities for achieving target specificity, control of drug concentration, and exposure kinetics at the tissue, cell, and subcellular levels. PLGA–PEG and PLA–PEG copolymer systems have been employed for physically encapsulating metallodrugs [64,65,66].

Poloxamer, also known as Pluronic^®^, comprises non-toxic and non-ionic triblock copolymers consisting of two hydrophilic chains poly(ethylene oxide) (PEO, [CH_2_CH_2_O]_n_) and the central hydrophobic chain poly(propylene oxide) (PPO, [C_3_H_6_O]_n_), PEO-PPO-PEO, in different proportions [67,68]. This copolymer is safe and extensively studied for use in medicine as well as cosmetics. The incorporation of drugs in the nucleus of micelles formed by Pluronic^®^ increases the water-solubility, stability, and blood circulation time of the active agent [69]. The most used and studied are Pluronic^®^ F68 (PF68) and F127 (PF127) [70], being PF127 the most commonly Pluronic^®^ used for encapsulating metallodrugs [71,72,73,74]. In addition, PF127 has been a very popular stabilizer for nanostructured emulsions [75].

### 3.4. Physical and Chemical Interactions between Metal-Based Agents and Polymeric NPs

The incorporation of metal-based compounds into polymeric nanocarriers may permit stabilizing the metal-based agent in the physiological environment, preventing undesired premature (extracellular) reduction, hydrolysis, or degradation involving ligand dissociation from the metal coordination sphere [76]. The higher stability in the physiological conditions of the drug-loaded nanoassembly compared to free drug promotes enhanced blood circulation time of polymer–metal complex ensemble, which grating an opportunity for tumor-targeting delivery.

It is worth mentioning that incorporating metal-based compounds into polymeric nanosized vehicles for metallodrug delivery is rather challenging due to the possibilities of chemical/physical interactions that the ensemble polymer–metallodrug can adopt. The combination of metal-based drugs with polymeric nanocarriers can be achieved by two main strategies: (1) non-covalent encapsulation or (2) covalent conjugation of the metal complex to a polymer. The second strategy, generally, generates a prodrug that can be converted into an active metal-based agent at a more controllable rate.

The encapsulation is a physical process that has the advantage of not affecting the integrity of the metal-complexes structures, such as the geometry, oxidation state, and stereochemistry of the metallic center [77]. The other advantages include that this method is known to be synthetically not challenging, and it could enhance the bioavailability of poorly water-soluble drugs. However, this incorporation method suffers from relevant limitations that hamper intravenous administration, such as (i) uncontrolled drug release, an effect known as “burst release”, which consists of the abrupt release of a large amount of drug post-administration; (ii) low drug-loading capacity (generally, less than 10%), thus demanding a high amount of drug delivery systems to obtain a therapeutic effect; and (iii) difficulty to encapsulate drugs that are poorly miscible in the polymeric matrix [21].

In some cases, it is not possible to classify the incorporation method as purely noncovalent since the polymer, previously designed to encapsulate the metallodrug, can bind to it through functional groups that act as ligands within the polymeric chain [13]. Indeed, to overcome the premature disassembly of micelles in blood circulation, one way is to cross-link the shell or core of the micelles with cross-linking agents or use the metallodrug itself as a cross-linking agent [15]. Despite the limitations, physical encapsulation is vastly investigated as an appropriate system for the controlled delivery of poorly soluble metal-based therapeutic agents.

Metal-based compounds offer a variety of opportunities for polymer conjugation, with specificities of each metallic center and compound. In addition, some strategies can be assumed, including conjugating the metallodrug to a self-aggregating or coordinating the metallodrug to one amphiphilic component of the polymeric platform. The covalent conjugation of the polymer chain to metal complexes can be reached by the following approaches: (i) by donor sites present in the polymer structure either in the main or in the ramification chain [78]; (ii) by tethering to polymeric nanostructure ligands that allow the metal coordination [79,80]; (iii) through coordination of a polymeric macroligand to the metal complex precursor [81]; and (iv) via metallodrug-initiated polymerization [63]. Regardless of the experimental method adopted, the polymer–metal complex ensemble is linked as a prodrug to the supramolecular aggregate, which acts as a nanocarrier [28].

Polymers containing in their backbone or branches amines, carboxylates, or hydroxyl, as functional groups, can bind metal complexes via nitrogen or oxygen donor atoms. The polymer attached to the metal-based agents can act as a stable ligand for metallodrugs or as a leaving group of a metallodrug. Polymer designed for providing temporary bind to the metallic center is more commonly used for Pt^2+^-based compounds, but this mode of platination is problematic because facile hydrolysis of the polymer–Pt^2+^ bonds may cause the drugs to prematurely leak from the carrier [14,15].

Some advantages of employing covalent conjugation of metal-based agents to the polymeric matrix are reproducibility, high drug-loading percentages, more controllable drug release, and good blood circulation stability. However, some shortcomings of covalent conjugation are (i) structural modification of metallodrug that can impact its biological activity, (ii) difficulties surrounding the synthetic procedure, and (iii) possibility of contamination by catalysts or other reagents during the polymer–metal complex formation. In addition, the synthetic strategy employed for polymer–metal complex formation depends on where the metallic center is conjugated to the polymer (e.g., the backbone, side chain, the center of a star polymer).

### 3.5. Metal-Based Therapeutic Agents Loading Capacity into Polymeric NPs

An important parament to evaluate in nanocarriers drug delivery systems is the drug-loading capacity, which has relation to the nature of the interactions formed between the metal-containing therapeutic agent and the copolymer matrix. Generally, in physical incorporation approaches, the drug-loading process often results in a low amount of metal-based agents loaded into the host, while covalent conjugation, in which the metallodrug is a constituent part of the nanocarrier, renders high drug-loading efficiency [82]. In addition, the covalent conjugation, besides enabling controlled high loading of metal-based agents into the metallopolymer, often ensures the prolonged release of the therapeutic agent during polymer biodegradation [77].

Indeed, a pivotal problem to overcome in metallodrug encapsulation into polymeric micelles is to maximize the payload amount in the nanosystem and decrease the relative amounts of the polymeric excipient used. In general, micelles have low loading efficiency and stability, while nanocapsules and nanospheres have better drug-loading capacity and high stability in biological systems and in storage conditions [62]. Some nanoformulations containing metallodrugs physically encapsulated in nanocapsules or nanospheres can reach almost 90% of encapsulated efficiency [83,84]. For polymeric micelles, the maximum achievable drug-loading mainly depends on the chemical affinity between the metal-based agent and the hydrophobic polymer block [48]. On the other hand, the main parameter that influences the drug entrapment efficiency into nanospheres/nanocapsules is the preparation method [64,85] since the packing density of the polymer matrix influences drug-loading capacity [86]. In addition, when NPs are loaded with a single drug, entrapment efficiency is higher independently of the drug employed, and the type of copolymer used [48].

The encapsulation efficiency of metal-based agents into NPs is commonly determined by applying a standard curve employing UV-Vis spectroscopy [51,87,88] for metallodrugs and fluorescence emission [42,62,89,90] for photosensitizers. Methods less used determine the amount of metal by inductively coupled plasma mass spectrometry (ICP-MS) [61] and inductively coupled plasma atomic emission spectroscopy (ICP-AES) [91].

### 3.6. Target Delivery and Controlled Release of Metal-Based Agents by Polymeric NPs

Beyond the benefits cited in Figure 1, polymeric nanoparticles (NPs) can selectively accumulate into cancer cells, and some structural modifications in the polymeric matrix permit controlling the drug release process.

The enhanced permeation and retention (EPR) effect, as first defined by Maeda and Matsumura [92], permits a passive entry of nanocarriers into the tumors through the existing inter-endothelial gaps (Figure 3). Despite being a concept currently controversially discussed [93,94], for almost three decades, the EPR effect was assumed as the preferential pathway of nanoparticles (NPs) entering cancerous tissues. However, a comprehensive study reported recently by Chan and coworkers [95] has proven that the active *trans*-endothelial transport pathways are the dominant mechanism of NPs extravasation into tumors (up to 97%), Figure 3. However, further research is necessary to define which molecular mechanisms drive NPs into solid tumors [95].

Other advantages of polymeric systems are the chance of an infinity of structural modifications, which permits the polymeric carrier to be composed of subunits featuring: (i) some molecular entity enabling facilitated cell entry or targeting moiety for site-specific action; (ii) water-solubilizing groups; and (iii) functional groups suitable for the critical conjugation step involving metallodrug binding to the polymer [96]. In addition, due to the versatile surface chemistry of polymeric NPs, it is possible to provide the functionalization with target receptors, such as folic acid (FA), biotin (vitamin B7), and other biomolecules, whose receptors are overexpressed in certain cancer cells [71,73,97,98,99]. This strategy promotes an active site-specific delivery to preferred targets without undesired accumulation in essential body organs.

Furthermore, a controlled activation or release of metallodrugs-loaded NPs can be achieved by using exogenous or endogenous stimuli, which protect the NP’s content from external influences and release the therapeutic agent only in response to specific environmental conditions [100]. The responsive stimuli can be intrinsic, such as intracellular reduction condition, low intracellular pH, and presence of enzymes, or extrinsic stimuli, such as light, variation in temperature, magnetic field, and others [79,101,102,103] (Figure 1).

## 4. Non-Platinum-Containing Metal Therapeutic Agents Loaded into Polymeric Nanosystems

The first polymeric nanocarriers for delivering metal-based drugs were developed for platinum therapeutic agents. Indeed, platinum-based chemotherapeutics are widely employed as anticancer drugs, being applied for various therapies against a broad spectrum of solid malignant tumors [104]. Though, to approach issues such as adequate pharmacokinetic profile, lower systemic toxicity, and reduced drug resistance, many studies have been carried out to elucidate the mechanisms of action of platinum-based drugs aiming to design novel platinum chemotherapeutic agents. As a result of efforts devoted to preparing novel platinum-containing metal drugs, three categories are currently in clinical treatments and advanced preclinical trials: classical Pt^2+^ drugs, non-classical Pt^2+^ drugs, and Pt^4+^ prodrugs [15].

Although there is a large array of platinum-based drugs, and some of them are widely used in clinical practice, drawbacks such as severe side effects, low absorption rate, and development of intrinsic or acquired resistance led to the orientation of research toward finding new therapeutic approaches. Then, aiming to enhance the therapeutic performance, minimize the toxic effects on normal tissues, and circumvent the emergence of resistance, a number of platinum drug delivery systems have been developed [105]. Amongst them, the polymer-based systems have been intensively explored [15,106].

Undoubtedly, the polymer-based vehicles for delivering platinum therapeutic compounds are the most studied and employed amongst all classes of metal-based drugs; however, our discussion will focus on polymeric nanocarriers for nonplatinum-containing metal drugs delivery. For a more comprehensive analysis of the state of the art of polymer-based platinum drug delivery systems, the reader is referred to several excellent reviews [15,16,105,106,107]. Notably, the efforts dedicated to improving the pharmacological properties of platinum drugs by employing polymer-based delivery systems opened the way to further development of polymeric nanocarriers for non-platinum metal-based drugs, which are discussed here below.

The abovementioned drawbacks of platinum-based drugs have encouraged increased research efforts in the search for novel non-platinum metal-based species as chemotherapeutic agents. The development of new metal-based agents with modes of action distinct from platinum-based drugs is necessary due to the increased resistance phenomena to platinum drugs. Pharmacological studies and preclinical investigations for cancer treatment have been revealing that complexes with gold(I/III), copper(II), or iridium(III) as central metal ions present promising outcomes, while ruthenium(II/IIII) and gallium(III) compounds have advanced to the clinic or reached the phase I/phase II of clinical trials.

Unfortunately, as expected, non-platinum metal-based agents also have some limitations restringing their use, so the employment of nanocarriers also could be an attractive approach for overcoming these drawbacks. As summarized in Figure 2, the use of polymeric nanosized delivery systems for releasing metal-based drugs improves the therapeutic profile of these drugs.

Table 1 provides a general overview of polymeric nanocarriers used for delivering non-platinum metal-containing drugs and photosensitizers. The list is comprehensive, and it illustrates the wide range of polymer-based nanosized delivery systems that could be applied for metal-based agents’ delivery.

As shown in Table 1 and Figure 4, for therapeutic agents containing gallium(III), copper(II), and zinc(II) as the metallic centers, the non-covalent entrapment into polymeric nanocarriers seems to be the most used, as well as is reported for platinum-containing agents [20]. Furthermore, the nanosystems designed for ruthenium(II/III) and gold(I/III) compounds’ delivery have a more egalitarian distribution of encapsulation into and covalent conjugation to the polymeric NPs. In contrast, for iridium(III) complexes, the metallodrug–polymer covalent interactions in nanocarriers are by far the dominating approach.

### 4.1. Ruthenium Complexes

Ruthenium-containing compounds are the most promising non-platinum candidates for metal-based cancer therapy due to their suitable features for anticancer drug design, such as the ability to strongly bind nucleic acids and proteins [160], ligand exchange kinetics similar to platinum compounds [161], preferential accumulation in neoplastic tissues [162], the possibility of photodynamic approaches to therapy [163,164], ability to mimic iron binding in biological systems [165], and lower toxicity than platinum compounds [166]. Indeed, four ruthenium coordination compounds [167,168,169,170] (Figure 5A) and three organoruthenium compounds [171,172] (Figure 5B) have reached the stage of clinical evaluation in humans for cancer treatment.

For ruthenium-based compounds to reach clinical use, they must, beyond having significant antiproliferative activity and reduced side effects, enter tumor cells selectively, present good water solubility, and have adequate plasma half-life. In addition, especially for light-reactive polypyridyl Ru^2+^ complexes, it is also required, for medical application, to minimize the residual toxicity in the dark. Several strategies have been explored to incorporate ruthenium agents into a number of nanosized devices, ranging from organic to inorganic materials [167,173,174,175,176]. The incorporation of Ru-based compounds into polymeric NPs through physical encapsulation or conjugation has recently gained much attention due to the superior chemotherapeutic benefits compared to the free ruthenium agents.

#### 4.1.1. Mono- and Dinuclear Ruthenium(II/III) Based Complexes

The low solubility and stability in an aqueous solution of the Ru^3+^ drug KP1019 (**Ru1**), especially at physiological pH, is a significant limiting aspect of its medical application. So, Keppler and coworkers prepared PLA NPs with two non-ionic surfactants (PF68 and Tween 80) for encapsulating KP1019 [57]. Tween 80 allowed reproducible preparation of **Ru1**-containing NPs with very high loading efficiency. They observed a color change from brown to green, which was associated with the generation of Ru^2+^ species. A 20-fold increase in cytotoxic activity, compared with free KP1019, was found in the produced **Ru1**-loaded. Another relevant contribution of this investigation was providing a new approach employing polymeric NPs to stabilize a biologically active Ru^2+^ species of KP1019 in an aqueous solution [57]. In another investigation, the same research group reported the encapsulation of the drug KP1019 into PEGylated polymeric micelles [50]. The obtained KP1019-loaded micelles proved to be highly biocompatible and efficient with cellular accumulation and cytotoxic activity higher than the free Ru compound.

Employing covalent conjugation, Stenzel et al. designed an amphiphilic block copolymer to act as an appropriate carrier for NAMI-A (**Ru2**) [108]. They observed that NAMI-A copolymer micelles have an improved antimetastatic ability and better inhibition of cell invasion and migration than the metallodrug alone in ovarian and pancreatic cancer cells.

Fregona et al. carried out chemical modifications of PF127 to generate micelles conjugated to active substrates for encapsulation of a ruthenium(III) and copper(II) dithiocarbamate derivative complex (**Cu1** and **Ru3**) [97]. The modifications were performed in the terminal hydroxyl groups of the copolymer PF127 to generate active substrates towards fluorescein, glucosamine, and the reaction for conjugation of β-D-glucopyranoside. The authors observed that all the obtained complex-containing micelles are stable under physiological conditions and have in vitro antiproliferative activity against different aggressive human cancer cell lines in the same concentration domain of the corresponding free complexes tested in DMSO [97].

Some dinuclear compounds of ruthenium show significative antiproliferative activity. Hence, Fregona et al. proposed novel mono- and dinuclear Ru^3+^-complexes with aromatic and non-aromatic dithiocarbamates as ligands (**Ru4**–**Ru8**) [72]. The encapsulation of the synthesized Ru^3+^-based compounds into PF127 copolymer micelles improved their water solubility, stability, and bioavailability. In a similar line, Oliveira Silva et al. documented two investigations involving the encapsulation of diruthenium(II, III) coordinated with the drugs ibuprofen and naproxen (**Ru9** and **Ru10**) [83,109]. The obtained metallodrug-loaded NPs showed improvement of the antiproliferative effect compared to the corresponding free diruthenium metallodrugs in breast, prostate, and glioblastoma cancer cells and were effective against cancer cells chemoresistant to cisplatin [83,109].

#### 4.1.2. Ruthenium(II)-Based Polypyridine Complexes

Lemercier and coworkers reported the physical encapsulation of two Ru^2+^ polypyridine complexes (**Ru11** and **Ru12**) in PLGA NPs in the presence of PF68 [61]. The obtained NPs were much less toxic in C6 glioma cells than the related free Ru^2+^ complexes after irradiation. Chen et al. prepared a biotin-conjugated tri-block polymer (consisting of PLGA, polyethyleneimine, and PEG) delivery system and tested it as a nanocarrier of a potent Ru^2+^ polypyridine complex (**Ru13**) [87]. The authors demonstrated that the prepared **Ru13**-loaded NPs were efficiently internalized and triggered ROS overproduction in cancer cells promoting the p53-mediated apoptosis process. Meanwhile, upon encapsulation, the water solubility of the complex increased.

Glazer’s research group used crosslinked polymeric assemblies to load Ru^2+^ complexes focusing on improving the efficacy of Ru^2+^-based PSs (**Ru14**–**Ru16**) [51]. The authors efficiently performed the physical entrapped of hydrophilic and hydrophobic Ru complexes into PEG-poly(aspartate) block copolymers and observed enhanced blood circulation time of the Ru complex loaded into the nanosystems compared to the free Ru^2+^ complexes. However, the cytotoxicity of Ru-loaded nanosystems in cancer cells, before and after irradiation, did not significantly change compared to the free corresponding Ru^2+^ complexes [51].

In the same direction, Gasser’s research group dedicated efforts to the encapsulation of photoactive ruthenium(II) polypyridine complexes into polymer-based nanocarriers for selective delivery. Their first study approaching this field described the physical encapsulation of a Ru^2+^-based PS (**Ru17**) in stimuli-responsive block copolymers with poly(*N*,*N*-dimethylaminoethyl methacrylate) and poly(methyl methacrylate) [110]. The authors demonstrated that Ru^2+^ complex release could be triggered by temperature, pH variation, and ultrasound. Biological tests were not reported. In another work, they performed an encapsulation of a Ru^2+^ polypyridine complex (**Ru18**) within copolymer system PF127, which employs different loading ratios and showed that the obtained NPs presented high stability in water and physiological medium, higher cellular uptake, and superior phototoxicity compared to the free Ru^2+^ complex [73]. They also reported the encapsulation of a promising Ru^2+^-based PS (**Ru19)** into a PEGylated phospholipid polymer with a terminal biotin group [111]. The encapsulation of the Ru^2+^ complex in a biotin-containing polymer consists of a strategy to provide an additional cancer cell targeting effect since biotin’s receptor is overexpressed in a variety of cancers. The resulting **Ru19**-loaded NPs presented much higher selectivity for cancer cells in comparison to noncancerous cells, an improved accumulation inside a human adenocarcinoma tumor of a mouse up to a factor of almost nine compared to the Ru^2+^ complex itself, and a high phototoxic effect with the eradication of tumor inside a mouse model. In a recent study, they used the same strategy of accoupling to polymer chain a biologically active molecule enabling facilitated cell entry [98]. In this investigation, folate groups were attached to the end of the hydrophilic part of a PEGylated phospholipid polymer. Due to the high folate demand for rapidly growing cells, the folate receptor is overexpressed in cancer cells compared to healthy cells. Indeed, the encapsulated Ru^2+^ polypyridine complex (**Ru20**) was found to be almost nine times more selective for cancerous cells than noncancerous cells and presented activity in drug-resistant cancer cell lines. In addition, the encapsulated complex demonstrated reduced cytotoxic action in the dark and improved photophysical properties compared to the free Ru^2+^ complex [98].

In 2018, Vallis et al. documented an innovative approach combining theranostic (therapeutics and diagnostics) properties of radionuclides with controlled release of chemotherapy and cancer cell targeting [46]. The authors prepared radiolabelled PLGA NPs with ruthenium(II) metallo-intercalator and radiosensitizer encapsulated (**Ru21**). Besides this, the PLGA NPs were functionalized with a targeting ligand for epidermal growth factor receptor (EGFR), which is overexpressed by cancerous but not normal cells. The outcomes of this advanced strategy for the preparation of metallodrug formulations were that the co-delivery of ruthenium-based radiosensitizer alongside radionuclide (^111^In) decreased cell survival compared to single-agent formulations. Moreover, a substantial decreased uptake and radiotoxicity of NPs towards cells with normal EGFR levels were also observed [46].

Although several works described the incorporation of Ru^2+^ polypyridyl PSs into polymer matrices through physical interactions, the limitations of this method for drug release, as outlined in Section 3, prompted researchers to dedicate efforts to finding more appropriate carriers for Ru^2+^ complexes employing covalent conjugation. Indeed, in the literature, several reports describe ruthenium(II) polypyridine complexes covalently conjugated to polymeric materials, which contain the ruthenium moiety in either the polymer main chain or the side chain [112].

Sleiman and coworkers reported the preparation of polymeric nanostructures conjugated to biotin and ruthenium(II) bipyridine complex (**Ru22**–**Ru27**), aiming to obtain luminescent detectors of biomolecules [112,113].

In 2016, Wu’s research group introduced the concept of “photoresponsive metalloblock polymers” as an innovative approach for photo-controlled delivery of ruthenium(II)-based polypyridine complexes with the potential to treat cancer by PDT [79]. In the first report, they prepared three red-light-responsive self-assembled nanostructures of the synthesized PEG-*block*-poly(6-(4-cyanophenoxy)hexylmethacrylate) copolymers containing a Ru complex as a side group (**Ru28**–**Ru30**) [79]. The obtained micelles exhibited enhanced biocompatibility and cellular uptake, performing improved anticancer performance compared to the corresponding free metallodrug.

To build upon the attempts to prepare photoresponsive Ru^2+^-containing block polymers for PDT, the same research group reported a triblock copolymer containing Ru complexes in the main chain (**Ru31**) [114], which were more stable under physiological conditions than the free corresponding Ru complex [177]. The authors observed that the NPs of the metallopolymer prodrug **Ru31** efficiently accumulated at tumor sites, inhibited the growth of tumors under light irradiation and showed minimal systemic toxicity [114,178]. In a subsequent investigation, they synthesized a Ru complex linked with the commercial anticancer drug chlorambucil and conjugated it to side chains of PEG-modified polymer (**Ru32**) for photoactivated chemotherapy in a hypoxic tumor environment [56]. They observed that red light induces the release of the ruthenium conjugates and the drug chlorambucil, which inhibit tumor cell growth even in a hypoxic tumor environment and were non-toxic to mice. The prepared metallopolymer prodrug act for a mechanism known as photoactivated chemotherapy (PACT), which consists of a promising approach against hypoxic tumors [56]. More recently, Su’s research group continued the investigation of the Ru-containing block copolymers and prepared a metallopolymer prodrug similar to **Ru32** without the presence of chlorambucil (**Ru33**) [115]. They observed that while the correspondent free Ru complexes were unstable in the tested solutions, the NPs of the metallopolymers **Ru31** and **Ru33** were more stable under imitated physiological conditions. In addition, the Ru-containing block copolymers could be activated with red light, which could penetrate tissue deeply [115]. As an extension of these works, recently, the same group documented a dual-responsive Pt^2+^/Ru^2+^ bimetallic polymer to treat cisplatin-resistant tumors (**Ru34**) [101]. The authors employed PEG as a hydrophilic portion, while the hydrophobic block contained reduction-responsive Pt^4+^ and red-light-responsive Ru^2+^ moieties. The bimetallic-containing polymer self-assembles into NPs that were efficiently taken up by cisplatin-resistant cancer cells. Irradiation of cancer cells containing **Ru34** NPs generated ^1^O_2_, induced metallopolymer prodrug degradation, and triggered the release of the Ru^2+^ anticancer agent. Meanwhile, in the intracellular environment, cisplatin was released via reduction of the Pt^4+^ moieties. The synergistic effect of the combination of Ru^2+^ complex and cisplatin inhibited the growth of drug-resistant cancer cells as well as halted tumor growth in a mouse model since **Ru34** NPs could circulate in the bloodstream and accumulate at tumor sites [101].

In a similar direction, recently, Barz et al. prepared photocleavable core cross-linked polymeric micelles containing polypyridyl–ruthenium(II) complexes combined with polypept(o)ides (**Ru35** and **Ru36**) [116]. Block copolymers of polysarcosine-*b*-poly(glutamic acid) were synthesized and modified with aromatic nitrile groups on the glutamic acid side chain. The modified copolymers self-assembled into micelles and were cross-linked by Ru^2+^ complexes. The ruthenium cross-linked micelles displayed colloidal stability in human blood plasma. Analysis in HuH-7 tumor cells revealed increased cytotoxicity via micellar delivery of [Ru(bpy)_2_(H_2_O)_2_]^2+^ (bpy = 2,2′-bipyridine) but mostly irradiation damage for [Ru(biq)_2_(H_2_O)_2_]^2+^ (biq = 2,2′-biquinoline) that could not be completely released by light-induced cleavage [116].

In this regard, Gasser and coworkers also report covalent conjugation of a non-phototoxic Ru^2+^ complex (**Ru37**) to PLA by a drug-initiated ring-opening polymerization of lactide through the formation of a zinc initiator [179]. Compared to the free Ru^2+^ complex, the **Ru37** nanoconjugates exhibited enhanced photophysical properties and higher cellular uptake and, consequently, an improved photocytotoxicity.

#### 4.1.3. Organoruthenium Compounds

Considering the potentially rich field for the medicinal application of organometallic compounds, Barry, Sadler, and coworkers documented some works involving the incorporation of organoruthenium compounds into nanosized polymeric systems. Firstly, they report the encapsulation of highly hydrophobic organometallic ruthenium and osmium carborane complexes (**Ru38** and **Os1**) into PP123 core–shell micelles [180]. The entrapment of **Ru38** and **Os1** reduced their anticancer activities compared to the corresponding free organometallic compounds, but in return, the **Ru38** and **Os1**-loaded Pluronic^®^ micelles were more selective towards cancer cells compared to normal cells [180]. Shortly afterward, in another investigation, the authors observed that although less potent, **Ru38**-loaded micelles have a higher cellular accumulation of the organometallic compound and were sensitive to neutron irradiation in human ovarian cancer cells at a micromolar concentration in ruthenium [117]. Barry and collaborators also prepared caffeic acid-modified chitosan NPs covalently conjugated to an organoruthenium complex **Ru39** as a potential nanocarrier for Ru^2+^-arene compounds [80].

Nanostructured PP123 copolymeric micellar systems were also employed by Kyzioł and coworkers as nanocarriers to deliver two organometallic Ru^2+^ complexes with phosphine derivatives of fluoroquinolones (**Ru40** and **Ru41**) [118]. The prepared micelles presented high drug loading efficiency (ca. 95%), enabling efficient drug accumulation inside human lung adenocarcinoma cells.

### 4.2. Gallium Complexes

Besides platinum, gallium is the second metal ion to be inserted in clinical practice for cancer treatment. Gallium(III) nitrate is applied as a therapeutic agent in oncological diseases and disorders of calcium in bone metabolism; moreover, radioactive gallium (^68^Ga) also is used in clinical medicine as a diagnostic agent of cancer [181]. Two gallium-containing agents with considerable antineoplastic activity reached clinical trials: tris(8-quinolinolato)gallium(III) (KP46) [182]and gallium maltolate [183]. According to clinicaltrials.gov, accessed on 19 April 2022, gallium maltolate is currently in a phase-one clinical trial for the treatment of glioblastoma [183]. The mechanisms of action of the newer gallium-based agents are not yet completely understood, while the antineoplastic activity of the early generation of gallium compounds is related to the capacity of gallium(III) to mimic iron(III) ions [184]. Inside cells gallium(III) ions disturb iron-dependent proliferation processes by inhibiting ribonucleotide reductase enzyme and mitochondrial function [185].

The development of polymeric nanocarriers for gallium-based therapeutic agents is relevant considering that gallium compounds, generally, are poorly bioavailable when taken orally (early generation), have a lack of specificity, and have significant toxicity. However, nanostructured systems are not yet widely applied for gallium-based therapeutic agents’ release, although the presence few reports of the incorporation of gallium(III) compounds into polymeric nanosized carriers in the literature.

Mann and coworkers prepared micelles of PF127 to encapsulate KP46 (**Ga1**), which has low solubility under a physiological environment [52]. The **Ga1**-loaded micelles presented increased water solubility and similar cytotoxic activity in leukemia cell lines to that of conventional formulations of **Ga1** in DMSO, an organic solvent. Moreover, the **Ga1**-loaded micelles remained functional as cell cycle inhibitors, and cell death was characterized by a dose-dependent increase in apoptosis [52].

Nyokong et al. documented PF127 micelles for the incorporation of two Ga^3+^ porphyrins derivatives (**Ga2** and **Ga3**). The authors found that PF127 micelles improved the ^1^O2-producing ability of the porphyrins and, hence, their possible application as PDT agents [74]. In another investigation, the authors prepared PF127 micelles alone and functionalized them with folic acid (FA) for incorporation of a Ga^3+^ porphyrin derivative (**Ga4**) [71]. The authors observed that the FA-conjugated micelles resulted in some decreases in ^1^O_2_ quantum yields compared to non-conjugated micelles [71].

More recently, da Silva and collaborators related gallium(III)-phthalocyanine (**Ga5**) encapsulated into PEGylated PLGA NPs [64]. The outcomes of this investigation indicated that encapsulation of **Ga5** enhanced the cytotoxicity activity on the Hepa-1C1C7 cell in relation to the free compound and increased the photodynamic effect on the reduction of the viability of tumor cell. In addition, **Ga5**-loaded NPs were also more efficient in causing the hemolysis of the red blood cells than their free form [64].

### 4.3. Gold Complexes

Various gold compounds are currently being examined for their antitumor potential [186]. Gold(I) compounds, such as Auranofin^®^, which is commonly related to present anti-arthritic activity, have been recently investigated for their potential use as antitumoral agents. Gold(I) compounds can inhibit the activity of the mitochondrial protein thioredoxin reductase, an important target for controlling the progression of several chronic diseases, including rheumatoid arthritis and some cancers [187]. This new antiproliferative mode of action, not related to DNA, is encouraging given current treatment limitations employing the conventional anticancer agents. Moreover, gold(III) compounds also have attracted attention as antitumoral agents owing to the similarity of their square-planar coordination geometry to that of platinum(II) compounds [186]. Even though Au^3+^ compounds present high cytotoxicity toward various cancer cells, they commonly have low stability and low solubility/biocompatibility under physiological conditions [188]. Due to the poor water solubility and low stability of gold complexes, delivery systems can be appropriate alternatives to overcome these limitations.

In the literature, there are some examples of using nanosized polymer-based systems to increase the therapeutic index of gold-based drugs. Lin et al. developed pH-sensitive micelle-like NPs constituted of modified PEG-poly(β-amino ester), which were able to encapsulate a gold(I)-complex (**Au1**) and induce cancer cell death through regulation of autophagy [102]. Accardo and colleagues prepared micelles functionalized with an analog of the bombesin peptides anchored on their surface to act as a driver for specific receptors, which are commonly overexpressed in several cancer cells [119]. The micelles were incorporated with gold(III)-dithiocarbamato complex (**Au2**) [119]. Another example of functionalized micelles able to encapsulate poorly water-soluble, such as the complex **Au2**, was related by Nardon et al. [120]. The authors demonstrated that incorporating **Au2** into micelles enhanced its stability in saline solution and increased its water solubility while preserving or enhancing its antiproliferative activity over the free gold-based drug. Moreover, they also showed that functionalization on the hydrophilic shell of the micelles with a targeting moiety (the cholecystokinin octapeptide, CCK8) increased the cytotoxic activity on human epidermoid carcinoma cell line (A431) overexpressing CCK2 receptors about tenfold compared to free gold-complex or the complex encapsulated into non-functionalized micelle [120].

Employing covalent conjugation of gold(I)-complexes to the copolymer core, Pearson and collaborators developed two kinds of micelles for carrying auranofin [38,121]. The authors developed synthetic pathways to create polymers that mimic the structure of deacetylated auranofin (**Au3** and **Au4**). Then, they carried out a micellization process of the block copolymers containing pendant auranofin-like units. The outcomes indicated that the produced nanocarriers with the auranofin are retained in the aggregate cores, which protect the drug from ligand exchange and its consequent inactivation but not significantly enhance their cytotoxic activity [38,121]. Another example of covalent conjugation of gold(I)-complexes to a glycopolymer (**Au5**) was reported by Narain et al. [122]. The authors conjugated the cation triphenylphosphinegold(I), [AuPPh_3_]^+^, to hydrophilic copolymers using dithiocarbamate as a stabilizing ligand. They synthesized two amine-based glycopolymers at different molecular weights (10 and 30 KDa). The gold(I)-aggregates, particularly the aggregate with 10 KDa, showed higher accumulation and greater cytotoxicity toward different cancer cell lines without significantly affecting normal tissues [122].

On another note, Chung et al. documented two gold(III) porphyrin–PEG conjugates (**Au6** and **Au7**) that undergo self-assembly into core–shell micelles [47]. The nanostructures showed higher uptake into cancer cells than non-tumorigenic cells and released the anticancer gold(III) porphyrin moiety without the formation of toxic byproducts. At the same time, the produced **Au6**- and **Au7**-supramolecular aggregates could encapsulate doxorubicin (DOX) [47].

### 4.4. Copper Complexes

Copper is an essential element associated with numerous biological processes and plays a relevant role in diverse cellular functions [189]. Copper compounds display cytotoxicity similar to platinum-based compounds and generally possess lower systemic toxicity than platinum compounds [190]. These features suggest that copper complexes are suitable alternatives to platinum anticancer drugs. Several studies also identified that copper compounds perform mechanisms of action distinct from platinum-based therapeutic agents. While platinum-based drugs’ principal mode of action consists of establishing covalent binding to DNA, copper complexes can cleave the DNA strands due to their ability to catalytically generate reactive oxygen species (ROS) [191]. Copper-based compounds can still exert an inhibitory action on topoisomerases, a class of enzymes that participate in the regulation of DNA topology [192].

Karmakar et al. described the development of folic acid-tagged hydrophobic-modified chitosan NPs for target delivery of the poorly soluble copper acetylacetonate complex (**Cu2**) [123]. **Cu2**-loaded FA-conjugated chitosan NPs selectively target the folate receptor expressed in cancer cells.

Suntharalingam and coworkers showed that the encapsulation of the copper complex (**Cu3**) using PEG-PLGA copolymers enhances breast cancer stem cells (CSC) uptake and reduces overall toxicity. In addition, the **Cu3**-loaded NPs selectively kill breast CSCs over bulk breast cancer cells with a similar cellular response to the corresponding free complex [65].

Furthermore, we found three distinct investigations in the literature aiming to stabilize the copper(II) diethyldithiocarbamate complex (Cu(DDC)_2_) in polymer nanocarriers [66,91,103]. This complex formed by disulfiram, an alcohol-aversion drug in the presence of copper(II) ions, has significant anticancer activity and entered phase-one trials [193]. Although the Cu(DDC)_2_ complex presents expressive anticancer activity, including the inhibition of cancer stem cells, it has poor in vivo stability and rapid degradation, which significantly compromise its clinical outcomes [66]. In 2018, Gaspar and coworkers prepared polymeric NPs for physical encapsulation of copper diethyldithiocarbamate complex (**Cu4**) [66]. The authors prepared the drug-loaded micelles by two distinct approaches: the traditional one, employing the film-dispersion method, and a new procedure designed by them as SMILE (stabilized metal ion ligand complex). The authors observed drug-loading efficiency close to 100%, increased drug concentrations (over 200-fold compared to the traditional micelle formulation), and optimal particle size in the sub-100 nm range. Furthermore, NPs were stable in serum for 72 h and induced cell death in drug-resistant prostate cancer cells [66].

In the following year, He and his collaborators proposed a coordination strategy for loading Cu(DDC)_2_ (**Cu5**) into polymeric NPs [91]. The authors provided the coordination of Cu^2+^ to carboxylic groups of PEG-*b*-poly(ester-carbonate) for efficient loading of the copper complex, which was generated by the in situ reaction of the ligand disulfiram and Cu^2+^. The NPs of metallopolymer **Cu5** were stable in both neutral and weak acidic solutions and could enable a combination therapy of complex-based chemotherapy and chemodynamic therapy mediated by bioavailable Cu^2+^ that was not coordinated to the ligand. The in vivo antitumor results demonstrated that the copper complex-loaded NPs showed superior antitumor efficacy to the corresponding free complex [91]. In 2020, Ding, Zhou et al. employed a distinct strategy for constructing a self-delivery system for Cu(DDC)_2_ [103]. They prepared a nanoscale Cu(DDC)_2_ coordination polymer (**Cu6**), which was subsequently coated with hyaluronic acid (HA). The resulting NPs showed pH/GSH dual-responsive drug release due to the triggered breakage of the copper−sulfur coordinate bond. Glutathione (GSH) is an abundant natural tripeptide that performs vital roles such as antioxidation and could be found within almost all cells. However, in cancer cells, the level of GSH is elevated, which could cause resistance to a number of chemotherapeutic drugs [194]. The presence of HA allowed target delivery of Cu(DDC)_2_ into tumor cells. In addition, upon intravenous administration, the nanosystem passively accumulated into the tumor, and the in situ-released Cu(DDC)_2_ provided potent tumor growth inhibition at low drug doses [103].

Karmakar and collaborators prepared chitosan NPs for the target release of bis(2,4-pentanedionato)copper(II) (**Cu7**) [99]. For in vitro assays, two targeting strategies were employed. Folic acid or the HER2-specific peptide (which is a peptide overexpressed in many breast cancer cells) was attached to stearic acid-modified chitosan NPs loaded with **Cu7**. The authors observed that **Cu7**-loaded chitosan NPs induced ROS-mediated DNA and mitochondrial damage, followed by apoptosis. The targeting was further tested on in vivo mice, and promising results were observed since the tumor volume reduced significantly and the survivability of mice increased [99].

For copper-based therapeutic agents, target delivery nanosystems are interesting for minimizing the cytotoxic effects on normal cells and improving the anticancer properties. However, to date, there are only a few reports of encapsulation of anticancer copper complexes into polymeric nanocarriers. We hope that the positive results recorded in the investigations mentioned above can provide an additional inducement for developing new polymeric nanoformulations for other promising anticancer copper complexes.

### 4.5. Zinc Complexes

Zinc(II) is an essential metal ion for human health as a constituent of more than 300 enzymes, and, consequently, it is involved in many biological pathways [195]. Continuing research in the field of PSs, some complexes of zinc(II), due to their efficient capability of generation of ^1^O_2_ after irradiation, have been receiving increasing attention for PDT to treat cancer [196].

Zinc(II) phthalocyanine (ZnPC, **Zn1**) and its derivatives, for instance, are promising PSs for PDT due to their extremely high ^1^O_2_ generation ability, but they suffer from aggregation in a physiologic aqueous environment and have poor selectivity for tumor cells [193]. The self-aggregation of PSs in aqueous solutions severely decreases their photosensitizing efficiency limiting their direct use in biological fluids [196]. Fortunately, nanosized polymeric drug delivery systems could overcome these drawbacks since they tend to prevent aggregation, improve water solubility, and promote enhanced PSs accumulation in tumor cells. Thus, there are several studies wherein the highly hydrophobic **Zn1,** and its derivatives are incorporated into various nanosized polymeric systems.

In 1995, in a pioneering work, Leroux, Gurny, and collaborators reported the incorporation of hexadecafluoro zinc(II) phthalocyanine (**Zn2**) into three different polymeric vehicles for target delivery [18]. The authors employed PLA and PEG NPs, and Cremophor EL (CRM) oil-water emulsion. The authors observed that the PEG-coated NPs delivered **Zn2** into mammary mouse tumors. The two types of NPs investigated gave rise to significantly different distributions of **Zn2**-loaded NPs in organs [18]. Wu and collaborators synthesized a series of metallopolymer prodrug composed of zinc(II) phthalocyanine covalently conjugated to the copolymers poly(phthalocyanine-co-sebacic anhydride) (**Zn3**) [129]. They found that the rate of degradation increases with the pH and temperature and that the release occurred almost simultaneously with the degradation [129].

Marchetti and coworkers prepared micelles constituted of PEG_5000_-DSPE for encapsulation of ZnPC [125]. The obtained Zn1-loaded micelles provided better stability of the incorporated PS. In a second report, Marchetti et al. described PLGA NPs for encapsulation of Zn1 [62]. The authors observed that Zn1-loaded PLGA NPs had a sustained release rate. The author concluded that both nanosystems presented suitable conditions for their utilization in PDT and photodiagnostic procedures [62,124].

In a similar direction, Wilk and coworkers prepared PLA-*b*-PEG copolymer micelles for encapsulation of **Zn1** [125]. The obtained **Zn1**-loaded PLA-*b*-PEG micelles could effectively deliver the PS to the cancer cells with low toxic effects on control cells. Furthermore, the studies showed that **Zn1**-loaded PLA-*b*-PEG micelles have good biocompatibility [126]. In a recent report, Wilk, Kulbacka and collaborators engineered folate-functionalized micelles of metoxiPEG-*b*-PLA functionalized with FA for entrapment of **Zn1.** The **Zn1**-loaded functionalized micelles exhibited higher cytotoxicity toward cancer cells with overexpressed folate receptors than the unfunctionalized standard micelles. The folate-grafted *m*PEG-*b*-PLA nanocarriers uptake was proportional to the percentage of functionalization [127].

Tsai and collaborators reported two nanosystems for the delivery of **Zn1** and enhancing its photodynamic profile in cancer cells [42,128]. In the first report, they prepared bio-reducible heparin–polyethyleneimine (HPC) nanogel NPs for the encapsulation of **Zn1** and application as an imaging-guided PSs delivery vehicle in PDT. They observed **Zn1** release in vitro from disulfide-containing HPC nanogels in a redox-sensitive manner at low pH. The HPC nanogels prevented **Zn1** aggregation, thus enhancing ^1^O_2_ generation and PDT efficacy. Additionally, the HPC nanogels possess bright blue fluorescence, which eliminates the use of additional probing agents in image-guided drug delivery [128]. In the second report, they prepared pH-sensitive micelles composed of heparin, phospholipids, and histidine to act as a nanocarrier for **Zn1**. They observed that the release of **Zn1** from micelles was 91% in acidic conditions (pH 5.0) and 63% in physiological conditions (pH 7.4) after 96 h of incubation. The obtained micelles protect **Zn1** from aggregation and enhance its ^1^O_2_ generation. In addition, the **Zn1**-loaded in pH-sensitive micelles were up taken by HeLa cells and presented higher photodynamic activity than free **Zn1** [42].

Recently, Obata et al. investigated the effect of the hydrophobic segment of an amphiphilic block copolymer on micelle formation, **Zn1**-loading ability, and in vitro photocytotoxicity [129]. The results indicated that aromatic R groups form larger polymeric micelles with loosely packed **Zn1** aggregates, leading to enhanced release of **Zn1** in the cellular environment and, therefore, higher photodynamic effect than aliphatic R groups [128].

The combination of PDT and chemotherapy is an interesting strategy since the result in cancer cells will be the summation of the individual effects of each therapy. Hence, Quaglia and coworkers reported NPs constituted of amphiphilic block copolymers based on poly(ε-caprolactone) and PEO for co-delivery of docetaxel (DTX), a conventional anticancer drug, and the PS of ZnPC (**Zn1**) [48]. The authors observed ^1^O_2_ generation when **Zn1**-loaded NPs were irradiated, despite that **Zn1** was not released in the medium. This strategy of designing nanocarrier for combined chemo-photodynamic therapy of cancer was successful since it was observed superior antitumor activity in an animal model of **Zn1**/DTX-loaded NPs compared to DTX-loaded NPs [48]. A similar approach of photo/chemotherapy combination was adopted by Ji et al. [49]. They reported the co-encapsulation of another traditional anticancer drug, DOX, and zinc(II) phthalocyanine (**Zn1**) in thermal-responsive NPs. Encapsulation of **Zn1** significantly improves its aqueous solubility and increases the generation of ROS almost fivefold to the free **Zn1** [49]. Jin and coworkers designed a pH-responsive ZnPC metallopolymer nanocarrier **Zn4** for combining PDT and chemotherapy [130]. They used DOX as a chemotherapeutic agent. The obtained metallopolymer nanocarrier showed a high tumor-targeting effect, pH-responsive drug release, and significant anticancer activity [130].

Awruch and collaborators reported poloxamine polymeric micelles to encapsulate a derivative of ZnPC (**Zn5**) [131]. The author verified that **Zn5**-loaded NPs have the aqueous solubility increased up to 30 times and presented less aggregation in comparison to the free PS. Furthermore, **Zn5**-loaded NPs increase their photocytotoxicity by 10 times in comparison with the free form of **Zn5** [131].

Zinc(II) protoporphyrin complex (ZnPP) and its derivatives are also promising PSs for cancer treatment employing PDT, but the scarce solubility in aqueous solutions and the limited tumor selectivity of these compounds are factors that hinder their medical application. In this regard, Maeda’s research group made significant contributions to the engineering of suitable polymeric nanocarriers for this class of PSs. Initially, they developed PEG-conjugated ZnPP NPs (**Zn6**) and demonstrated that a PEGylated zinc derivative has good water solubility, inhibits heme oxygenase (HO) activity, and exhibits antitumor effects in vivo [132,133,197]. However, the critical drawbacks of PEGylated ZnPP were its poor loading capacity and undesirable high viscosity.

Therefore, pursuing the research in this field, Maeda and coworkers prepared nanosized micelles of amphiphilic copoly(styrene–maleic acid) (SMA) to encapsulate ZnPP (**Zn7**) by non-covalent method [134]. They obtained **Zn7**-loaded micelles with high loading capacity, water solubility, potent dose-dependent HO-1 inhibitory potentials, and the cytotoxic effect on human esophageal cancer cells [134]. In another research work, they demonstrated the synergistic effects of light induced photosensitizing capabilities and HO-1 inhibitory potentials of the PEG conjugated **Zn6** and **Zn7**-loaded micelles when tested in vitro and in vivo on tumor models under localized and mild illumination [198]. They used activated PEG in order to enhance the loading capacity in this last report [198]. Continuing the investigations about the properties of PEG conjugated **Zn6** and **Zn7**-loaded micelles, Maeda et al. found that **Zn7**-loaded SMA micelles have a higher and more efficient intracellular uptake rate than **Zn6** [199]. Besides this, they observed that both nanosystems were localized at the endoplasmic reticulum and inhibited the target enzyme HO-1 similarly [199].

Since ZnPP encapsulated in SMA micelles (**Zn7**) showed good pharmacokinetics and tumor accumulation, Maeda and coworkers decided to investigate the properties of ZnPP covalently conjugated to SMA copolymer (**Zn8**) [135]. The obtained metallopolymer **Zn8** was stable in circulation, had high tumor-targeted accumulation, and, consequently, exhibited a clear tumor imaging effect upon light irradiation. Thus, based on the properties of ZnPP as a photosensitizer and fluorescenceing agent, the prepared nanosystem was suitable for acting as a theranostic agent for cancer [135].

Along this line, Maeda’s research group also prepared ZnPP conjugated to *N*-(2-hydroxypropyl)methacrylamide (HMPA) (**Zn9**) and evaluated its application for tumor detection by imaging and treatment by light exposure using in mouse sarcoma model [136]. The micellar system of **Zn9**-conjugated with HPMA had a long plasma half-life and high tumor-targeting ability. The tumor-selective accumulation and the significant PDT effect in a murine sarcoma S180 model indicated that the prepared system is a potential theranostic PDT agent [200].

Following their previous research, Maeda, Fang, and collaborators investigated the impact of different chemical bonds in pegylation of ZnPP on drug release, intracellular uptake, and therapeutic effect in the tumor [137]. They prepared two types of PEG–ZnPP conjugates with different chemical bonds between PEG and ZnPP: ester bonds **Zn10** and ether bonds **Zn11**. They concluded that ester-linked conjugates manifested more efficient cleavage of PEG and greater yield of the active principle from the conjugates in tumor tissues than in normal tissues.

Dinarvand and coworkers prepared PLGA micelles for the non-covalent encapsulation of ZnPP (**Zn7**) [201]. ZnPP entrapment in PLGA NPs did not influence the HO inhibitory kinetics of free ZnPP, and the NPs exhibited potent cytotoxic effects on PC3 human prostate cancer cell lines as well as high cellular uptake [201].

Lee et al. developed PEG micelles, which covalently incorporate cinnamaldehyde, a ROS-generating agent, and ZnPP, an HO-1 inhibitor, in its backbone for combinational oxidation anticancer therapy (**Zn12**) [54]. The authors observed that **Zn12** micelles release cinnamaldehyde and ZnPP in acid-triggered manners and exhibit strong antitumor effects in a mouse xenograft model without conspicuous adverse effects. Additionally, the obtained micelles suppressed antioxidant HO-1 and generated ROS to synergistically elevate oxidative stress, giving rise to apoptotic cell death [54].

Caetano and collaborators reported spectroscopic studies of two zinc-porphyrins (**Zn13** and **Zn14**) encapsulated in two Pluronic^®^-based nanostructured systems (PP123 and PF127) [139]. The authors concluded by the values of K_b_ (binding constant) that the porphyrins interacted better with PP123 than PF127.

Gasser and coworkers documented the encapsulation of zinc(II) PS (**Zn15**) into micelles composed of DSPE-PEG_2000_ copolymers with biotin as a targeting moiety [139]. The generated NPs selectively accumulate in the lysosomes of cancer cells and fully penetrate the 3D tumor. Due to the biotin-targeting moiety, the **Zn15**-loaded NPs selectively accumulated in cancerous cells over noncancerous cells [139]. In a subsequent study, the authors use a similar polymeric system, DSPE-PEG_2000_-OCH_3_, for encapsulation of similar zinc(II) PS with the difference in the presence of four iodine atoms in the ligand (**Zn16**) [140]. The **Zn16**-loaded NPs showed improved water solubility and photophysical properties. In addition, the generated NPs were able to accumulate selectively in lysosomes, contrary to the complex itself, which accumulated in the cytoplasm. Upon light exposure, the loaded NPs caused cell death at very low micromolar concentrations in cancer cells as well as in a 3D tumor model [140].

Photothermal therapy (PTT,) compared to conventional cancer therapy approachs, represents an effective and noninvasive method for cancer therapy. This procedure can damage by heat site of interest in the presence of accumulated photothermal agents and with localized NIR laser without harming the surrounding healthy tissues [202]. In this line, Liu and coworkers reported a zinc(II) porphyrin-conjugated polymer (**Zn17**) encapsulated into DSPE-PEG_2000_-Mal for the highly efficient PTT [141]. The obtained and loaded NPs showed good biocompatibility and photostability and high photothermal conversion efficiency for in vitro and in vivo PTT. In addition, the Zn-loaded NPs surface was functionalized with cell-penetrating peptide (Tat) to enhance the cellular uptake [141].

Liu, Hsu, Lee, and coworkers also reported a zinc(II) porphyrin conjugated polymer (**Zn18**) encapsulated into a copolymer system with PEG for PTT [142]. The authors used a light-harvesting unit side group in polymer for enhancing photothermal conversion. The developed **Zn18**-loaded NPs presented excellent photothermal performance with a high value of photothermal conversion efficiency. In addition, the photoacoustic imaging signal in the tumor site was still strong even after 24 h post-injection [142].

### 4.6. Iridium Complexes

Iridium(III) complexes, including some organoiridium(III) “half-sandwich” compounds, have recently drawn considerable interest due to their significative anticancer activities, versatile photophysical properties, and reduced side effects [203]. Iridium(III)-based compounds have as an advantage the possibility of developing dual-functional anticancer agents and luminescent probes due to their unique spectroscopic properties. Indeed, several iridium(III) complexes bearing ligand-engineered have been incorporated into functionalized NPs that had been exploited as the PSs and/or chemotherapeutic agents for cancer treatment.

Chen and coworkers contributed three papers to the research field of incorporating iridium complexes with anticancer potential into nanostructured polymeric systems [145,146,147]. In the first report, they conjugated an iridium complex to a polymeric chain (**Ir1**) [145]. They used the chloro-bridged dimer [Ir(pq)_2_]_2_Cl_2_ (pq = 2-phenylquinoline) as reagent for polymerization with olygolysine. They observed that the luminescence of the obtained metallopolymer NPs enhanced when they were internalized by cancer cells. When incubated with HeLa cells, the NPs went through lysosome, cytosol, and finally into the nucleus of the cancer cells [145]. In a subsequent investigation, the authors used the same iridium binuclear complex, [Ir(pq)_2_]_2_Cl_2_, but employed PEG and poly(4-vinylpyridine) polymers functionalized with FA for encapsulating iridium complex (**Ir2**), facilitating its intracellular release [146]. The iridium-loaded NPs with FA on their surface were able to enter HeLa cells with folate acceptors overexpressed, and, after cellular internalization, the intracellular compound histidine could trigger the release of the [Ir(pq)_2_]^+^ residue into the nucleus from the NPs [146]. Continuing the research, they reported other iridium(III) complex covalently entrapped in PEG-benzoic imine-oligo-L-lysine NPs (**Ir3**) functionalized with FA for nucleus-targeted drug delivery [147]. Compared to the free iridium complex, the prepared nanosystem improved the cytotoxicity in vitro by about 20-fold and enhanced tumor suppression in a mouse model [147].

Wenbin et al. reported on the polymeric systems constituted of PEG-*b*-poly(glutamic acid) for conjugation to two iridium and one ruthenium DNA targeting organometallic compounds (**Ir4**, **Ir5**, and **Ru42**). The obtained micelles selectively released the loaded organometallic compounds within the endosomes and lysosomes in a sustained fashion. In addition, the in vitro cytotoxicity of the obtained metallopolymers against four ovarian cancer cell lines, including one cisplatin-resistant cell line, showed higher cytotoxicity activities compared to that of cisplatin [78].

Zhao, Huang, and collaborators concerned about the deficient O_2_ supply in hypoxic tumors, which is needed for an effective PDT, designed a multifunctional water-soluble polymer conjugated to naphthalene and iridium(III) complex (**Ir6**) to deliver extracellular ^1^O_2_ into cancer cells to induce the oxidative damage without intracellular O_2_ consumption [148]. The obtained metallopolymer could carry ^1^O_2,_ and the release of ^1^O_2_ was triggered by the photothermal effect of Au nanorods under near-infrared (NIR) light irradiation, promoting a controllable therapy combining both oxidative and photothermal damage [148].

Lu, Fan, and coworkers designed a water-soluble phosphorescent polymer conjugated to the Ir^3+^ complex (**Ir7**) for tumor-targeted PDT [149]. The brush structure rendered PPF-Ir-*g*-(POEGMA-*b*-PGal) exhibited excellent water solubility and efficient ^1^O_2_ generation. Additionally, the obtained nanosystem could competently cause the death of Hep G2 cancer cells in vitro and could significantly inhibit Hep G2 tumor-bearing nude mice in vivo [150].

Very recently, Bokias et al. reported on the complexation of the Ir^3+^ complex with a vinylic unit containing quinoline-pyridine groups (VQPy) by two synthetic methodologies, namely pre- or post-polymerization. Two water-soluble polymeric Ir^3+^ complex nanomaterials were obtained: fully complexed VQPy units (by prepolymerization methodology), **Ir8**, and terpolymers containing free and Ir^3+^-complexed VQPy units (by post-polymerization methodology), **Ir9**. The authors mostly focused on the optical properties, especially the emission, of the novel metallopolymers in an aqueous solution. **Ir9**-conjugated copolymers showed characteristic pH-responsive color changes under UV illumination in the acidic pH region [150].

### 4.7. Other Metal-Based Compounds

Some compounds with metallic centers distinct from the above-mentioned have also been evaluated by their anticancer therapeutic potential.

Chloroaluminium phthalocyanine (**Al1**) is a PS with suitable photophysical properties for PDT, but its insolubility in water restricts its possible medical applications. To overcome this limitation, the use of polymeric nanocarriers has been an applicable strategy. In 2001, Leroux and coworkers prepared hydrophobically-modified copolymers of *N*-isopropylacrylamide bearing a pH-sensitive moiety for encapsulation of **Al1** for PDT [19]. **Al1**-loaded micelles were found to exhibit increased cytotoxicity against EMT-6 mouse mammary cells in vitro than the control Cremophor EL formulation [19]. In this sense, Mosqueira and coworkers prepared three kinds of nanocapsules consisting of PLGA, PLA, and PLA–PEG and a type of nanosphere of PLA for encapsulation of **Al1** and evaluated their release profiles and phototoxicity in vitro [89]. PLGA and PLA–PEG nanocapsules resulted in a sustained **Al1** release profile. In vitro cell studies revealed that the photoactivity of **Al1** was improved by encapsulation [89]. Soler and collaborators reported a nanosystem based on PF127 micelles also for incorporation of **Al1** [84]. A549 human lung carcinoma cell line incubated with low dosages of **Al1**-loaded PF127 micelles, followed by 18 min of light irradiation (660 nm) showed high cellular damage. Meanwhile, no cytotoxicity occurred on non-irradiated cells [84].

Chen and coworkers reported two distinct acid-triggered micellar systems for synergistic chemo-photodynamic therapy employing DOX, a hydrophobic anticancer drug, and iron(II) tetraphenylporphyrin (FeTPP) [90,151]. In the first work, they designed micelles obtained by the self-assembling histidine modified-PLA and PEG conjugated to FeTPP (**Fe1**) [90]. Following this line, in another report, the same research group proposed a micellar system assembled by flexible polycaprolactone modified with histidine and PEGylated metalloporphyrin (**Fe2**) [151]. In summary, both nanosystems could, in the neutral environment, effectively co-encapsulated DOX and the PS FeTPP improving their water solubility; while in the intracellular microenvironment, the changed acid environment would trigger the drug and FeTPP release due to the reduction of metal-coordinated interaction between histidine and FeTPP leading to the disintegration of assembles. In addition, the micellar systems showed high in vitro cytotoxicity toward HeLa and MCF-7 cells, confirming that synergistic chemo-photodynamic therapy could effectively enhance cellular proliferation inhibition [90,151].

Arbiser and collaborators, while aiming to develop chemotherapeutic NPs to target receptors overexpressed in most advanced melanomas (IGF1R and CD44), prepared hyaluronic acid (HA) NPs for physical encapsulation of an active organopalladium compound (**Pd1**) [152]. They prepared HA NPs containing either IGF-1 to target IGF1R expressing melanomas or HA NPs not containing IGF-1. HA itself targets CD44, a cell surface marker associated with highly malignant behavior. Curiously, the authors observed that the **Pd1**-loaded HA NPs without antibodies to IGF1R were more effective in vivo, likely due to the competition for cell targets or steric hindrances [152].

Abedi and coworkers investigated the biological activity of the new cobalt(III) complex (**Co1**) in its free form. They encapsulated it into a nanocarrier composed of PLGA–PEG di-block copolymer [153]. They observed that **Co1**-loaded PLGA–PEG NPs have pH-dependent release. In addition, **Co1**-loaded PLGA–PEG NPs presented enhanced antiproliferative activity toward Gram-positive and Gram-negative bacteria and against ovarian cancer cells (A2780) compared to the free complex [153].

Eskandaria and Suntharalingam reported an active manganese(II) complex (**Mn1**) and its encapsulation into PEG-PLA nanoparticles in an attempt to improve its delivery [154]. Mn1-loaded nanosystem delivered 15-fold higher levels of manganese into cancer stem cells (via an endocytotic pathway) than the free payload. In addition, the obtained nanosystem significantly reduced overall toxicity toward bulk breast cancer cells (HMLER) and breast CSCs (HMLER-shEcad) [154].

## 5. Metal-Based Agents Incorporated into Polymeric Nanocarriers Applied for Other Therapeutic Approaches beyond Cancer Treatment

Some investigations have revealed that the polymeric nanocarriers could be successfully expanded for the incorporation of metal-based agents for the treatment of other diseases beyond cancer. In this regard, the great potential of polymer incorporation in enhancing the antimalarial activities of metal-based agents was demonstrated by Garcia et al. [159]. They investigated the capacity to block the viability of the lethal human malaria parasite *Plasmodium falciparum* of six encapsulated metalloporphyrins-IX, MPPIX, (Fe^2+^, Co^2+^, Cu^2+^, Mn^2+^, Ni^2+^, and Zn^2+^) in micro- and nanocapsules of marine atelocollagen, a polyamino-polymer. The effects of the loaded-metalloporphyrins-IX on cultures of red blood cells infected with *P. falciparum* showed a toxic effect at nanomolar concentrations. Zn^2+^-protoporphyrin-IX (ZnPPIX) showed the best antimalarial effect, with an 80-fold increase in the antimalarial activity compared to the antimalarial activity of non-encapsulated ZnPPIX [159].

Taking advantage of the polymer nanocarriers’ capacity to have good skin permeability, which particularly favors the treatment of psoriasis, Jin and coworkers designed nanostructures of an amphiphilic zinc phthalocyanine polymer conjugate (**Zn19**) for PDT of psoriasis [143]. Psoriasis is a chronic inflammatory skin disease that severally affects the quality of life of 2–5% of the population worldwide. The ZnPC was conjugated with the PEG chain. The metallopolymer nanostructures had good water dispersiveness, and light-responsive abilities and presented significant anti-psoriasis activity in psoriasis animal models [143].

Photodynamic inactivation of microorganisms (PDIM) is considered an innovative approach to causing irreversible damage to pathogens after light exposure. As classical PDT, PDIM needs a PS, oxygen, and light to generate ROS. Generally, the PSs applied in PDIM have poor water solubility, which could be improved by their incorporation into polymeric nanocarriers. In this line, Xu et al. reported NPs constituted of poly(glycidyl methacrylate) covalently conjugated to zinc(II)-based PS (**Zn20**) for PDIM [144]. The Zn-conjugated NPs exhibited potent photodynamic antibacterial activity, and the obtained promising results are an initial step for engineering self-sterilizing surfaces to combat bacterial infections. Another example of the incorporation of metal-based PS into polymeric nanocarriers for PDIM is presented in Table 1 as **Ru37** [63]. This investigation showed that **Ru37** nanoconjugates promoted a modest photo-bactericidal activity on Gram-positive bacterial strains, while the correspondent Ru^2+^ complex in its initial form was inactive in bacterial cells in the dark and under light irradiation [63].

Still, in terms of antimicrobial activity, some polymeric NPs were developed for encapsulating silver carbene complexes (SCCs). SCCs have significant antimicrobial activity in vitro but have their use limited in vivo due to the amount of chloride in the bloodstream that precipitated the SCCs as silver chloride and the presence of sulfur-containing proteins that inactivated the SCCs [155]. The encapsulation of SCCs into poly(acrylic acid)-*b*-polystyrene, PEG-PLA, and PEG-PLGA NPs demonstrated potential for treating urinary tract infections and other bacterial infectious diseases [155,156,157], SCCs compounds were generically designed in Table 1 as **Ag1**. Furthermore, other silver(I) complexes **Ag2** encapsulated into polymeric NPs also have demonstrated potential as a therapeutic strategy for the control of *Helicobacter pylori* infections [158].

Another interesting approach is employing polymeric NPs coupled with iron chelators as a potential therapeutic strategy for treating neurologic disorders associated with trace metal imbalance. The suitable design of the NPs chelator system is essential to transport chelators and chelator–metal complexes in both directions across the blood–brain barrier, providing safer and more effective chelation treatment for Alzheimer’s disease and other neurodegenerative disorders [204,205].

In addition, the literature relates some works involving polymeric NPs loaded with metallic radionuclides for biomedical applications beyond cancer diagnostics and therapy. Polymeric NPs loaded with metallic radionuclides can be designed by the incorporation of a radioactive compound into polymeric nanostructures or by attaching the radioactive element to a chelator anchored on NPs’ surface [206]. Radioactive polymeric NPs provide a remarkable tool for theranostic applications for many diseases beyond cancer, including cardiovascular, inflammation, and infectious diseases [206,207,208].

Polymeric NPs can be a prospective solution to the challenges found in vivo application of metal-based drugs for the diagnostic and treatment of an assortment of diseases. The functionalization of nanocarriers can broaden their capacities for the delivery of metal-based therapeutic agents to different biological targets; however, the engineering of nanocarrier assisted-delivery systems is a challenging task that demands advanced research. In addition, although still not vastly explored in this field, metallopolymers can also be designed to treat many disorders beyond cancer.

## 6. New Trends in Nanotechnology Involving Polymeric Nanosized Systems and Metal-Based Compounds

This section illustrates recent trends in medicinal inorganic chemistry associated with nanotechnology employing polymers with considerable potential to design innovative generations of metal-based therapeutic agents.

### 6.1. Metal Coordination-Driven Polymeric Nanosystems for Drug Delivery

Metal coordination-driven composite systems have excellent stability and stimulus-responsive abilities [209]. In metal coordination-drive systems, the metal ion acts as part of the nanocarrier by conjugating to a classical organic drug and subsequently delivering it to a specific site. The lability of coordination bonds facilitates the fabrication of a new generation of metallopolymers with pH-, enzyme-, light-, or redox-responsive properties that may be used to control the targeted delivery of drugs [210]. Recently, Tan et al. exploited this approach to prepare conjugated polymer NPs based on copper(II) coordination for real-time monitoring of pH-responsive drug delivery [88]. They prepared an anionic conjugated polymer with a poly(p-phenylene ethynylene) backbone and iminodiacetic acid side chains (PPEIDA). Copper(II) ions established coordination interaction with the prepared polymer and the drug DOX. The resulting metallopolymer prodrug (Figure 6a) exhibited good stability, high drug-loading, low drug-leakage in a neutral environment, pH-responsive drug release under an acidic environment, and more potent cytotoxicity against tumor cells than against normal cells. In addition, the obtained nanosystem had suitable fluorescence properties useful for the real-time monitoring of drug uptake in cells [88].

Yin and Tao also used the strategy of coordinating a drug to a polymer for controlled and target delivery of the active principle. They prepared redox-responsive metallopolymers of dopamine-modified HA with copper(II) and 6-mercaptopurine (6-MP) for targeted drug delivery (Figure 6b) [211]. While the drug release rate was slow in the medium without glutathione (GSH), the accumulated release amount was approximately 94% in 2 h in the medium containing 10 mM GSH. Since the content of GSH in tumor cells is higher than in normal cells, favorable drug release in the presence of GSH is an important characteristic for the stability of the drug in systemic circulation and for reducing its side effects. The prepared nanosystem could be internalized by tumor cells through the CD44-mediated endocytosis process, and the cytotoxic activity against tumor cells was higher than that of the free 6-MP [211].

### 6.2. Polymeric Nanocarrier for Delivery of Free Metal Ions

Chemodynamic therapy (CDT), which employs Fenton and Fenton-like reactions to produce the highly cells deleterious ROS, consists of a promising strategy that has been vastly studied in the last seven years for cancer treatment [212,213]. CDT takes advantage of the specific tumor microenvironment, which involves acidity and the overproduction of hydrogen peroxide since these conditions are suitable for Fenton processes. The ROS production in the CDT is triggered by the intracellular conditions and the provided excess amount of Fenton-active metal ions. In this regard, some research groups dedicated efforts to design targeting and biocompatible nanocarriers for release into cancer cells of transition metal ions with Fenton activity. Target delivery is crucial since administering an overdose supply of Fenton-type heavy metals in the body may cause potential damage to human health.

Furthermore, Hu, Chen, and coworkers designed a self-assembled nanosystem of DOX and Fe^2+^ with drug-metal coordination interactions with surface decoration of multiarmed PEG-dipyridine [214]. The resulting micellar system provided high drug-loading capability and stability in physiological conditions and showed facilitated intracellular co-delivery of Fe^2+^ and DOX. Moreover, intracellular overload of Fe^2+^ significantly enhanced the generation of ROS via the Fenton reaction, exhibiting potential for synergistic chemo/chemodynamic therapy [214].

Chen, Chen, and coworkers adopted a strategy that uses an intracellular labile iron pool as the endogenous source of Fenton-reactive metals encapsulated in pH-sensitive polymer PEG-poly(2-(diisopropyl amino) NPs to enable enhanced antitumor CDT [215]. In another report, they designed a pH/ROS dual-responsive nanosystem consisting of β- lapachone, pH-responsive polymer, and ROS-responsive polyprodrug [216]. The released Lap generated hydrogen peroxide, which was further converted into highly toxic hydroxyl radicals via the Fenton reaction. Moreover, the DOX was released by ROS-induced cleavage of the thioketal linker from the polyprodrug. Tests in vivo indicated that the cascade of ROS generation and anti-tumor drug release inhibited tumor growth [216].

Very recently, He, Li, and coworkers documented an efficient anticancer nanosystem employing an elegant and innovative strategy [217]. They prepared a highly biocompatible and light-responsive NPs by incorporating PEGylated polygalacturonic acid as a carrier, a porphyrin derivative as a PS, and a low content of Fe^3+^ for ferroptosis. Ferroptosis is an alternative mechanism of inducing cell death that has recently received much attention. The prepared nanocarrier could efficiently load cationic porphyrin derivative as well as the Fe^3+^ ions by electrostatic interactions and the chelation effect of the Fe^3+^ to the carboxylic acid groups in the polygalacturonic acid chain [217].

### 6.3. Polymeric Nanosystems Conjugated to a Metallic Ion Chelator

Recently, Jiang and coworkers proposed an elegant strategy for chelating Cu^+^ from tumors, which consists of an effective and promising approach for cancer therapy through antiangiogenesis [218]. Concerned that the systematic removal of Cu ions by chelators results in unavoidable side effects since Cu is indispensable to the body, they designed a targeted and innovative micellar nanosystem that could co-deliver a Cu^+^-chelator and DOX, a chemotherapeutic agent, for tumor treatment. The micelles were built from a body-friendly polypeptide amphiphile anchored with cRGD that can target the neoangiogenesis of tumors. The chelator moiety also acted as an “OFF-to-ON” type tracer for Cu^+^ (Figure 7). The obtained micelles could target and accumulate into the tumor to deliver DOX and the chelator. The chelator could capture in vitro and in vivo Cu^+^ to give NIR fluorescence and photoacoustic signals for dual-mode real-time and non-invasive reporting of the events of Cu^+^-caging. After treatment, both Cu level and oxygen saturation were significantly lowered in tumors [218].

### 6.4. Polymer Dots Applied as Phototheranostic Agents

In the last decade, polymer dots (Pdots) have emerged as an attractive approach for the engineering of nanoplatforms for theranostic applications [219]. The fascinating characteristics of these materials are due to their extended 
π
-conjugated backbones and delocalized electronic structure, which impacts their optical properties, such as absorption and emission spectra, light-harvesting ability, and quantum yield [219,220]. The reactive functional groups on the Pdots surface offer the possibility to conjugate them with a variety of recognition elements, such as sugar, small-molecule ligand, and an antibody, to construct multifunctional nanomaterials for imaging, drug delivery, and therapy.

Taking advantage of the outstanding optical properties of Pdots, Zhao and coworkers used Pdots conjugated to iridium complexes for real-time images and PDT [221]. They designed hyper-branched phosphorescent conjugated polymer dots with iridium(III) complex (Figure 8a) as the core for both hypoxia imaging and PDT. The polymer poly(styrene-co-maleic) anhydride was incorporated into the phosphorescent conjugated polymer dots by coprecipitation to provide the negative charges and ensure the water-solubility and biocompatibility. The polymer dots designed could efficiently generate the ^1^O_2_ in an aqueous solution, and the image-guided PDT of the cancer cells was successfully realized and investigated in detail by a confocal laser scanning microscope [222]. Another successful example was reported by Huang, Zhao, and coworkers, who designed two ultrasmall (<10 nm) phosphorescent conjugated polymer dots with Ir^3+^ complex (Figure 8b) for ratiometric oxygen sensing and PDT [223]. An efficient energy transference from the polymer main chain to the phosphorescent Ir^3+^ complex was observed, and the obtained Ir^3+^-Pdots could act as an optical probe for monitoring oxygen in an aqueous solution with high sensitivity. In addition, in vitro experiments, Ir^3+^-Pdots have low cytotoxicity, and most of them can pass across the cell membrane to enter the cytoplasm. The ^1^O_2_ photo-generated from the Pdots (under irradiation at 488 nm) could induce the apoptosis and death of tumor cells for PDT [223]. Continuing the field of investigation, the same research group reported dual-emissive semiconducting polymer nanoparticles containing fluorescent BODIPY derivatives and near-infrared phosphorescent iridium(III) complexes [224]. The obtained nanoassemblies achieve high PDT efficiency as well as excellent O_2_ imaging capability.

Wu and collaborators employed Pdots covalently conjugated to Pt^2+^ porphyrin, an oxygen-sensitive, for hypoxia bioimaging (Figure 8c) [225]. The obtained Pdots incorporated with metalloporphyrin provided in vitro and in vivo remarkable hypoxia sensing with high sensitivity and full reversibility, showing potential applications in cancer diagnosis [225]. Huang, Zhao, and coworkers prepared phosphorescent conjugated Pdots containing Pt^2+^ porphyrin as an oxygen-responsive phosphorescent group and ^1^O_2_ photosensitizer. The obtained Pdots exhibited a high PDT efficiency [226].

### 6.5. Polymer Functionalized Upconverting Nanoparticles

A variety of nanomaterials employing polymer on their surface has been studied as smart nanoplatforms for enhancing therapeutic proprieties of metal-based agents. Among them, the upconverting nanoparticles (UCNPs), which have exceptional optical and chemical properties, have been intensely investigated for application as drug carriers and imaging probes in medicine. This class of material consists of lanthanide ion-doped crystals with the ability to absorb and convert low-energy near-infrared (NIR) into high-energy UV-visible light [221]. The current UCNPs-based drug delivery systems mainly focus on PDT photosensitizers, while there are only a few reports on UCNPs-based chemotherapeutic systems [227]. This new tendency of using UCNPs-based materials for delivery of PS is due to the higher tissue penetration depth of NIR and the fact that it is less harmful to health than using the UV directly.

Furthermore, the surface modification of UNCPs with polymers is an interesting strategy for modulating some properties of UNCP-based materials, such as biocompatibility, light and pH sensitivity, water-dispersibility, and further providing bioactive chemical moieties for bioconjugation and biofunctionalizations [228]. Some examples of these strategy will be treated sequentially.

Following this concept, Jiang, Wang, and coworkers prepared a core–shell hybrid nanosystem consisting of an amphiphilic Ru(II) complex, *cis*- [Ru(bpy)_2_(C_18_H_37_CN)_2_]^2+^ (bpy = 2,2′-bipyridine), embedded along with a PEG-modified UCNPs [229]. The resulted in UNCP-based material presented good water dispersibility, stability in the dark, and NIR-activated release of a DNA covalent-binding agent [227].

Zhang, Kong et al. aiming to meet the demand for high ^1^O_2_ production yield, prepared a nanophotosensitizer constituted by UCNPs covalently assembled to ZnPC and FA [230]. The resulted high ^1^O_2_ production shall lead to a secure and efficient image-guided PDT treatment. In addition, neither pathological changes nor inflammation was observed in various organs of tumor-bearing mice after in vivo PDT treatment following intra-tumoral direct injection [230].

Lin, Yang, and collaborators designed *trans*-platinum(IV) prodrug-conjugated UNCPs NIR light-triggered for imaging and cancer therapy [231]. The polyethylenimine and PEG were anchored on the UNCPs surface, which effectively delivered the platinum(IV) pro-drugs into the cells by endocytosis. The tumor inhibition by prodrug-conjugated UNCPs was more prominent under NIR irradiation than under direct UV irradiation. The obtained nanocomposite could also be used as multimodality bioimaging contrast agents and transducers by converting NIR light into UV for control of drug activity in practical cancer therapy [231]. Salassa and collaborators prepared core–shell UNCPs for mediating the NIR-photoactivation of Pt^4+^ complex, *cis*, *cis*, *trans*- [Pt(NH_3_)_2_Cl_2_(O_2_CCH_2_CH_2_CO_2_H)_2_] [232]. The UNCPs were surface functionalized with the biocompatible PEGylated phospholipid DSPE-PEG(2000)-NH_2_ for loading the cisplatin precursor. The novel photoactivatable nanomaterials obtained were capable of releasing Pt^2+^ species upon NIR light excitation [232].

Gou, Xu, Fang et al. aiming to improve the light absorption capacity of Ir(III) complexes, developed an Ir^3+^-containing amphiphilic block polymer (PEG_400_), UNCPs are included in the polymeric micelles to permit NIR excitation [229]. The Ir^3+^-nanoassemblies exhibit efficient ^1^O_2_ generation, insignificant dark toxicity, excellent tumor-targeting ability, and synergistic phototherapy–chemotherapy effect both in vitro and in vivo [229].

Ford, Zheng, and collaborators prepared UNCPs, core encapsulated by an amphiphilic phospholipid-functionalized PEG (DSPE-PEG_2000_) polymer, containing the complex *trans*-[Mn(bpy)(PPh_3_)_2_(CO)_2_] as a unique strategy for CO delivery to biological targets [233]. CO releasing moieties (CORMs) are employed for the treatment (or prevention) of ischemia/reperfusion injury and other therapeutic applications. The resultant water-soluble photo-CORMs nanosystem was able to release CO under NIR excitation [233].

## 7. Conclusions

Metal-based agents possess unique attributes that have led to the development of promising metallodrugs and photosensitizers for chemotherapy and photodynamic therapy for the treatment of cancer and other diseases. However, to be suitable for medicinal applications, metal-based agents must meet some requirements, such as stability in the physiological, and environmental conditions, selectivity for tumor tissues, and reasonable water-solubility. Added to this, metal-based PSs need to have low dark cytotoxicity. In this context, associating metal-containing therapeutic agents with polymeric NPs permits reaching the features mentioned above and provides benefits beyond the well-known advantages of drug delivery systems.

From the biological viewpoint, the metallodrug-loaded polymer may improve the drug pharmacokinetics, ensuring that the active compound reaches its pharmacological target with a controlled dosage and rate of the released drug. However, further research is necessary to comprehend the molecular mechanisms that drive NPs to cells. Furthermore, from the chemistry point of view, finding ways to synthesize stimuli-responsive polymers for targeted delivery and regulated release of metal-containing agents with well-controlled molecular weights and architecture is a major challenge in the field. Even for non-covalent incorporation, which is synthetically simpler than the covalent conjugation method, undesired reactions leading to structural changes of metal coordination compounds can also occur during the encapsulation process. Thus, the engineering of metallodrug nanocarriers requires rationally designed copolymer blocks with well-controlled synthetic procedures and wide characterization methods of the obtained biomaterial systems.

Despite the challenging of this field, some ground-breaking advances have been achieved in the preparation of stimuli-responsive polymeric nanosized drug delivery systems, which gradually provide hope in the search for more appropriate and less aggressive cancer therapy. Furthermore, the improvements reached for cancer therapy can also be expanded for the treatment of other diseases, creating an innovative generation of metal-based therapeutic agents.

## Figures and Tables

**Figure 1 pharmaceutics-14-01506-f001:**
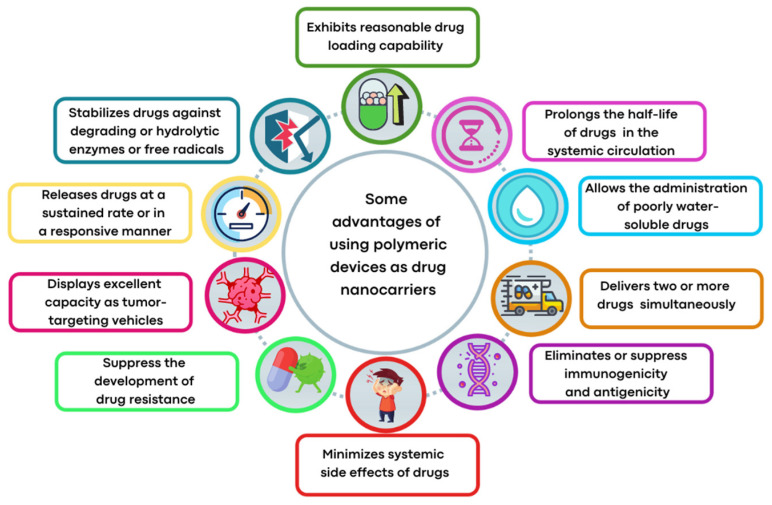
Some advantages of using polymeric nanocarriers to deliver metal-based therapeutic agents.

**Figure 2 pharmaceutics-14-01506-f002:**
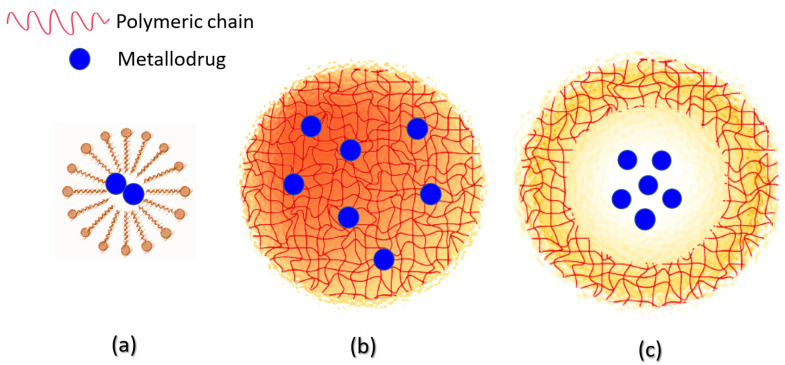
Schematic representation of polymer-based nanocarriers: (**a**) nanomicelle (constituted by a hydrophilic shell and hydrophobic core where the metallodrug are entrapped), (**b**) nanosphere (polymeric matrix where the metallodrug is uniformly dispersed), and (**c**) nanocapsule (the metallodrug is entrapped in a cavity that is surrounded by a polymeric membrane).

**Figure 3 pharmaceutics-14-01506-f003:**
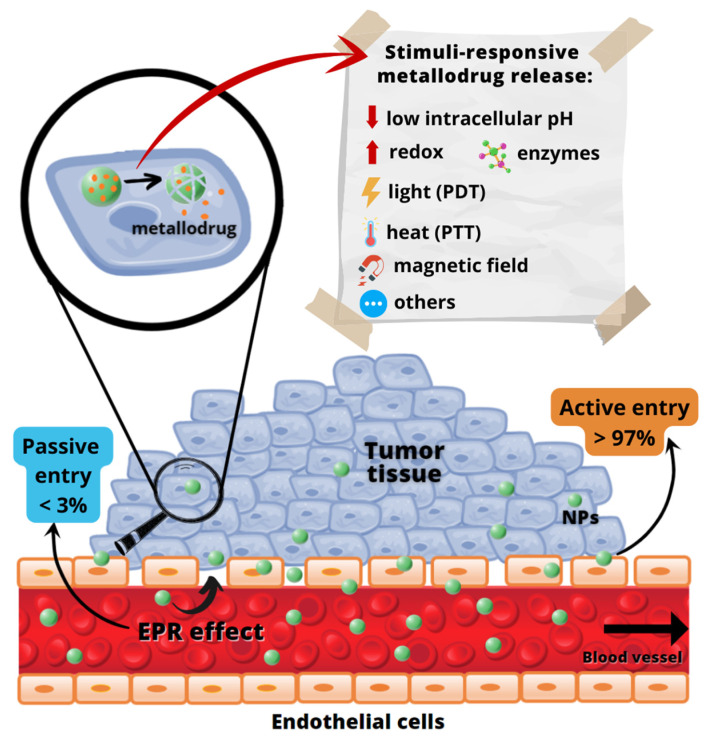
Schematic illustration of the passive and active pathways of nanocarriers’ internalization into tumor cells and controlled metallodrug release from NPs by intrinsic and/or extrinsic stimuli.

**Figure 4 pharmaceutics-14-01506-f004:**
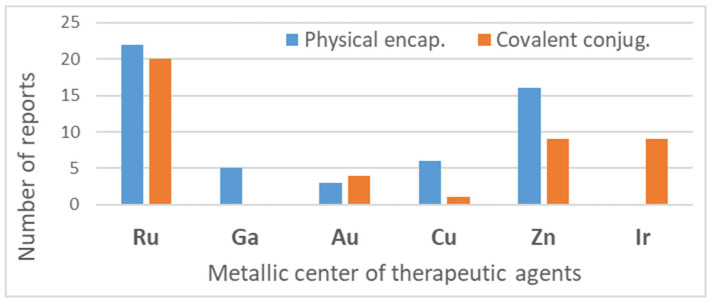
Distribution of noncovalent and covalent incorporation of metal-based therapeutic agents into polymeric nanocarriers.

**Figure 5 pharmaceutics-14-01506-f005:**
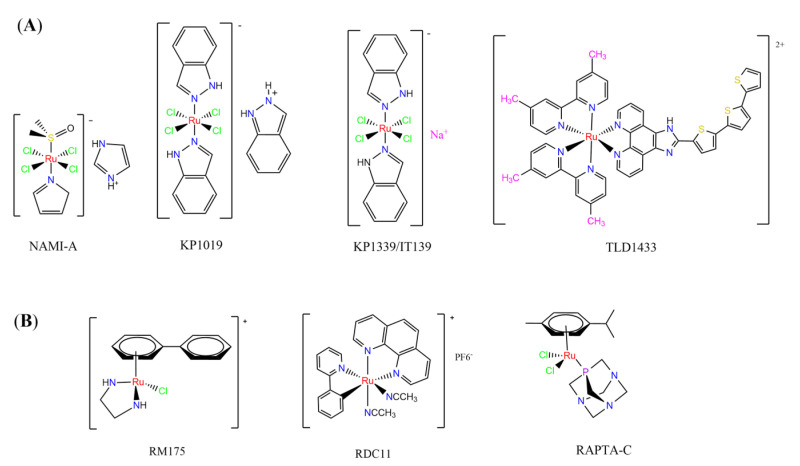
Chemical structures of ruthenium-based compounds that have entered clinical trials.

**Figure 6 pharmaceutics-14-01506-f006:**
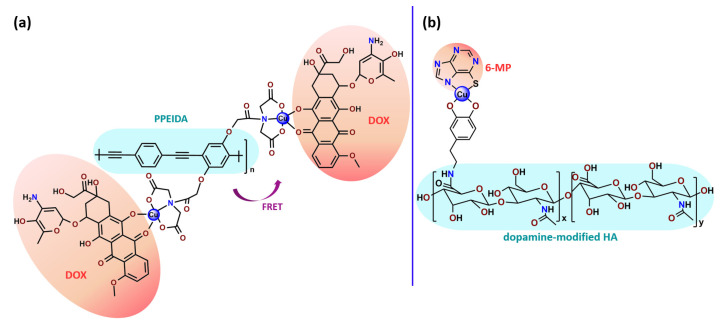
Examples of metal coordination-driven polymeric nanosystems for drug delivery: (**a**) DOX and (**b**) 6-MP delivery [88,211], respectively.

**Figure 7 pharmaceutics-14-01506-f007:**
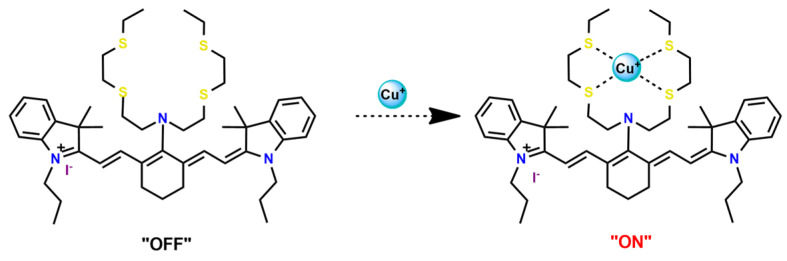
Structure of “OFF-to-ON” probe to Cu^+^ capture [218].

**Figure 8 pharmaceutics-14-01506-f008:**
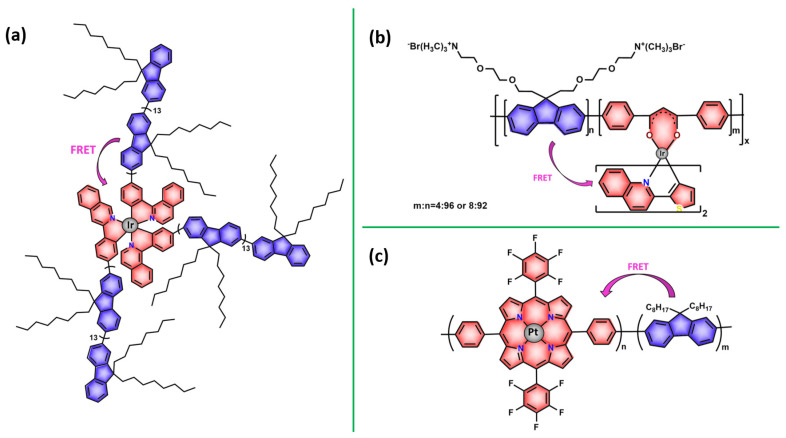
Some examples of metal complexes covalently conjugated to polymer dots for cancer treatment by PDT and imaging probes [222,223,225].

**Table 1 pharmaceutics-14-01506-t001:** Non-platinum metal-based therapeutic agents (metallodrugs and PSs) are physically loaded into or covalently conjugated to polymer-based nanosized devices.

Structures of Metal-Based Drug/ Metallopolymer Prodrug *	Metallic Center	Polymer/Copolymer Blocks	Method of Incorporation	Type of Nanocarrier	Average Diameter Size (nm)	Responsive Delivery/ Specific Targeting Moieties/Co-Delivery	Potential Therapeutic Approaches	Ref.
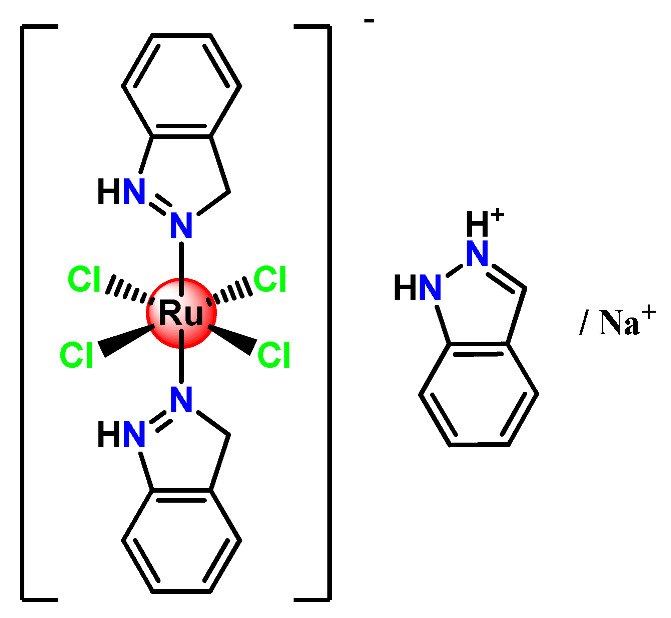 (KP1019)—**Ru1**	Ru^3+^	PLA	Physical encap.	Micelles	163	-	CTX	[57]
		PEG	Physical encap.	Micelles	<100	-	CTX	[50]
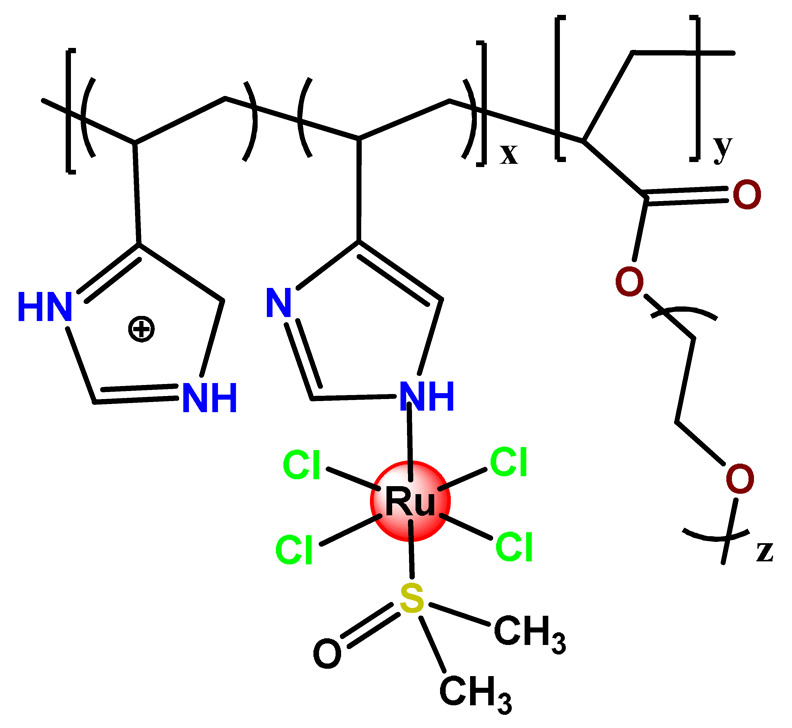 (NAMI-A prodrug)—**Ru2**	Ru^2+^	PEG-methyl ether acrylate	Covalent conjug.	Micelles	90	-	CTX	[108]
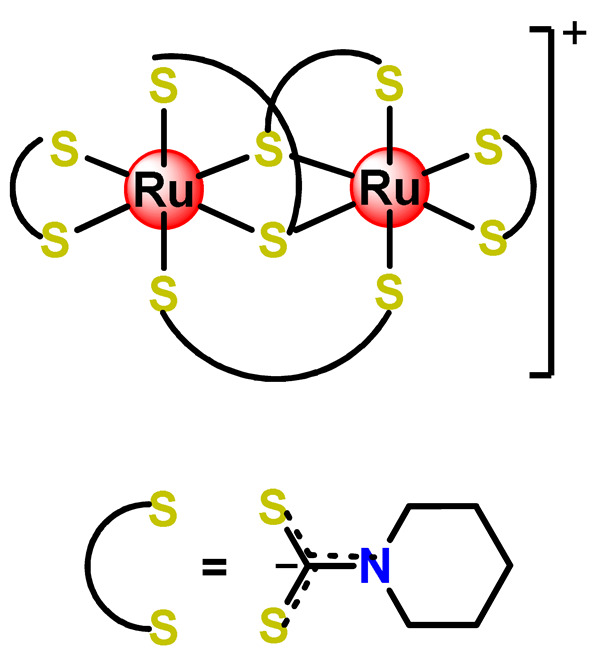 **Ru3**	Ru^3+^	PF127	Physical encap.	Micelles	24	PF127 was functionalized with carbohydrate (fluorescein, glucosamine, and β-D-glucopyranoside) for target delivery.	CTX	[97]
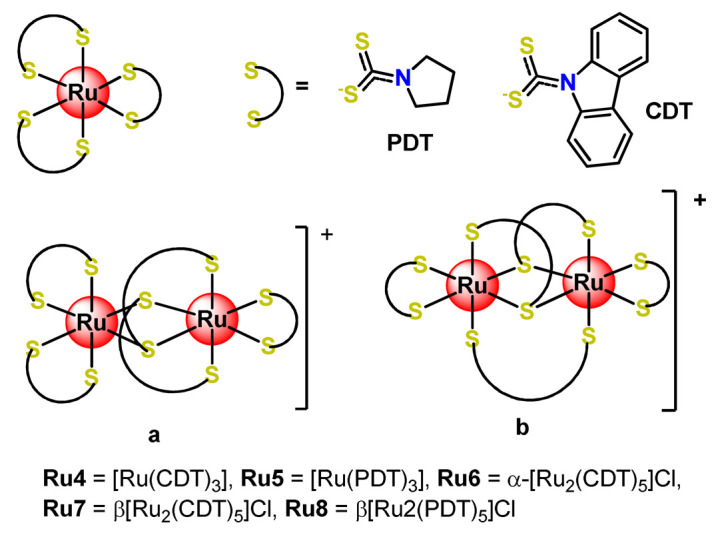 **Ru4–Ru8**	Ru^3+^	PF127	Physical encap.	Micelles	-	-	CTX	[72]
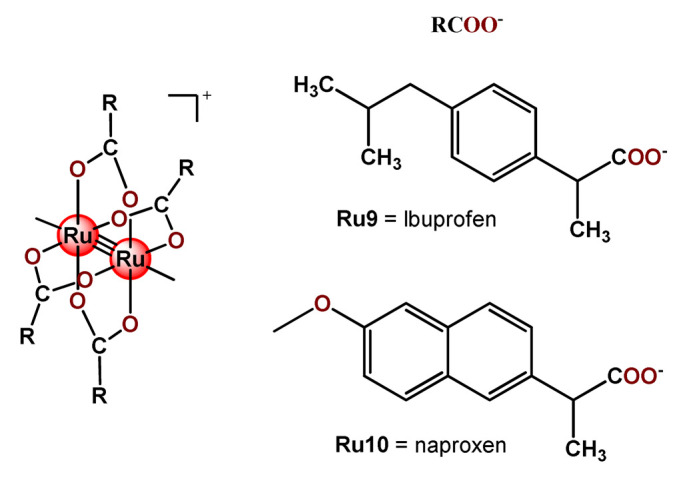 **Ru9** and **Ru10**	Ru^3+^/ Ru^2+^	stearate-PEG_40/100_, ethyl arachidate, and myristic acid	Physical encap.	NPs	120	-	CTX	[109]
		poly(methacrylic acid)-PS80-starch	Physical encap.	NPs	130	-	CTX	[83]
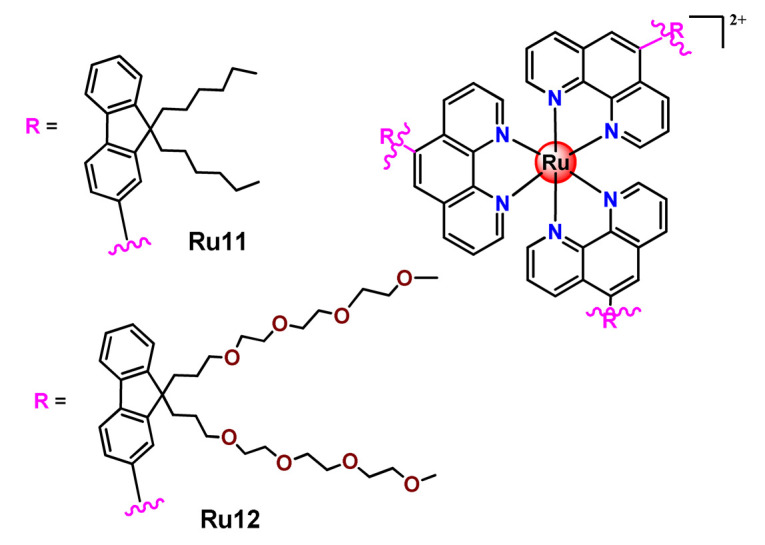 **Ru11** and **Ru12**	Ru^2+^	PLGA	Physical encap.	Nanosphere	100	-	CTX	[61]
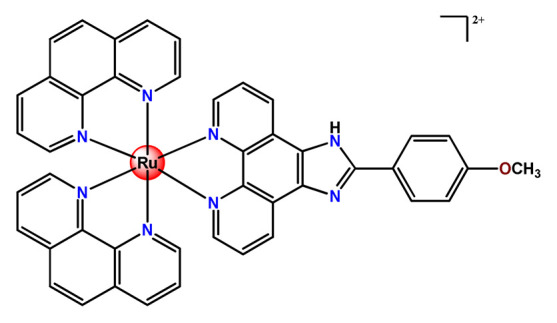 **Ru13**	Ru^2+^	PEI-*m*PEG_5000_/PLGA (PEI = polyethyleneimine)	Physical encap.	Nanosphere	153	Biotin was conjugated to polymer chain for target delivery	CTX	[87]
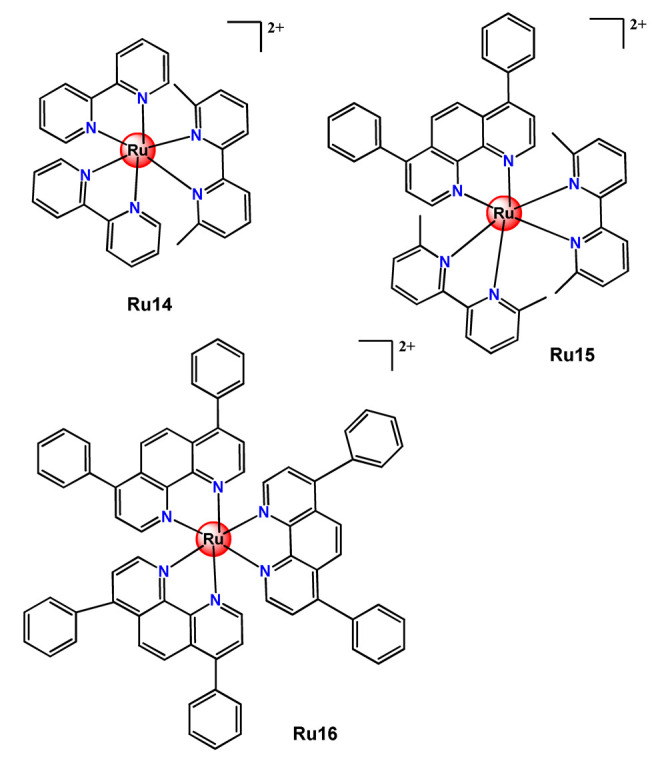 **Ru14–Ru16**	Ru^2+^	PEG- poly(aspartate)	Physical encap.	NPs	19	-	PDT	[51]
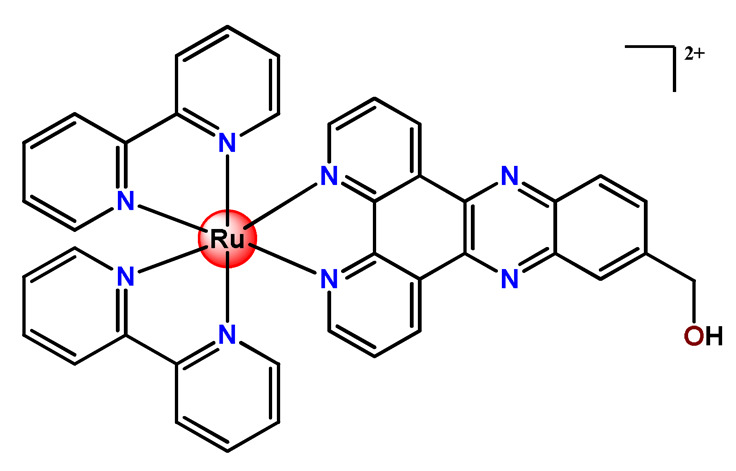 **Ru17**	Ru^2+^	poly(*N*,*N*-dimethylaminoethylmetha-cry-late-*co*-poly (methyl methacrylate)	Physical encap.	Micelles	8	-	PDT	[110]
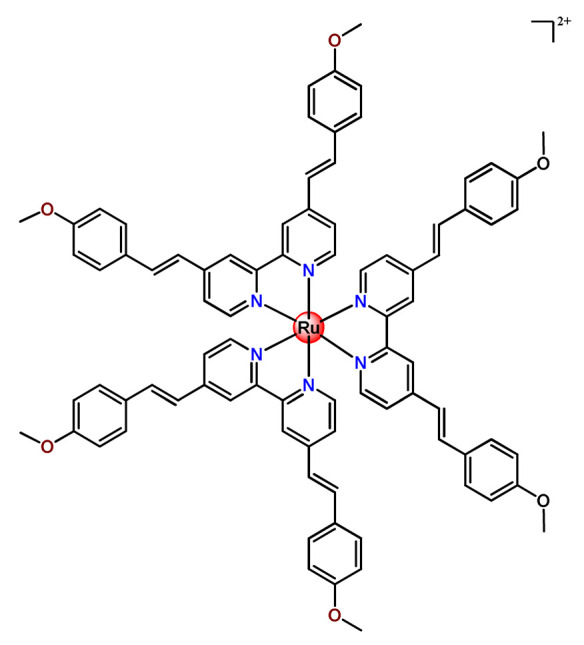 **Ru18**	Ru^2+^	PF127	Physical encap.	Nanosphere	31–162	-	PDT	[73]
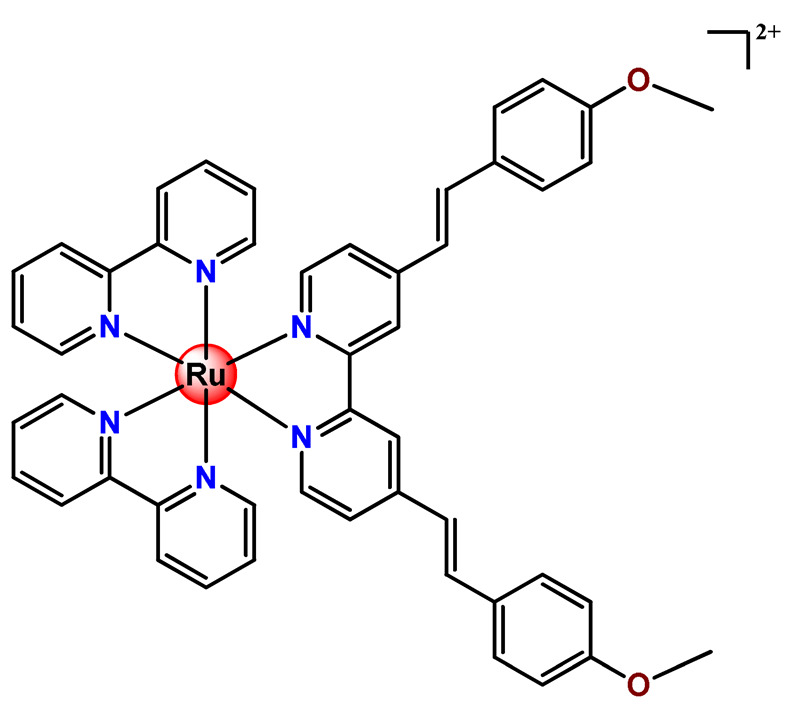 **Ru19**	Ru^2+^	DSPE-PEG_2000_	Physical encap.	NPs	100	Biotin was conjugated to the polymer chain for target delivery	PDT	[111]
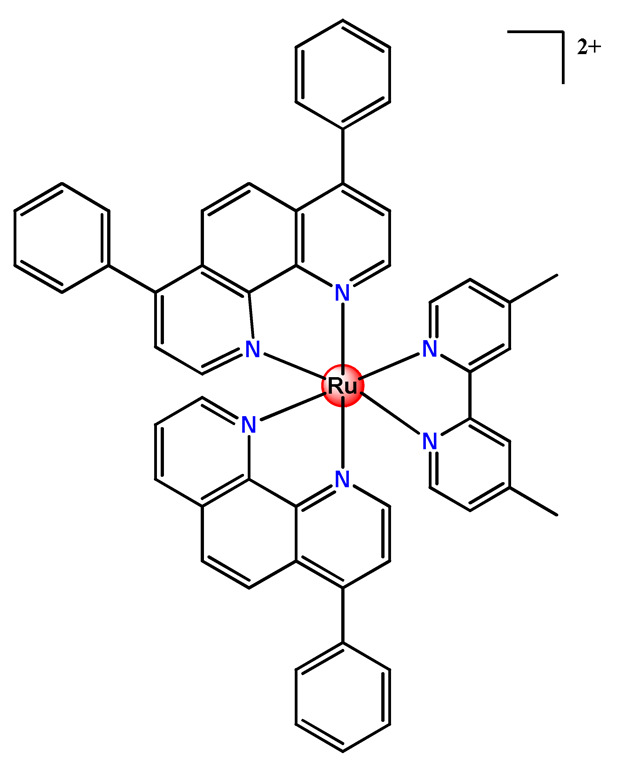 **Ru20**	Ru^2+^	DSPE-PEG_2000_	Physical encap.	Micelles	122	FA was conjugated to the polymer chain for target delivery	PDT	[98]
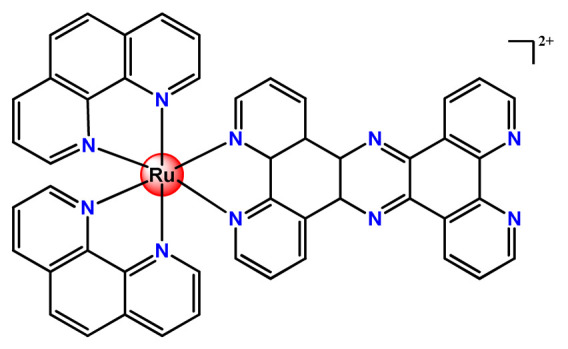 **Ru21**	Ru^2+^	PLGA	Physical encap.	NPs	130–140	Co-delivery of Ru-based radiosensitizer alongside radionuclide (^111^In). The functionalized PLGA surface incorporates the metal ion chelator DTPA for radiolabelling and the targeting ligand for EGF receptor (EGFR)	Ionizing radiation and radiosensitizing CTX.	[46]
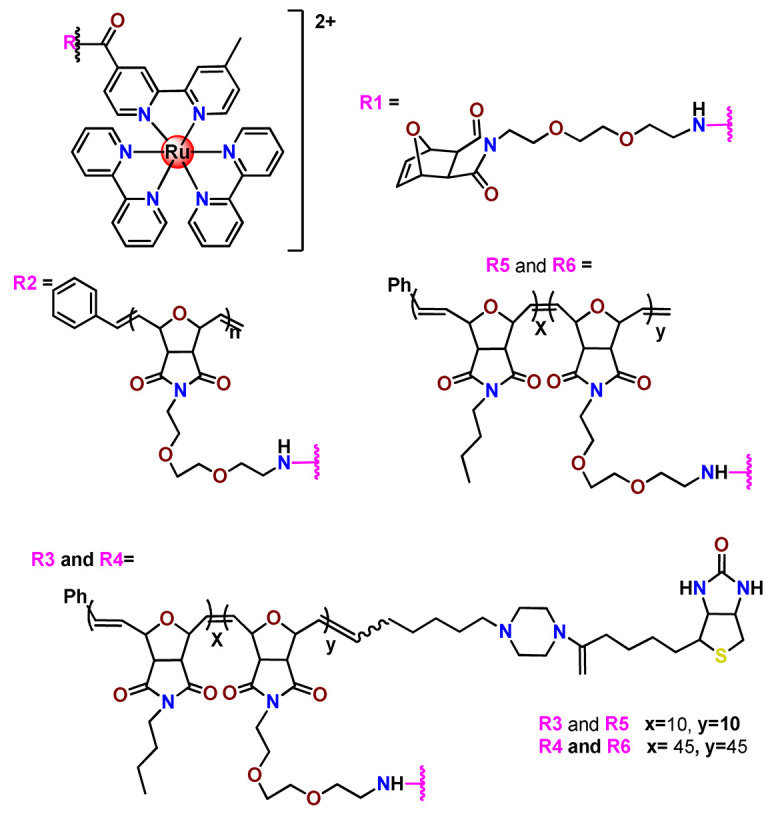 **Ru22–Ru27**	Ru^2+^	bipyridine-containing block copolymers	Covalent conjug.	Micelles	41 (**Ru22** and **Ru23**) 100–1000 with the protein streptavinmmndin.	Biotin was conjugated to polymer chain. Addition of streptavidin to the Ru-loaded micelles induces their cross-linking into larger networks, through biotin-streptavidin binding	Luminescent detector of biomolecules	[112,113]
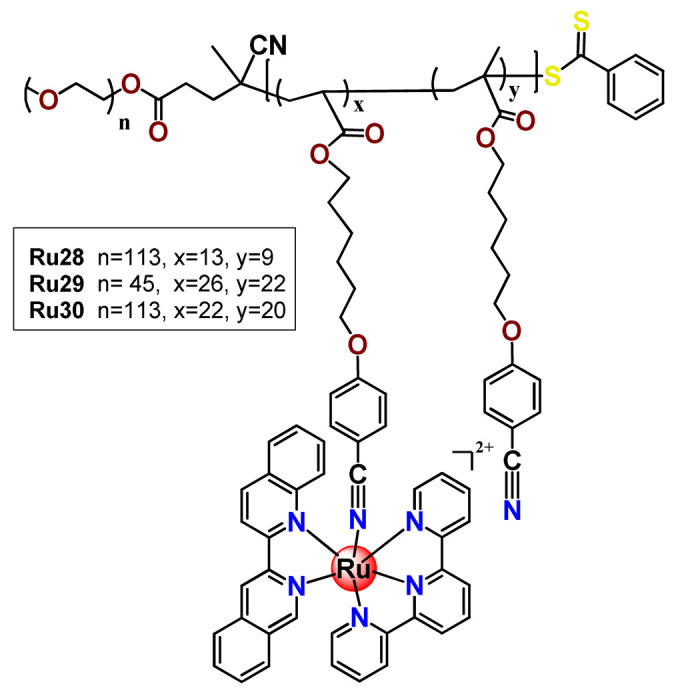 **Ru28–Ru30**	Ru^2+^	PEG-*b*-poly(6-(4-cyano-phenoxy) hexyl methacrylate)	Covalent conjug.	Micelles	**Ru28** = 12; **Ru29** = 48; **Ru30** = 120	Photo-controlled delivery of Ru^2+^ complexes	PDT	[79]
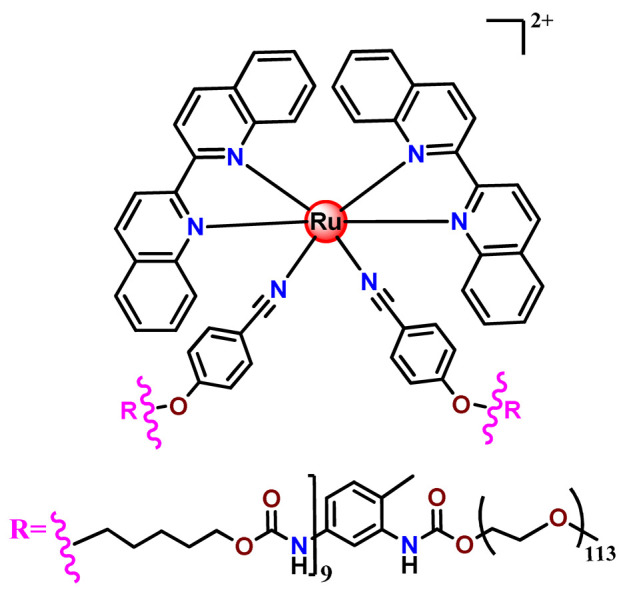 **Ru31**	Ru^2+^	2,4-diisocyanato-1-methy-lbenzene-PEG monomethyl ether	Covalent conjug.	Micelles	180	Photo-controlled delivery of Ru^2+^ complex	PDT + PACT	[114,115]
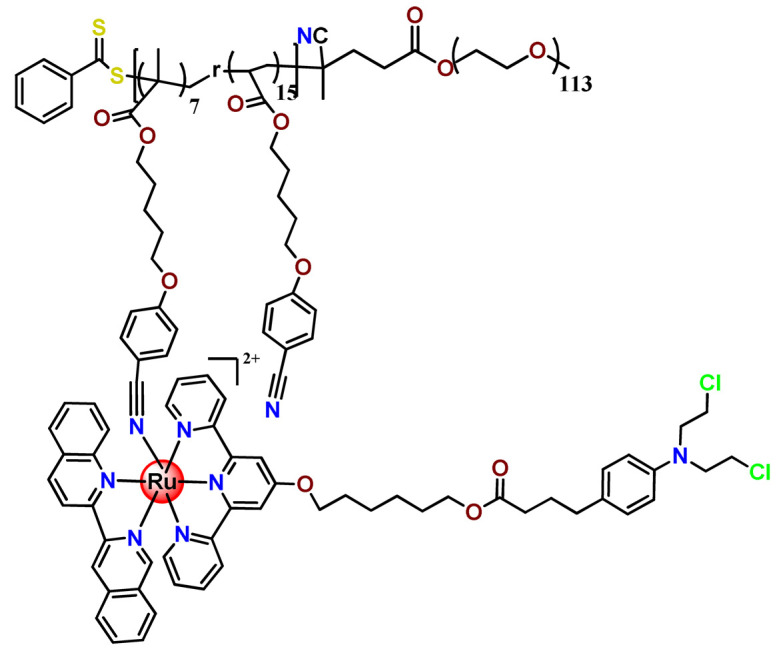 **Ru32**	Ru^2+^	PEG-*b*-poly (6-(4-cyano-phenoxy) hexyl methacrylate)	Covalent conjug.	Micelles	15	Photo-controlled delivery of Ru^2+^ complex. Polymer chain was covalently attached to anticancer drug chlorambucil	PACT	[56]
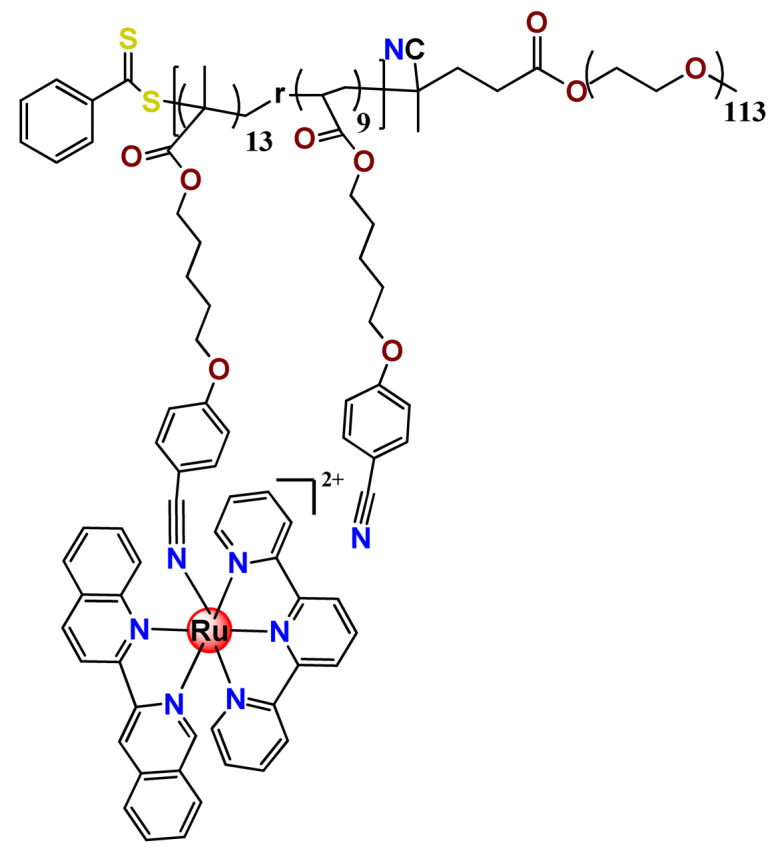 **Ru33**	Ru^2+^	PEG-*b*-poly (6-(4-cyano-phenoxy) hexyl methacrylate)	Covalent conjug.	Micelles	22	Photo-controlled delivery of Ru^2+^ complex	PDT	[115]
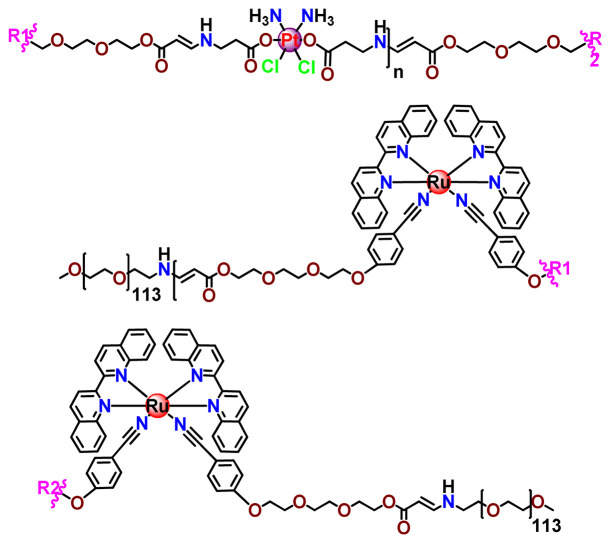 **Ru34**	Pt^4+^/Ru^2+^	PEG	Covalent conjug.	NPs	90	Dual-Responsive Pt^4+^/Ru^2+^. Reduction-responsive Pt^4+^ and red-light-responsive Ru^2+^ moieties.	CTX + PDT	[101]
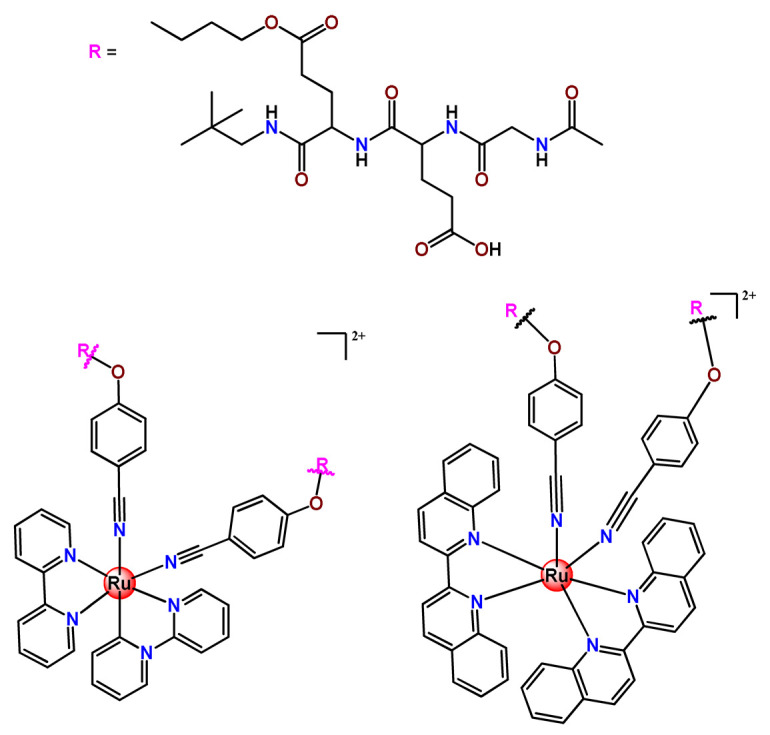 **Ru35** and **Ru36**	Ru^2+^	polysarcosine-*b*-poly (glutamic acid)	Covalent conjug.	Micelles	91	Photo-controlled delivery of Ru^2+^ complexes	PDT + PACT	[116]
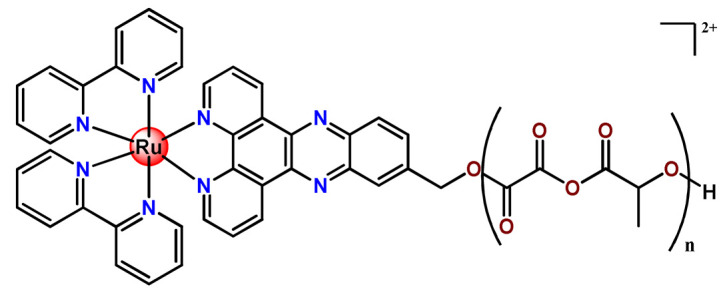 **Ru37**	Ru^2+^	PLA	Covalent conjug.	NPs	120–310	Photo-controlled delivery of Ru^2+^ complex	PDT + PDIM	[63]
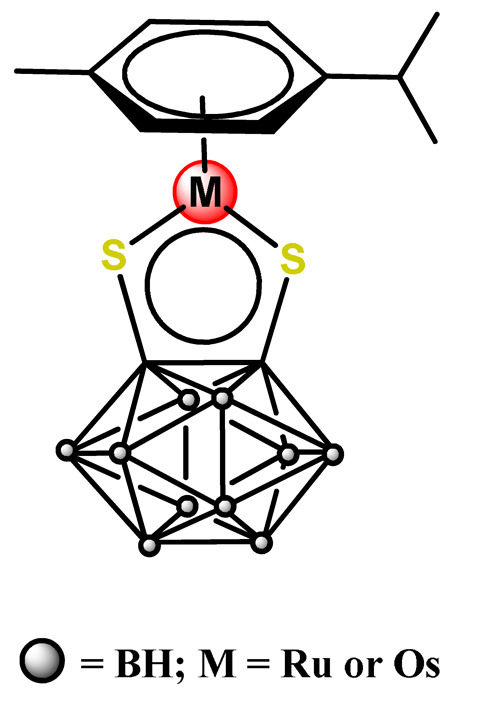 **Ru38** and **Os1**	Ru^2+^ or Os^2+^	PP123	Physical encap.	Micelles	15	-	CTX	[117]
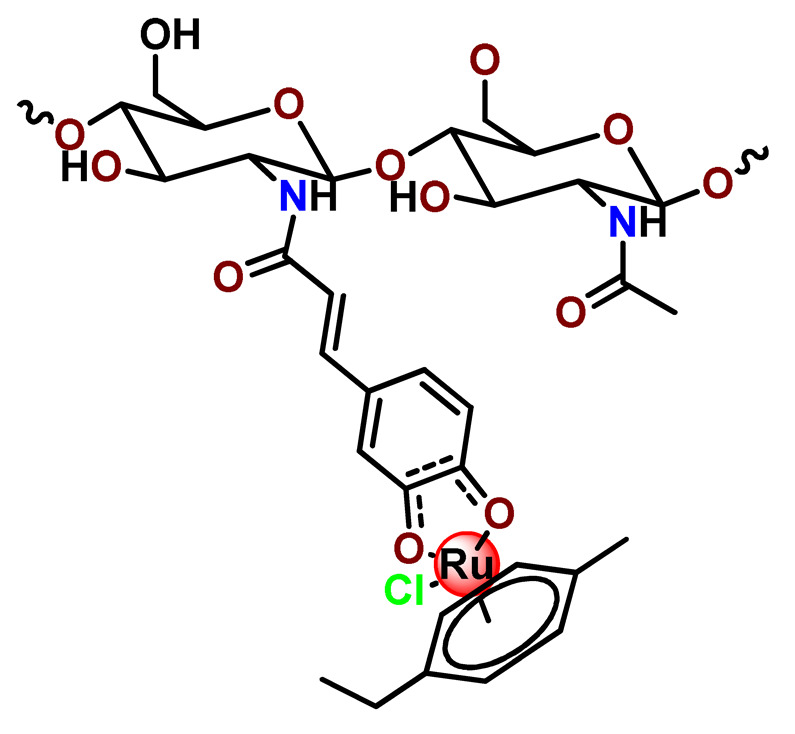 **Ru39**	Ru^3+^	caffeic acid-modified chitosan	Covalent conjug.	Micelles	30–120	-	CTX	[80]
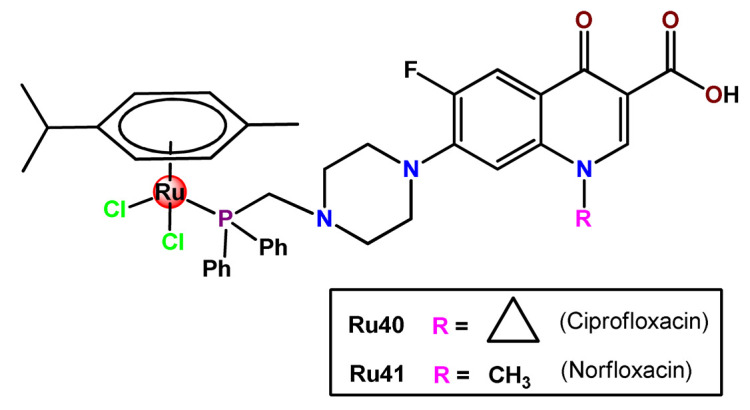 **Ru40** and **Ru41**	Ru^2+^	PP123	Physical encap.	Micelles	26	-	CTX	[118]
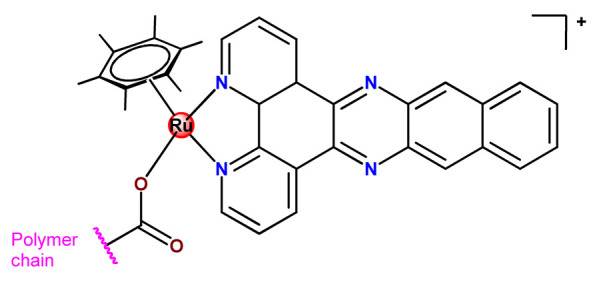 **Ru42**	Ru^2+^	PEG-*b*-poly(glutamic acid)	Covalent conjug.	Micelles	60	-	CTX	[78]
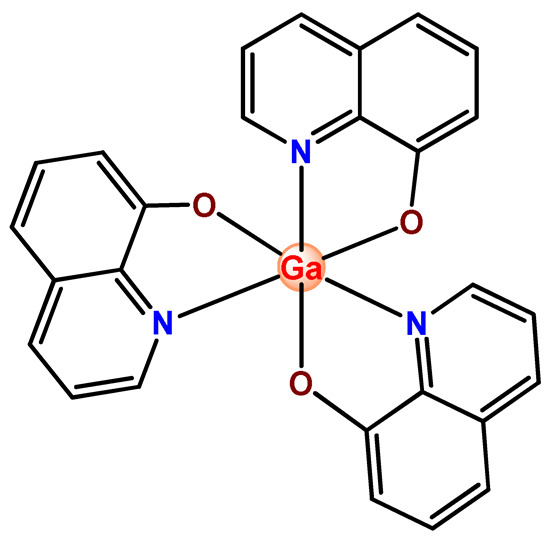 (KP46)—**Ga1**	Ga ^3+^	PEG-*b*-poly(glutamic acid)	Physical encap.	Micelles	17	-	CTX	[52]
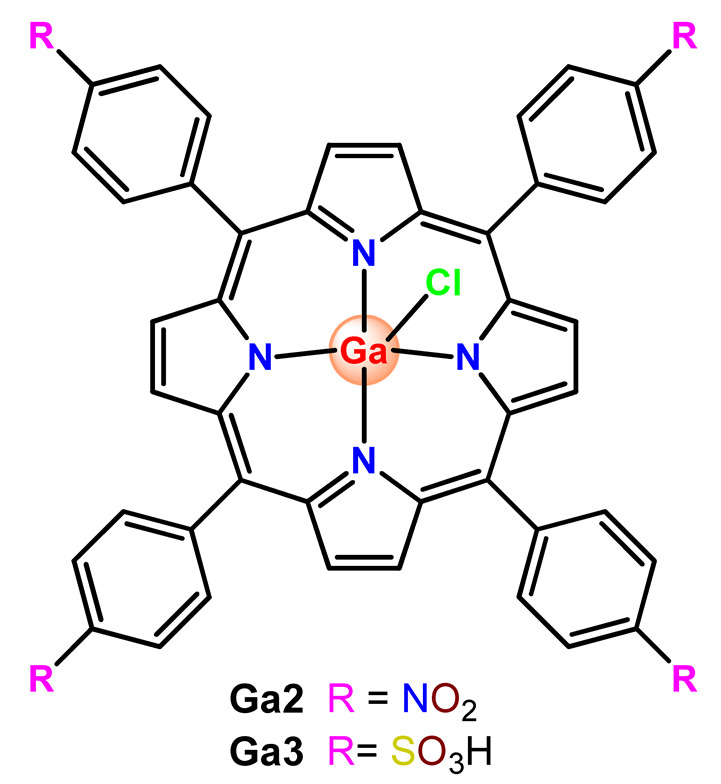 **Ga2** and **Ga3**	Ga^3+^	PF127	Physical encap.	Micelles	31–40	-	PDT	[74]
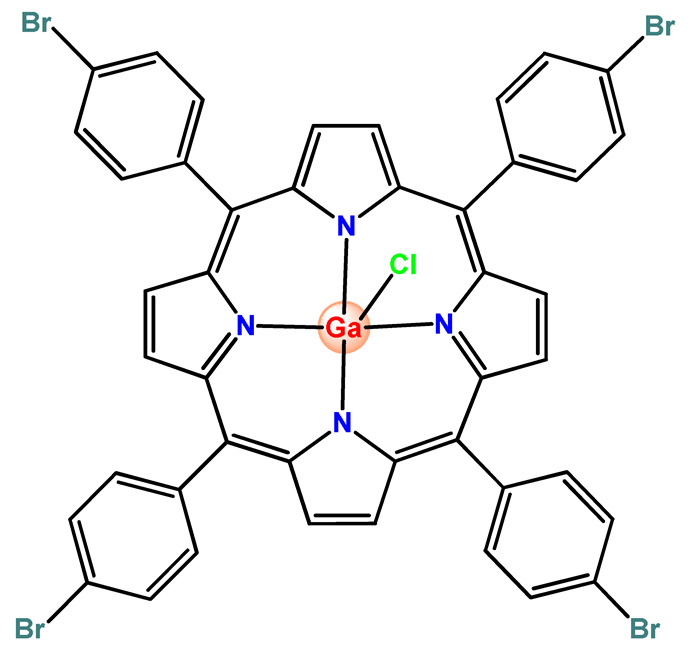 **Ga4**	Ga^3+^	PF127	Physical encap.	Micelles	33	Polymer chain was functionalized with FA for target delivery	PDT	[71]
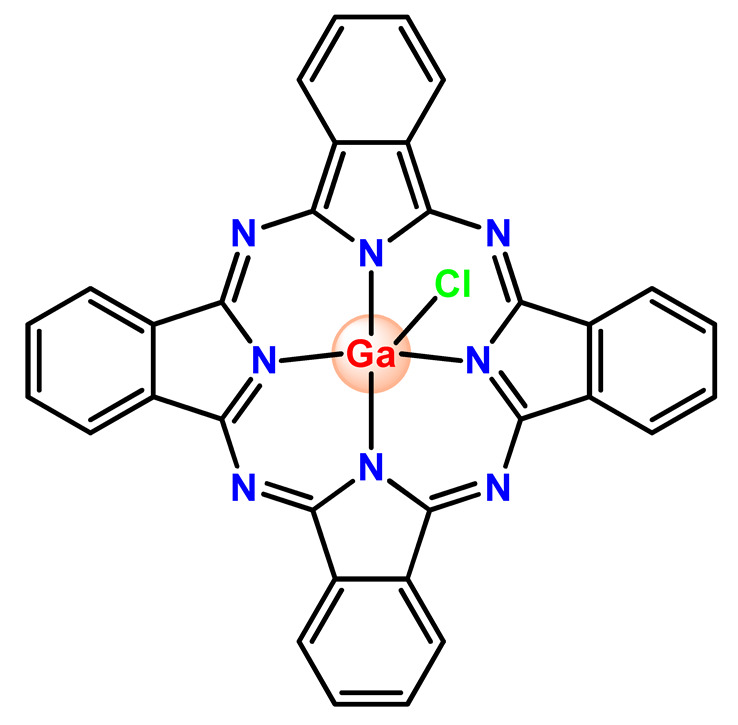 **Ga5**	Ga^3+^	PLGA–PEG	Physical encap.	Nanosphere	86–204	-	PDT	[64]
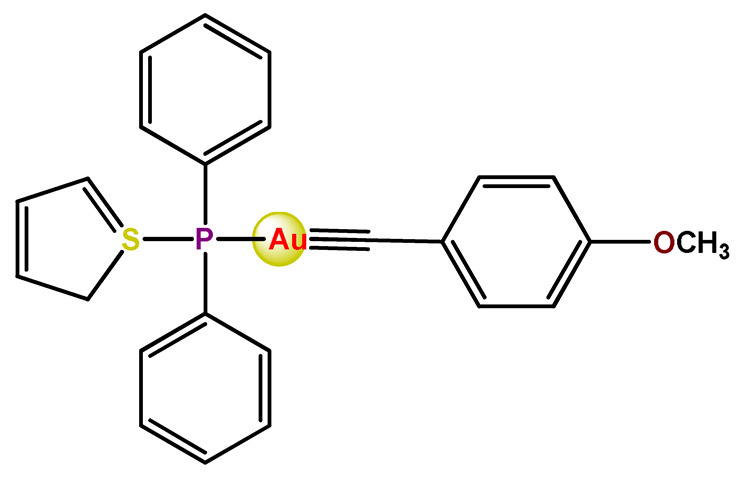 **Au1**	Au^+^	PEG-poly(β-amino ester)	Physical encap.	Micelles	43–50	pH-responsive drug delivery	CTX	[102]
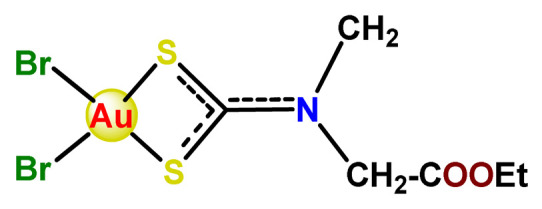 **Au2**	Au^3+^	DSPE-PEG_2000_	Physical encap.	Micelles	17–21	-	CTX	[119]
		PF127	Physical encap.	Micelles	26–32	The hydrophilic shell of micelles was labeled with bioactive CCK8 peptide for target-selective delivery	CTX	[120]
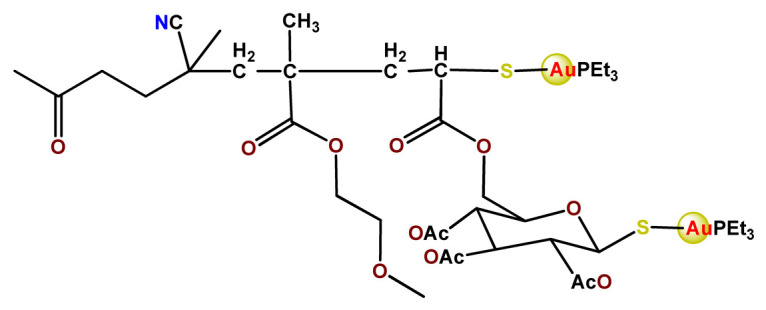 **Au3**	Au^+^	Oligo-(ethylene glycol) methyl ether methacrylate	Covalent conjug.	Micelles	160	-	CTX	[38]
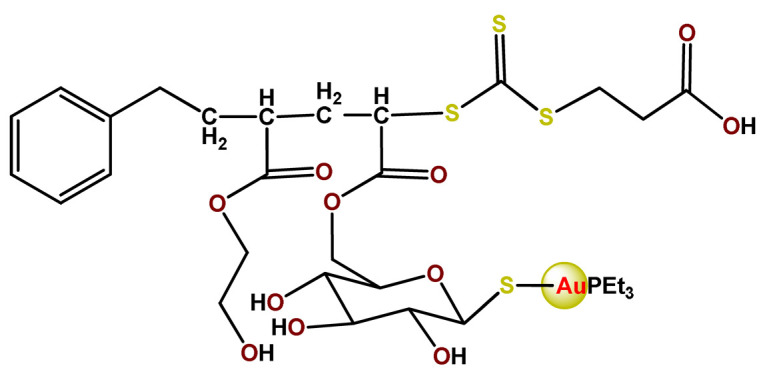 **Au4**	Au^+^	Poly(2-hydroxy- ethyl acrylate)	Covalent conjug.	Micelles	75	-	CTX	[121]
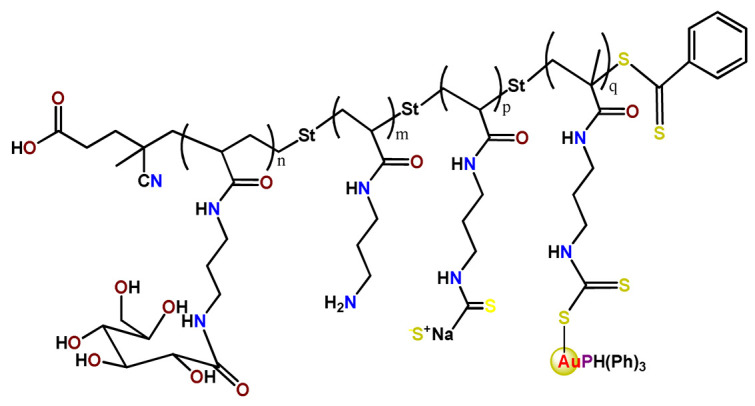 **Au5**	Au^+^	Glycopolymer based on 3-Gluconamido-propyl metha-crylamid and 3-aminopro-pyl methacry-lamid (10 and 30 KDa)	Covalent conjug.	-	100–250	-	CTX	[122]
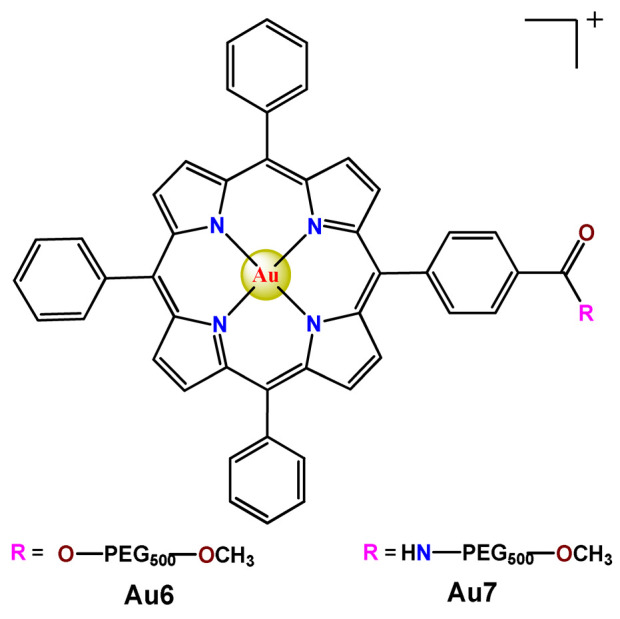 **Au6** and **Au7**	Au^3+^	PEG	Covalent conjug.	Micelles	121	Co-delivery of DOX	CTX	[47]
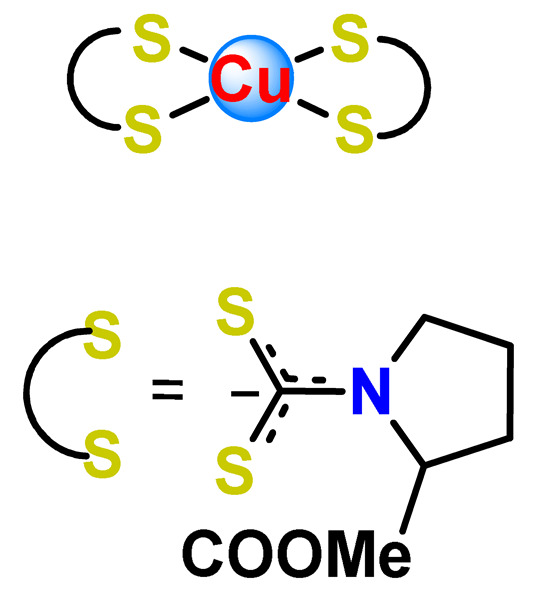 **Cu1**	Cu^2+^	PF127	Physical encap.	Micelles	23–26	PF127 was functionalized with carbohydrates (fluorescein, glucosamine, or β-D-glucopyranoside) for target delivery.	CTX	[97]
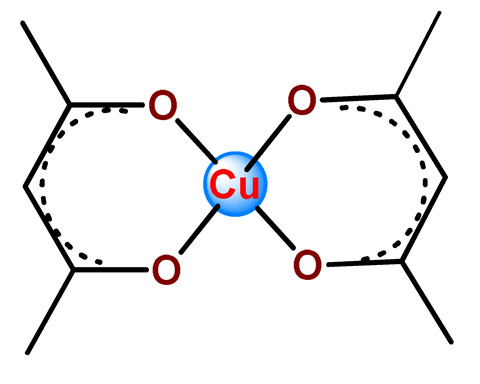 **Cu2**	Cu^2+^	Chitosan	Physical encap.	NPs	240	FA was conjugated to chitosan for target delivery	CTX	[123]
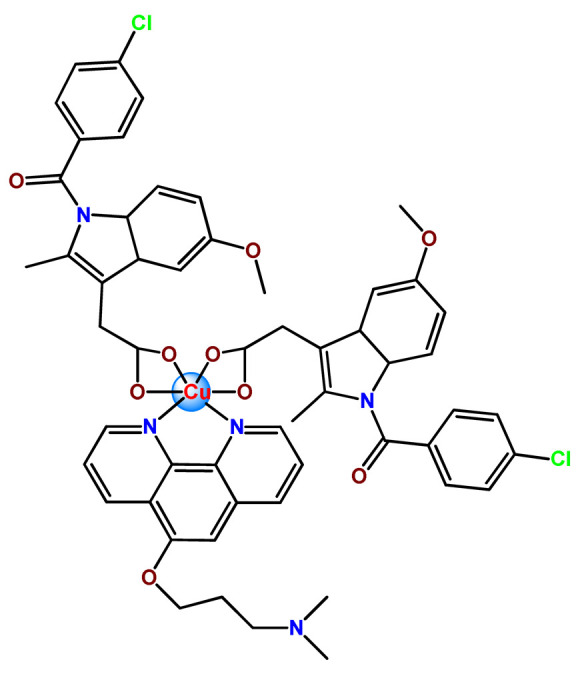 **Cu3**	Cu^2+^	PEG-PLGA	Physical encap.	Nanosphere	145	-	CTX	[65]
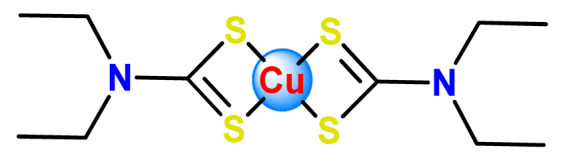 **Cu4**	Cu^+^	PLA–PEG	Physical encap.	NPs	50–150	-	CTX	[66]
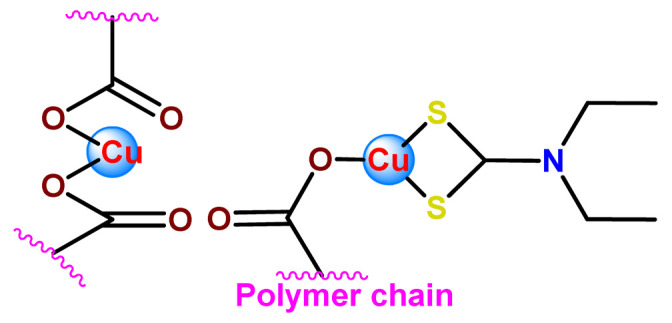 **Cu5**	Cu^+^/Cu^2+^	PEG-*b*-poly(ester-carbonate)	Covalent conjug.	NPs	120−135	-	CTX + CDT	[91]
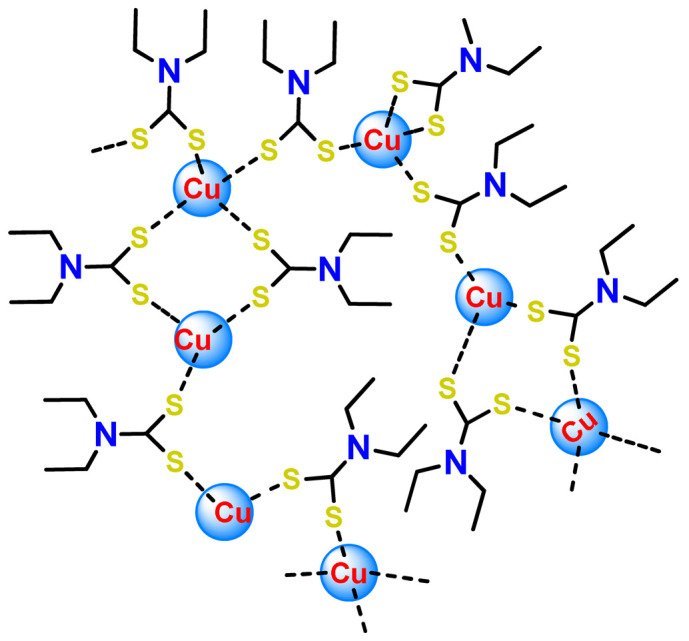 **Cu6**	Cu^2+^	Hyaluronic acid	Physical encap.	NPs	125	pH/GSH dual-responsive drug delivery	CTX	[103]
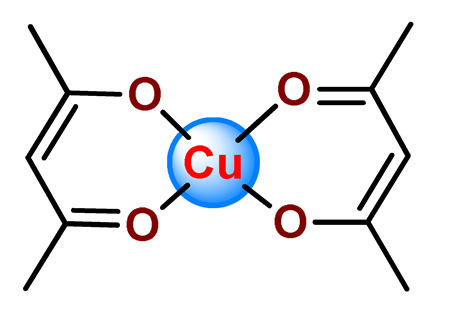 **Cu7**	Cu^2+^	Chitosan	Physical encap.	NPs	50–65	FA or her-2 (specific peptide overexpressed in many cancer cells) was attached to stearic acid-modified chitosan	CTX	[99]
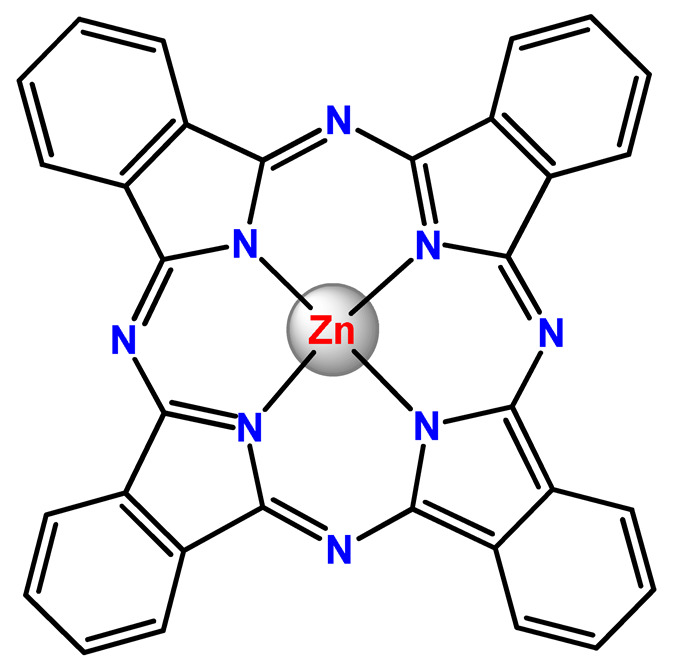 **Zn1**	Zn^2+^	PEG_5000_-DSPE	Physical encap.	Micelles	160	-	PDT	[124]
		PLGA	Physical encap.	NPs	285	-	PDT	[62]
		metoxiPEG-*b*-PLA	Physical encap.	Micelles	32–35	Polymer chain was functionalized with FA for target delivery	PDT	[125]
		PEG-*b*-PLLA	Physical encap.	Micelles	<150	Polymer chain was functionalized with FA for target delivery	PDT	[126]
		Heparin, polyethyleneimine and L-cysteine	Physical encap.	Spherical NPs	<200 nm	-	PDT	[127]
		Heparin, DSPE, and l-histidine	Physical encap.	Micelles	111	pH-responsive drug delivery	PDT	[42]
		poly(*N*-acrylamide)-*b*-PEG-monomethyl ether acrylate)	Physical encap.	Micelles	167–230	-	PDT	[128]
		poly(ε-capro-lactone)-PEO	Physical encap.	Micelles	61–97	Co-delivery of DTX.	PDT + CTX	[48]
		3-caprolactone, 1,4,8-trioxa [4.6]spiro-9-undecanone and PEG	Physical encap.	NPs	108 nm	Co-delivery of DOX.	PDT + CTX	[49]
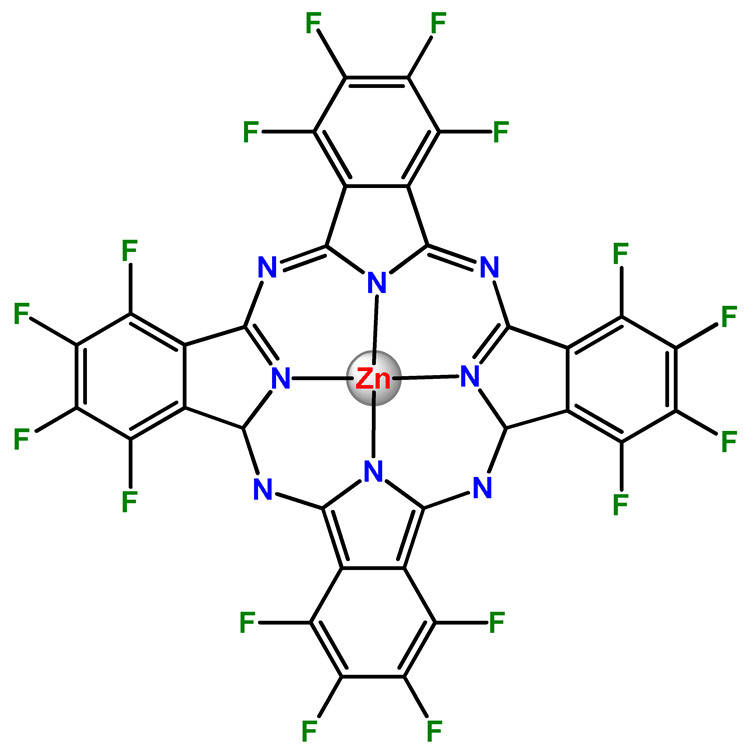 **Zn2**	Zn^2+^	PEG-PLA	Physical encap.	NPs	464–988		PDT	[18]
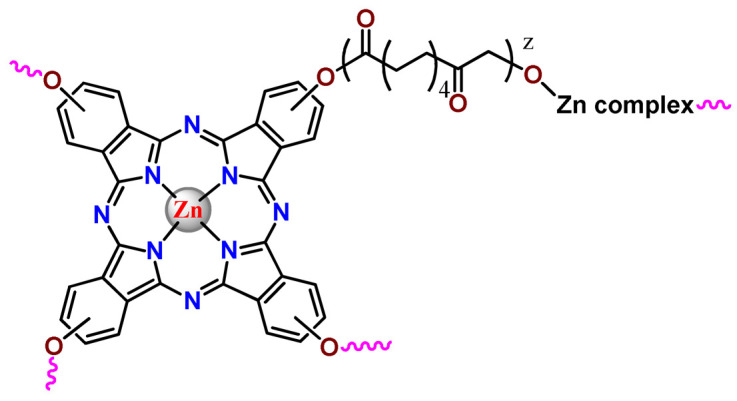 **Zn3**	Zn^2+^	poly(phthalocyanine-co-sebacic anhydride)	Covalent conjug.	NPs	166	-	PDT	[129]
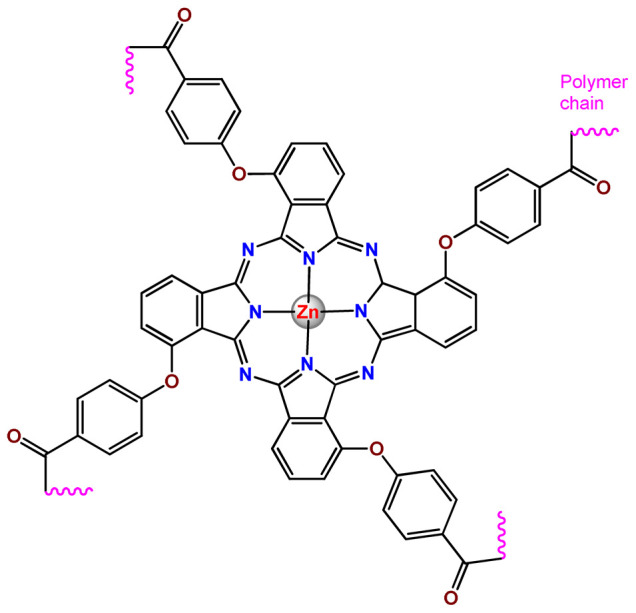 **Zn4**	Zn^2+^	tetra-(methoxy-PEGl)-poly(2-(*N*,*N*-diethyl-amino)ethyl methacrylate)-poly(ε-caprolactone) pentaerythritol	Covalent conjug.	NPs	51–342 (dependent on pH)	pH-responsive drug delivery and co-delivery of DOX	PDT + CTX	[130]
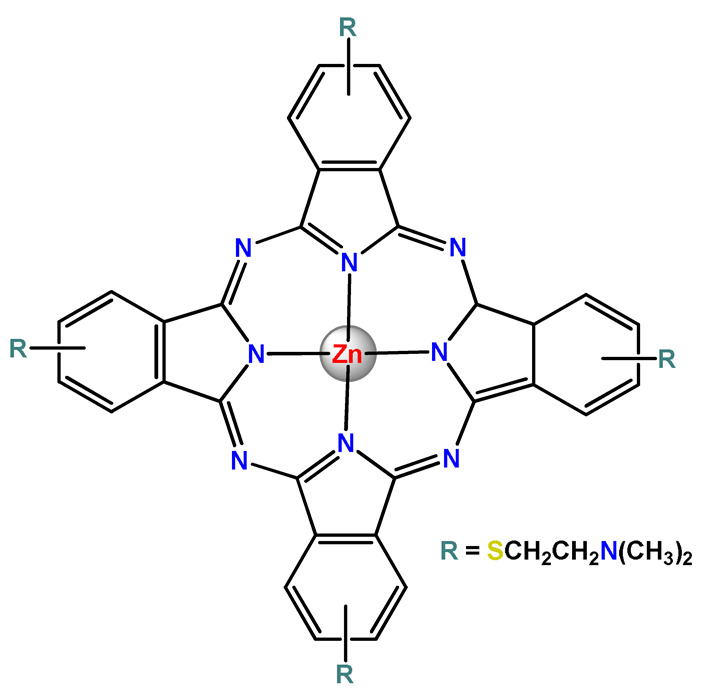 **Zn5**	Zn^2+^	PEO-poly(propylene oxide)	Physical encap.	Micelles	<405 nm	-	PDT	[131]
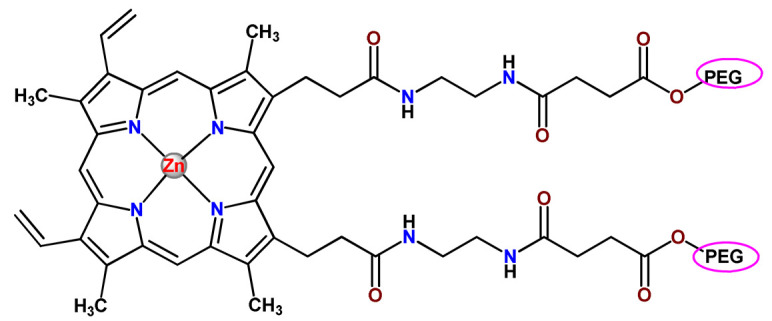 **Zn6**	Zn^2+^	PEG	Covalent conjug.	Micelles	350	-	CTX (HO-1 inhibitor)	[132,133]
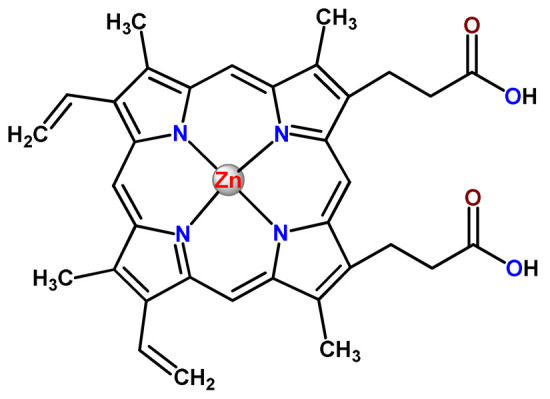 **Zn7**	Zn^2+^	SMA	Physical encap.	Micelles	176	-	CTX (HO-1 Inhibitor)	[134]
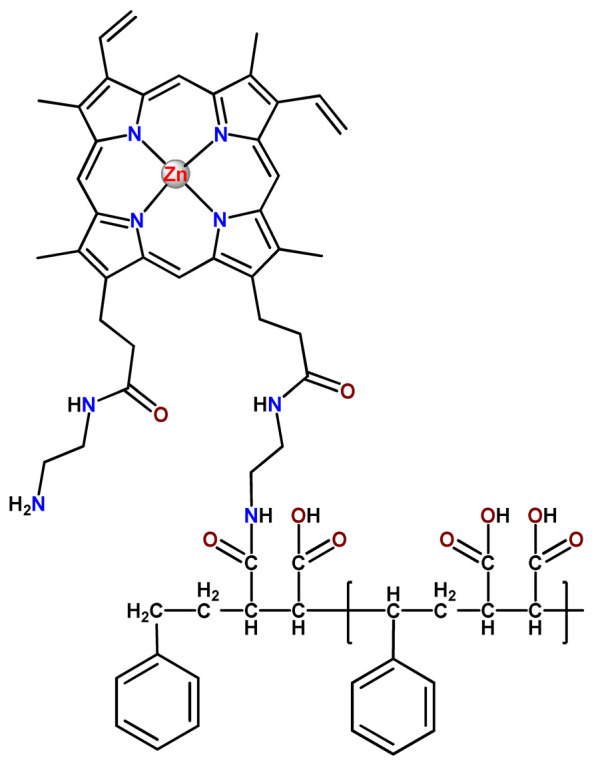 **Zn8**	Zn^2+^	SMA	Covalent conjug.	Micelles	112	-	Tumor imaging + PDT	[135]
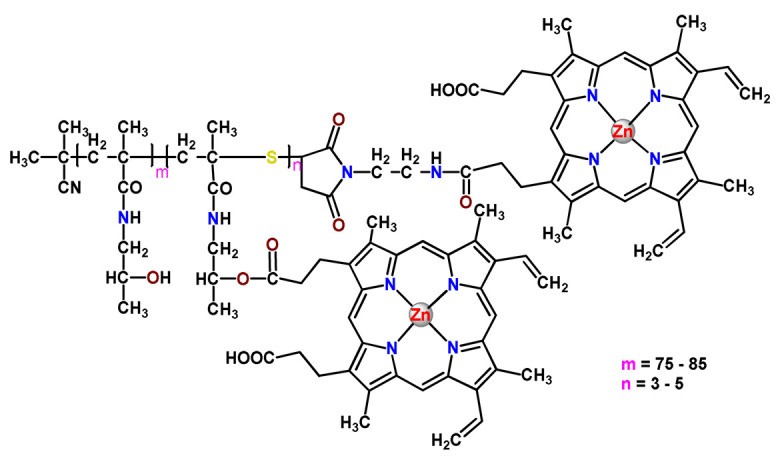 **Zn9**	Zn^2+^	HPMA	Covalent conjug.	Micelles	30–80	-	Tumor imaging + PDT	[136]
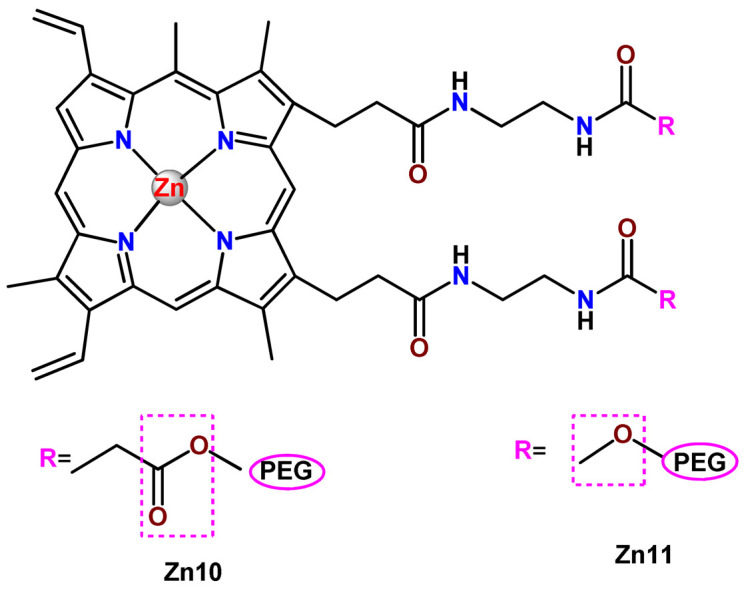 **Zn10** and **Zn11**	Zn^2+^	PEG	Covalent conjug.	Micelles	-	-	PDT	[137]
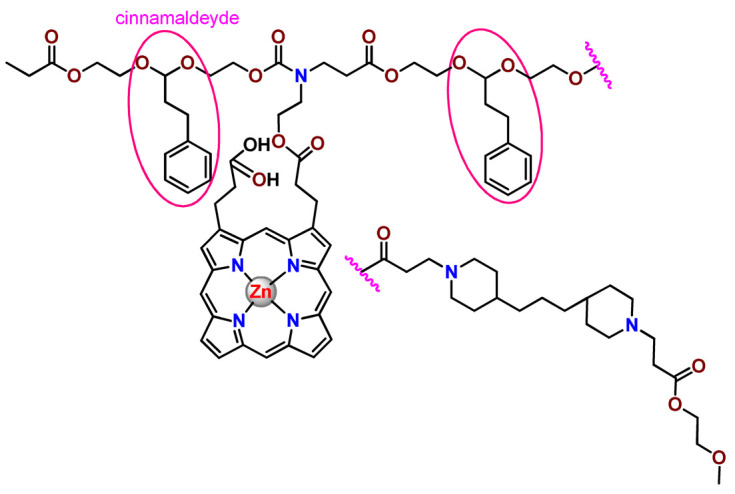 **Zn12**	Zn^2+^	PEG	Covalent conjug.	Micelles	100–150	Cinnamaldehyde was incorporated in polymer chain to enhance the ROS generation	CTX (oxidation and HO-1 inhibition)	[54]
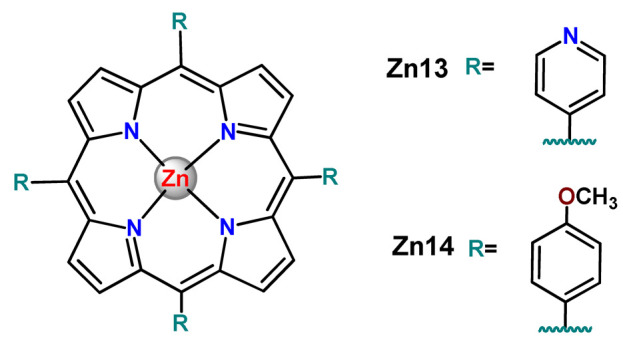 **Zn13** and **Zn14**	Zn^2+^	PP123 and PF127	Physical encap.	Micelles	-	-	PDT	[138]
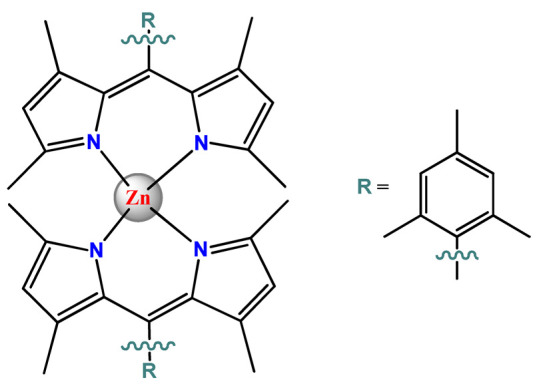 **Zn15**	Zn^2+^	DSPE-PEG_2000_	Physical encap.	Nanocapsule	100	Biotin was conjugated to polymer chain for target-selective drug delivery	PDT	[139]
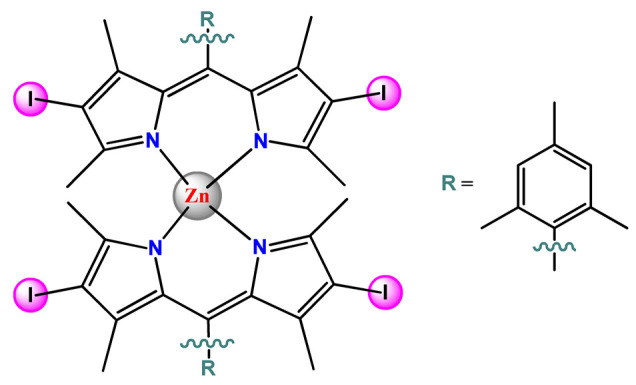 **Zn16**	Zn^2+^	DSPE-PEG_2000_-OCH3	Physical encap.	Nanocapsule	119	-	PDT	[140]
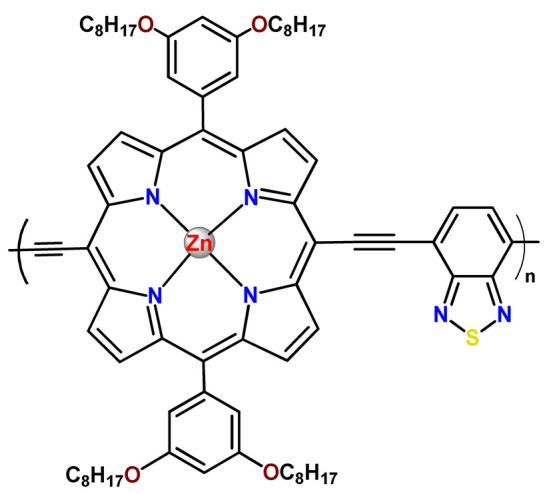 **Zn17**	Zn^2+^	DSPE-PEG_2000_- *N*-maleimide	Physical encap.	NPs	39	The NPs surface was modified with cell penetrating peptide (Tat) to enhance the cellular uptake	PTT	[141]
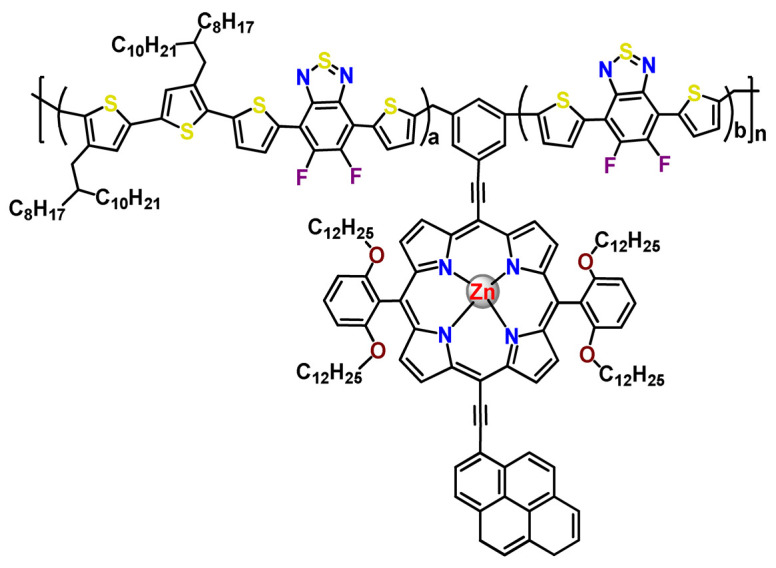 **Zn18**	Zn^2+^	HOOC-PEG-COOH, and PMHC_18_-*m*PEG	Physical encap.	NPs	81	-	Imaging and PTT	[142]
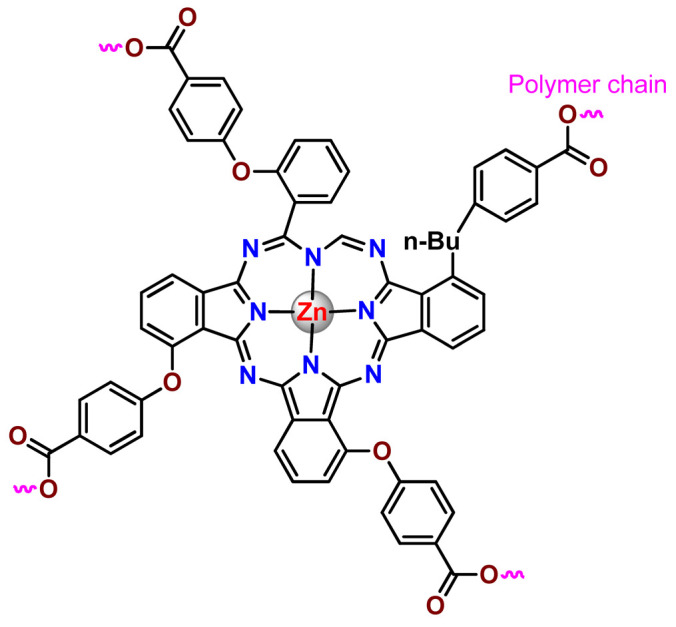 **Zn19**	Zn^2+^	PEG	Covalent conjug.	Micelles	25	-	PDT of psoriasis	[143]
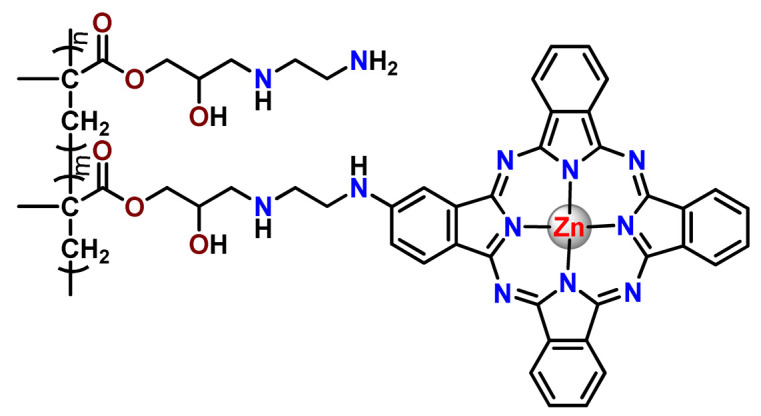 **Zn20**	Zn^2+^	poly(glycidyl methacrylate)	Covalent conjug.	NPs	160	-	PDIM	[144]
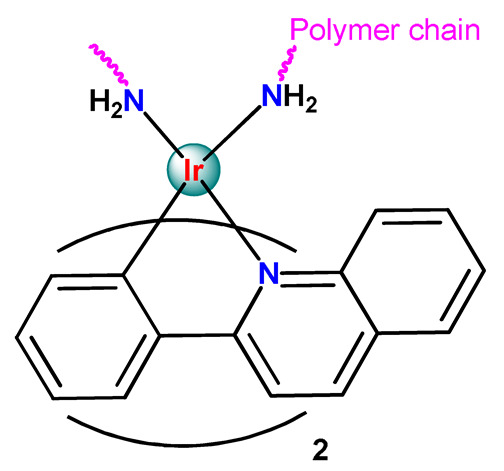 **Ir1**	Ir^3+^	oligo-L-lysine Lys30	Covalent conjug.	NPs	128	-	CTX	[145]
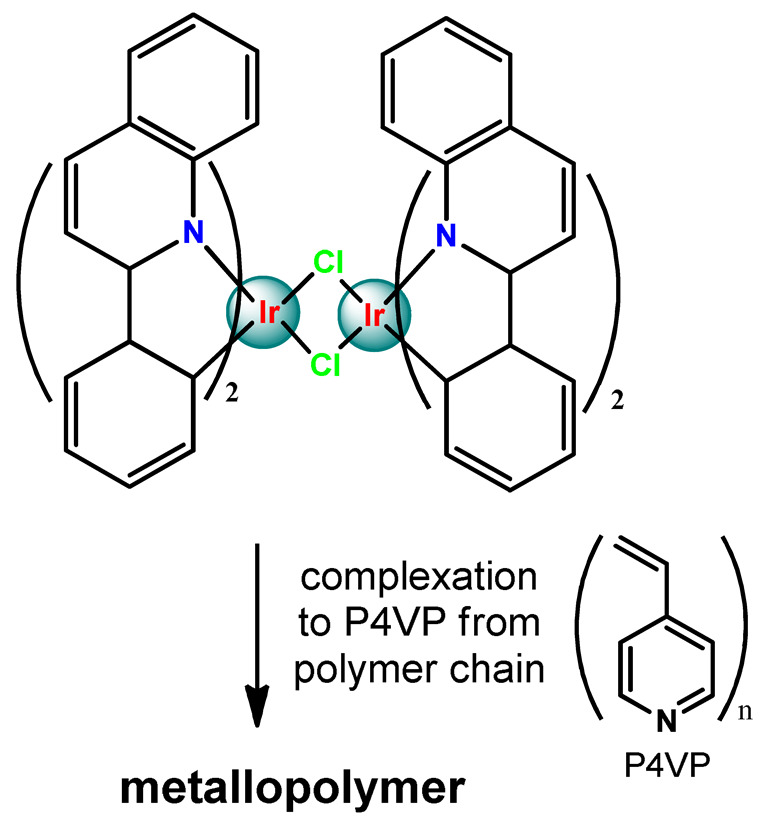 **Ir2**	Ir^3+^	PEG-*b*-P4VP (poly(4-vinylpyridine)	Covalent conjug.	Micelles	50	Polymeric chain was functionalized with FA for target delivery	CTX	[146]
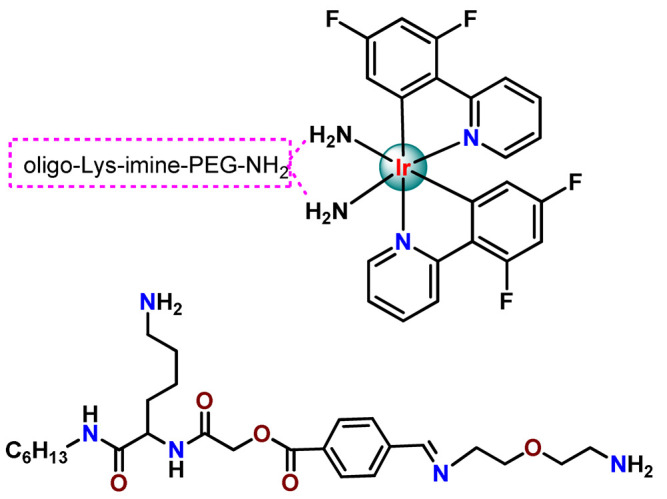 **Ir3**	Ir^3+^	PEG-benzoic imine-oligo-L-lysine	Covalent conjug.	NPs	150	pH-responsive drug delivery and the nanosystem was FA-modified for target drug release	CTX	[147]
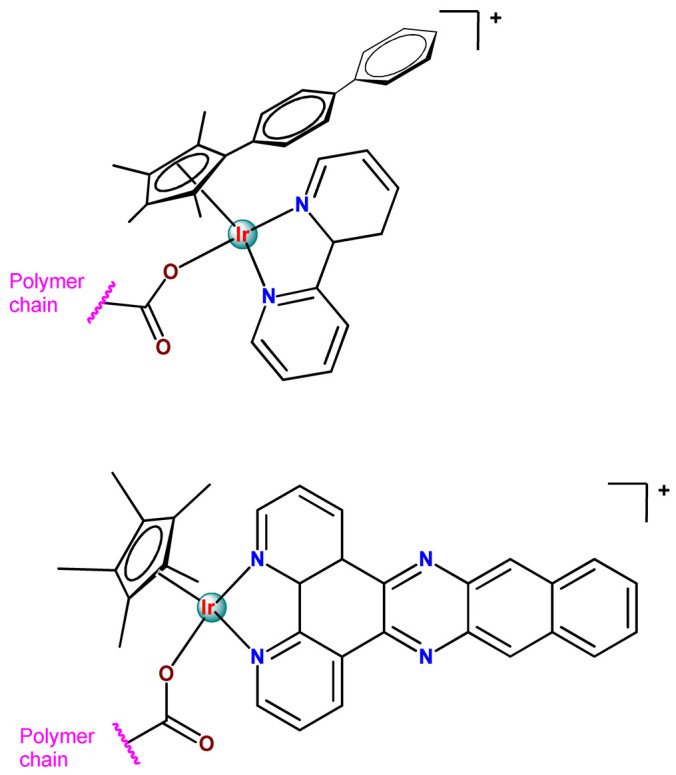 **Ir4** and **Ir5**	Ir^3+^	PEG-*b*-poly(glutamic acid)	Covalent conjug.	Micelles	58–69	-	CTX	[78]
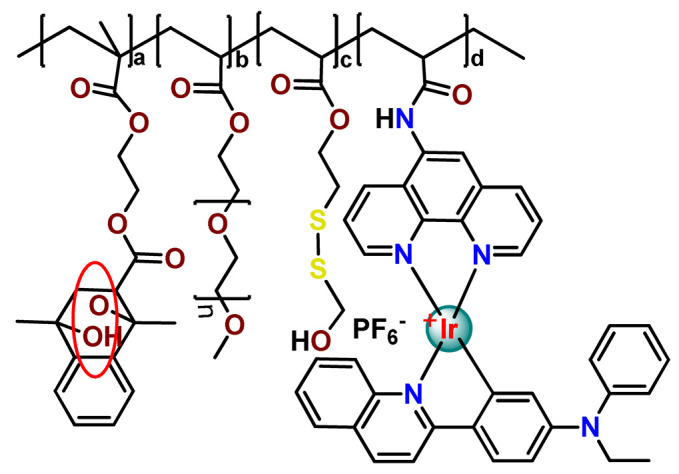 **Ir6**	Ir^3+^	1,4-dimethyl- naphthalene, PEG, and a modifiable disulfide group	Covalent conjug.	NPs	-	Photothermal-responsive delivery of extracellular ^1^O_2_	Phothermal release of ^1^O_2_	[148]
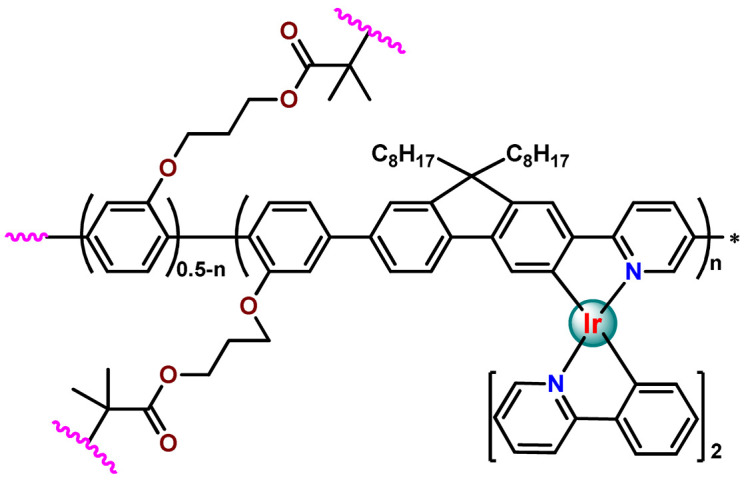 **Ir7**	Ir^3+^	POEGMA-*b*-PGal	Covalent conjug.	NPs	65	Photo-responsive drug delivery	PDT	[149]
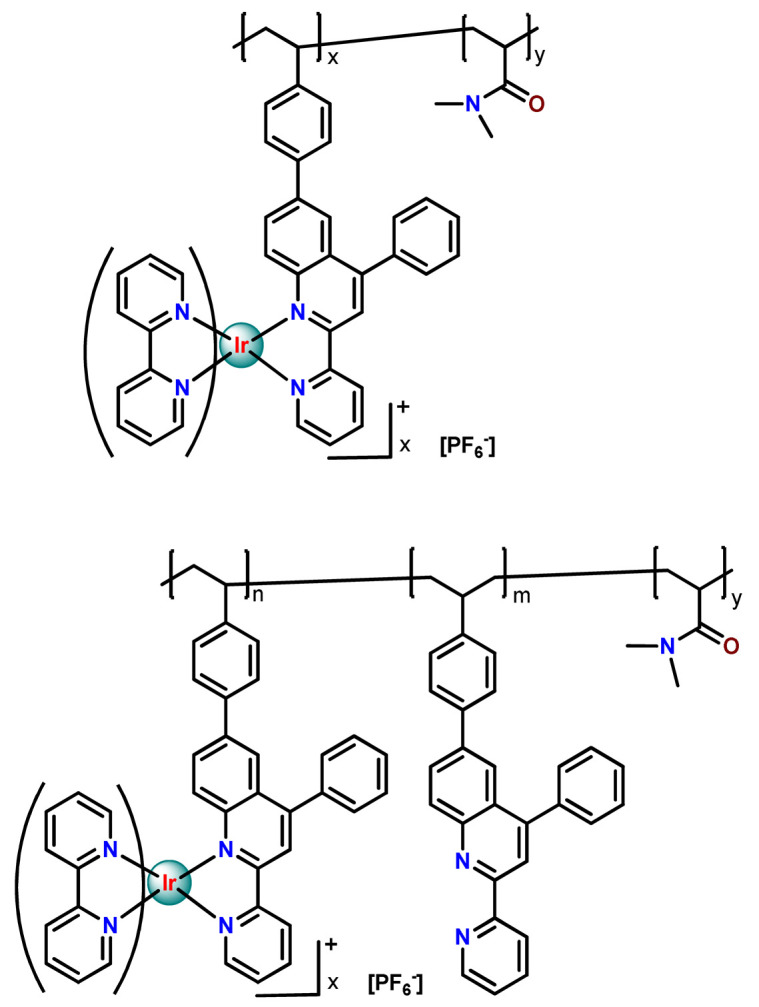 **Ir8** and **Ir9**	Ir^3+^	P [DMAM-co-VQPy] VQPy (quinoline-based unit 2-(pyridin-2-ylo)-6-styrene-4-phenylquinoline)	Covalent conjug.	NPs	-	pH responsive iridium-complex delivery.	PDT + CTX	[150]
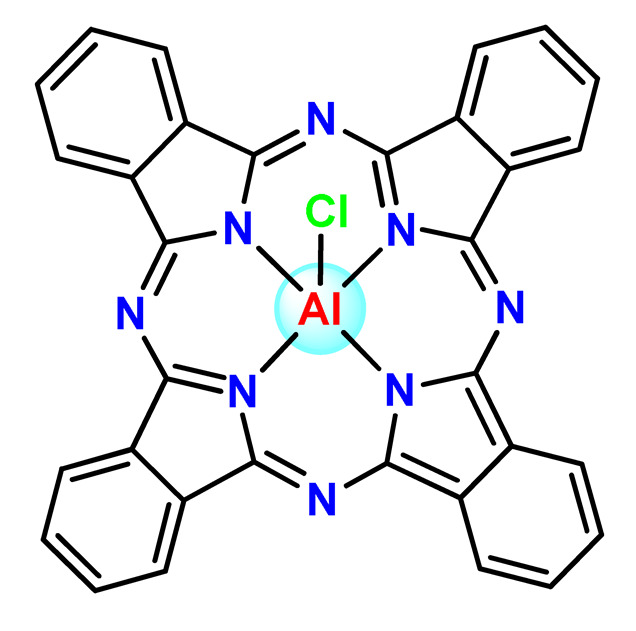 **Al1**	Al^3+^	*N*-isopropylacrylamide	Physical encap.	Micelles	35	pH-responsive drug delivery	PDT	[19]
		PLGA, PLA, and PLA–PEG	Physical encap.	NPs	139–199	-	PDT	[89]
		PF127	Physical encap.	Micelles	6	-	PDT	[84]
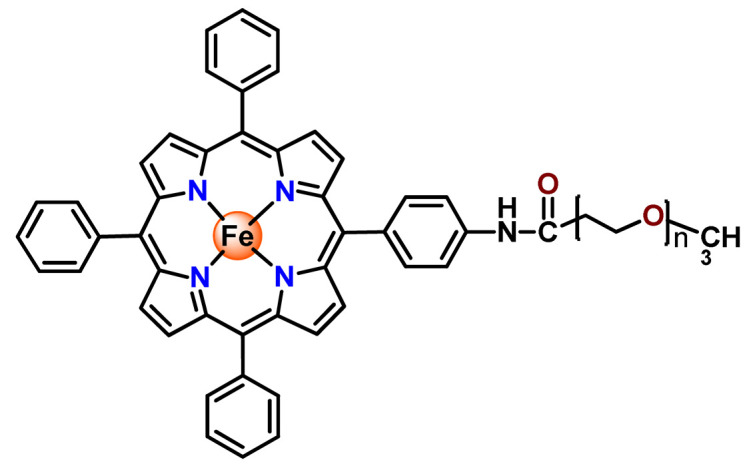 **Fe1**	Fe^2+^	PEG-PLA	Covalent conjug.	Micelles	59	pH-responsive drug delivery and co-delivery of DOX	PDT + CTX	[90]
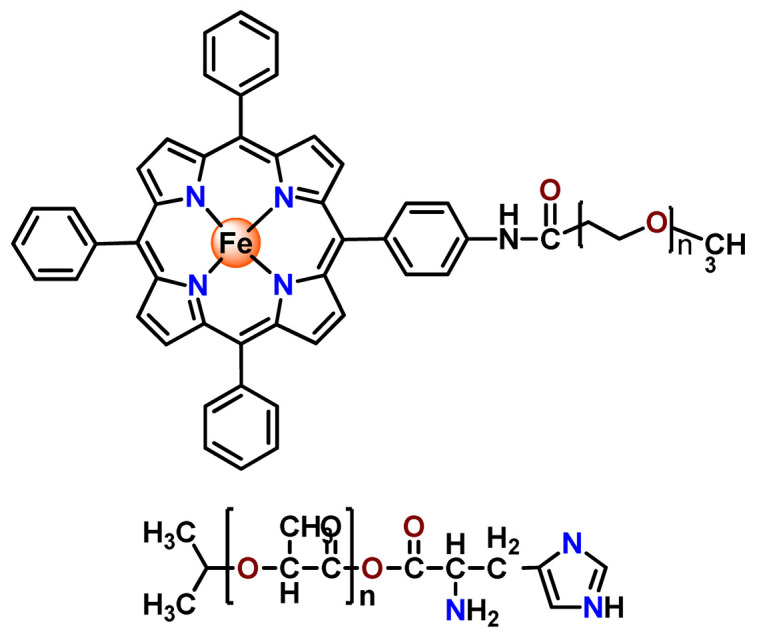 **Fe2**	Fe^2+^	polycaprolactone modified with histidine and PEG	Covalent conjug.	Micelles	33	pH-responsive drug delivery and co-delivery of DOX	PDT + CTX	[151]
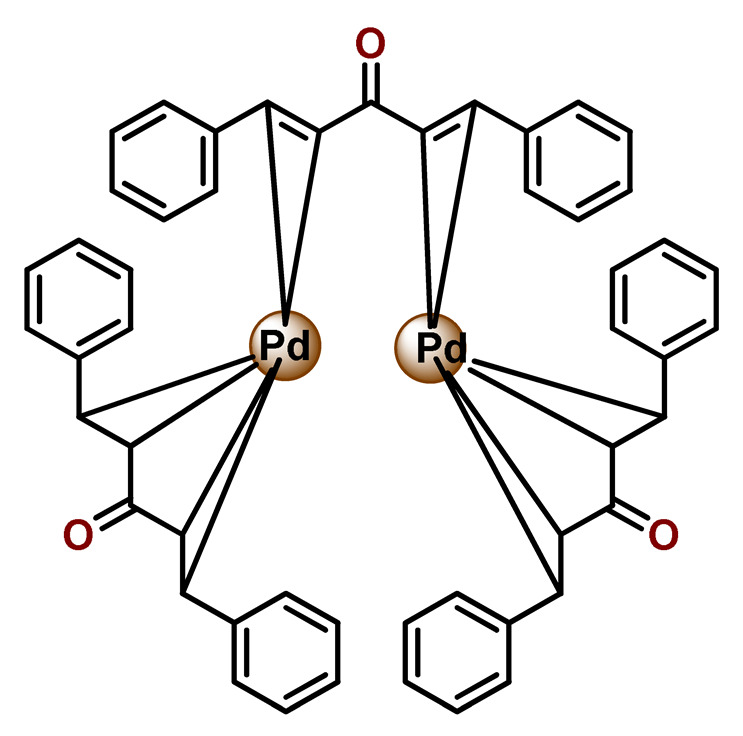 **Pd1**	Pd^+^	Hyaluronic acid	Physical encap.	NPs	204	-	CTX of advanced melanoma	[152]
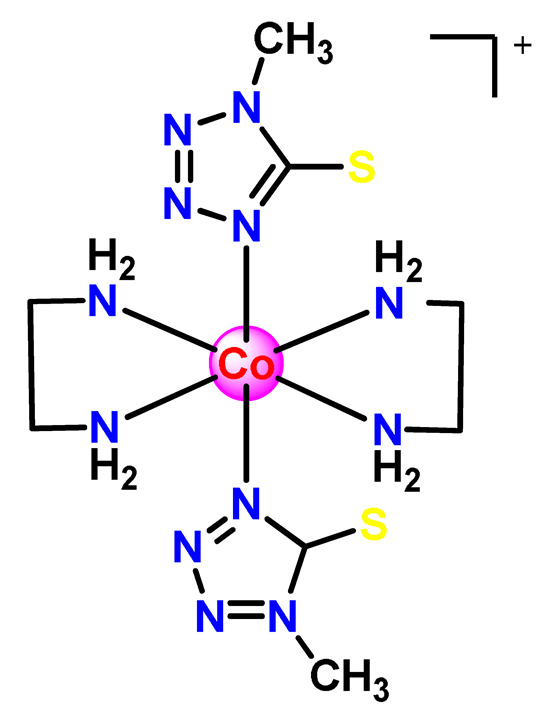 **Co1**	Co^3+^	PEG-PLGA	Physical encap.	Nanosphere	191	pH-dependent drug release	CTX against cancer and bacterial infection	[153]
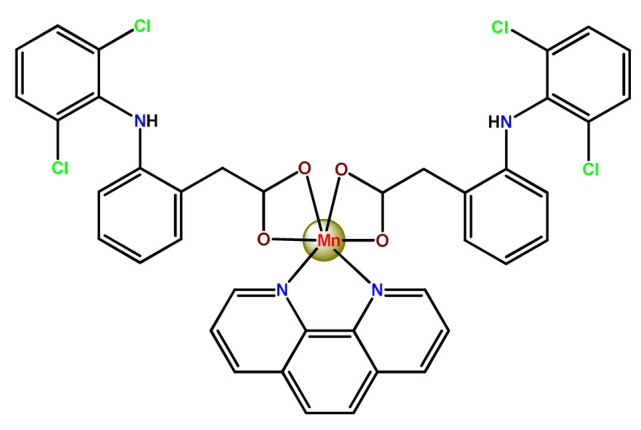 **Mn1**	Mn^2+^	PEG-PLGA	Physical encap.	Nanosphere	93	-	CTX	[154]
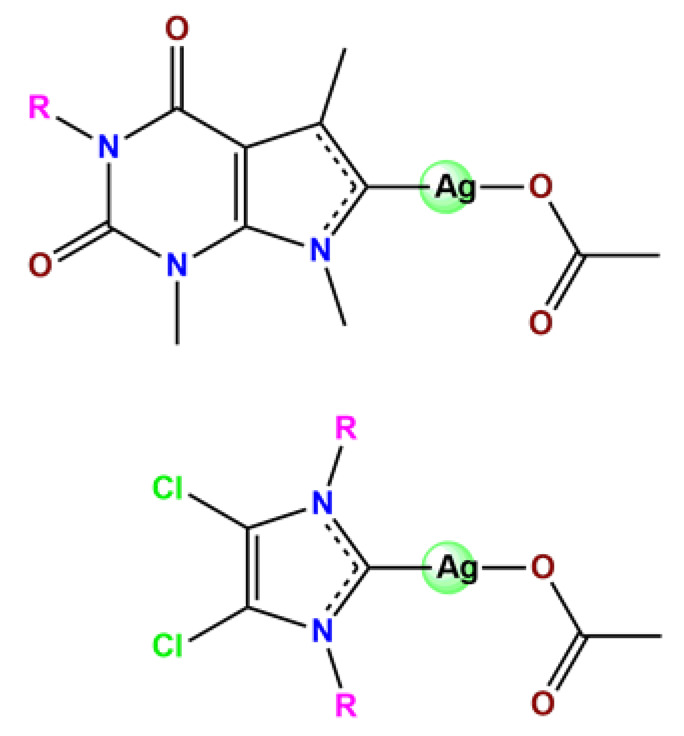 **Ag1**	Ag^+^	Poly(acrylic acid)-b-polystryrene/PEG/PEG-PLGA/polyphosphoester	Physical encap.	NPs	-	-	Antimicrobial treatment	[155,156,157]
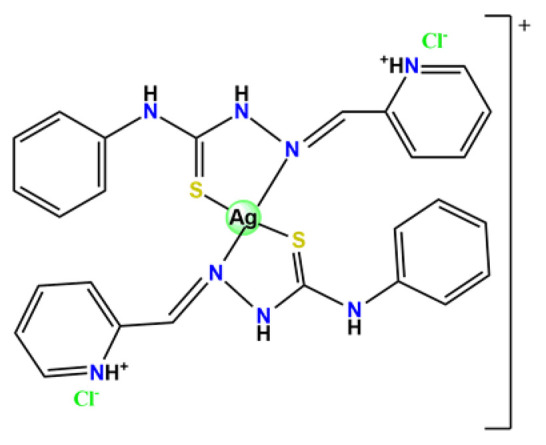 **Ag2**	Ag^+^	Poly(e-caprolactone)	Physical encap.	NPs	155–162	-	Anti-*Helicobacter pylori* activity	[158]
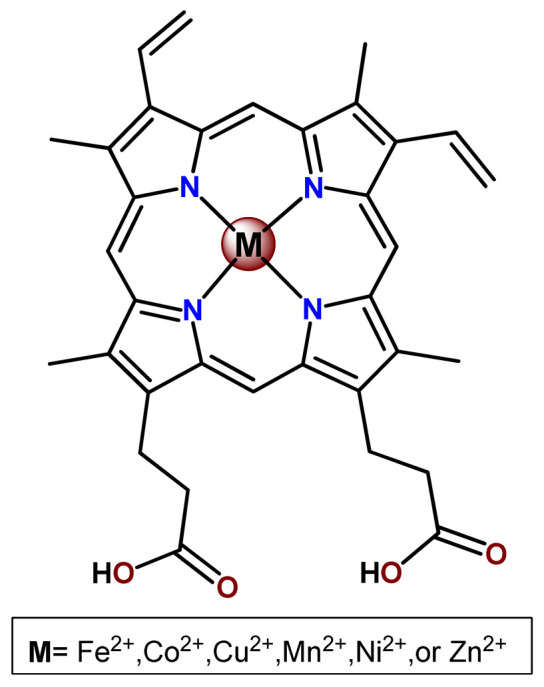 **MPPIX**	Fe^2+^ Co^2+^ Cu^2+^ Mn^2+^ Ni^2+^ Zn^2+^	Marine atelocollagen	Physical encap.	Nanocapsules	-	-	Treat human malaria parasite	[159]

* The representations of the chemical structures were produced by employing the chemical structure drawing program ChemDraw Ultra 12.0.

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
