# Peer review of "Polymeric Nanosystems Applied for Metal-Based Drugs and Photosensitizers Delivery: The State of the Art and Recent Advancements"

_pharmaceutics, 2022, doi:10.3390/pharmaceutics14071506_

Round 1

Reviewer 1 Report

The current manuscript provides an interesting account of various polymeric nanodevices for targeting delivery and controlled release of metal-containing therapeutic agents. The manuscript is comprehensive and Table 1 provides a one-stop-shop for multiple polymer-based nanosized devices.

1. The one aspect that is lacking in the manuscript is the figures representing targeting delivery and controlled release.

2. It is always challenging to quantify the encapsulation and release of the metal complexes and a separate section on this aspect would add value to the manuscript.

Author Response

The current manuscript provides an interesting account of various polymeric nanodevices for targeting delivery and controlled release of metal-containing therapeutic agents. The manuscript is comprehensive and Table 1 provides a one-stop-shop for multiple polymer-based nanosized devices.

We appreciate the reviewer’s comments.

1.The one aspect that is lacking in the manuscript is the figures representing targeting delivery and controlled release.

As suggested, we provide a schematic illustration representing targeting delivery and controlled release (Figure 3).

  1. It is always challenging to quantify the encapsulation and release of the metal complexes and a separate section on this aspect would add value to the manuscript.

We included sections 3.5 and 3.6, which approach the theme.

Reviewer 2 Report

This paper presents a large amount of information on polymeric nanoparticles for the delivery of metal-based therapeutic and diagnostic molecules. There are, however, several issues to address: 

-I do not agree with "nanodevices" and "prospects" in the title. The first is not something that can be used as such, nanoparticles or nanosystems etc would be more appropriate, while for the second, I am not sure I read any related discussion.

-English needs improvement. It is better in some parts, but still needs improvement throughout the whole manuscript.

-Intro has missing refs eg 37-43, 152-156, 214-218.

-Fig 1 "Exhibits high drug loading capability in the same formulation". This suggests the loading of more than one drugs. Is this the case?

-Line 112 "specially used for delivering hydrophobic anticancer agents". But PEG is used in many nanoparticles that can deliver hydrophobic or hydrophilic drugs. It is not clear in this paragraph if the authors mean that PEG can be used on its own or in combination with other molecules or nanoparticles. In addition, PEGylation is the process of conjugating PEG on any given molecule, eg drug lipid or peptides.

-Line 187 "Indeed, platinum-based chemotherapeutics comprise almost 50% of all currently used anticancer drugs". This statement is incorrect.

-Table 1

--Not all works have highlights. I think they should.

--It is useful to have all the structures, however, this table is very big for a paper probably. The final decision will be that of the editor in this case.

--Some numbers in Fig 3 are not in agreement with the table eg Ru and Ga

-Section 4 is an extensive reference to anticancer applications. Section 5 is way too brief to justify the generallity of the review. Are there no other applications of these systems on other types of diseases?

-Section 6.1 what is the difference between these coordination-driven systems and the ones referenced in section 4? Why are these considered new trends?

-Section 6.3 "Concerned about the fact that systematic removal of Cu ions by chelators result in unavoidable side effects ...", but the system they developed was a Cu chelator, so it should lead to systematic removal...this part is not clear to me.

-Conclusion first sentence is on coordination complexes and third on stimuli-responsive systems. This does not feel like a general conclusion to the main subject.

Author Response

This paper presents a large amount of information on polymeric nanoparticles for the delivery of metal-based therapeutic and diagnostic molecules. There are, however, several issues to address:

Thank you for raising some relevant issues for improving the manuscript.

-I do not agree with "nanodevices" and "prospects" in the title. The first is not something that can be used as such, nanoparticles or nanosystems etc would be more appropriate, while for the second, I am not sure I read any related discussion.

We agree with the reviewer, so we replaced the word “nanodevices” with “nanosystems/nanocarriers” throughout the text and we removed the word “prospect” from the title.

-English needs improvement. It is better in some parts, but still needs improvement throughout the whole manuscript.

We have tried our best to improve the English throughout the manuscript.

-Intro has missing refs eg 37-43, 152-156, 214-218.

Sorry for not being able to meet this request. One of the referees suggested that we compact the Introduction, so to add more references in reduced version of Intro was difficult for us.

-Fig 1 "Exhibits high drug loading capability in the same formulation". This suggests the loading of more than one drugs. Is this the case?

The highlighted sentence refers to the good drug capacity of polymeric NPs. There is another sentence about the capacity of load more than one drug in the same formulation (“Delivers two or more drugs simultaneously”). However, we changed the sentence highlighted to make it clearer: “Exhibits reasonable drug loading capability”

-Line 112 "specially used for delivering hydrophobic anticancer agents". But PEG is used in many nanoparticles that can deliver hydrophobic or hydrophilic drugs. It is not clear in this paragraph if the authors mean that PEG can be used on its own or in combination with other molecules or nanoparticles. In addition, PEGylation is the process of conjugating PEG on any given molecule, eg drug lipid or peptides.

We decided to remove this sentence and we added others about the use of PEG for delivery of metal-based compounds.

-Line 187 "Indeed, platinum-based chemotherapeutics comprise almost 50% of all currently used anticancer drugs". This statement is incorrect.

We agree, so we modified the sentence to: “Indeed, platinum-based chemotherapeutics are widely employed as anticancer drugs (…)”

-Table 1Not all works have highlights. I think they should.

We changed the column title to “Responsive delivery/ specific targeting moieties/co-delivery” instead of “highlights”. Now we think that the column title is more appropriate for the information contained in the column.

--It is useful to have all the structures, however, this table is very big for a paper probably. The final decision will be that of the editor in this case.

We have tried our best to compact Table 1. We removed the column with cell lines/ bacterial strains and reduced the size of the structures. We hope the editorial office will be able to find a more suitable format for including this Table in the article.

-Some numbers in Fig 3 are not in agreement with the table eg Ru and Ga

The numbers of the systems with Ru- or Ga-containing metallodrugs in Table 1 are correct. Many lines represent more than one system. For example, there are 42 Ru-containing systems cited in Table 1, of which 22 involve physical encapsulation and 20 covalent conjugations. There are five Ga-containing systems in Table 1, and all of them present physical encapsulation in polymeric NPs.  

-Section 4 is an extensive reference to anticancer applications. Section 5 is way too brief to justify the generallity of the review. Are there no other applications of these systems on other types of diseases?

As suggested, we added some other therapeutic applications of polymeric nanosystems loaded with metal-based drugs beyond cancer treatment.

-Section 6.1 what is the difference between these coordination-driven systems and the ones referenced in section 4? Why are these considered new trends?

In coordination-drive systems, the metal ion acts as part of the nanocarrier by conjugating to a classical organic drug and subsequently delivering it to a specific site. In the nanosystems, referenced in section 4, the metal ion is part of the active therapeutic agent. To clarify we added one sentence to this part of the manuscript.

-Section 6.3 "Concerned about the fact that systematic removal of Cu ions by chelators result in unavoidable side effects ...", but the system they developed was a Cu chelator, so it should lead to systematic removal...this part is not clear to me.

We changed a little the sentence to make it clearer. The Cu chelator nanosystem was designed for specifically target the neoangiogenesis of tumors, as explained in the subsequent sentence in the manuscript.

-Conclusion first sentence is on coordination complexes and third on stimuli-responsive systems. This does not feel like a general conclusion to the main subject.

To clarify, we replaced “coordination complexes” with “metal-based agents”.

Reviewer 3 Report

This is an interesting review paper on metal-based drugs and delivery. However, it seems that revisions are necessary for publication.

1. The table 1 looks well organized, but it is not in MDPI format. Is there any particular reason for using the form?

2. It is recommended to reduce Table 1 to 8 pages or less.

3. In Table 1, describe how the molecules were drawn and related program information.

4. There is a minor error in the format (line 157 , line 185, line 226, and line 383) 

5. In the content of the title and text, "nanodevices" or "device" does not seem to be an appropriate expression for this paper.

6. In the "2.Polymeric nanocarriers for metal-based drug delivery", It is made up of polymer contents common to nanoparticles. It is recommended to rewrite it in relation to metal-based drug.

7. It would be better to draw and explain Figure 2 in more detail. 

8. The "line 104-106" does not seem to need 8 references. It is recommended to reduce the number of references in other sentences.

9. It would be better to reduce the text of session 4, emphasizing section 6 as it is meaningful.

Author Response

This is an interesting review paper on metal-based drugs and delivery. However, it seems that revisions are necessary for publication.

We thank the reviewer for the positive comments on our manuscript and for reading carefully the whole text, including Table1, highlighting several points to be improved and for providing us with suggestions.

1.The table 1 looks well organized, but it is not in MDPI format. Is there any particular reason for using the form?

We use the page in landscape orientation to build Table 1 just to facilitate the visualization of the chemical structures and due to the amount of information in each line, but the final decision about the Table format we leave to the editorial office.

  1. It is recommended to reduce Table 1 to 8 pages or less.

We have tried our best to compact Table 1. We removed the column with cell lines/ bacterial strains and reduced the size of the structures. In the first version, Table 1 occupied 35 pages, now it occupies 20 pages. Table 1 is a relevant part of our manuscript since it compiles useful information providing researchers in the field with a database. Unfortunately, we can't reduce it to just eight pages as suggested. We hope the editorial office will be able to find a more suitable format for including this Table in the article.

  1. In Table 1, describe how the molecules were drawn and related program information.

We added this information to the end of Table 1: “The representations of the chemical structures were produced by employing the chemical structure drawing program ChemDraw Ultra 12.0.”

  1. There is a minor error in the format (line 157, line 185, line 226, and line 383)

Thank you for pointing out these errors, we corrected them all.

  1. In the content of the title and text, "nanodevices" or "device" does not seem to be an appropriate expression for this paper.

We agree so we changed the word “nanodevices” with “nanocarriers/nanosystem” throughout the manuscript. Thank you for this relevant advertisement.

  1. In the "2.Polymeric nanocarriers for metal-based drug delivery", It is made up of polymer contents common to nanoparticles. It is recommended to rewrite it in relation to metal-based drug.

We liked your suggestion, so we included information referent to the employment of the cited polymers for encapsulating metallodrugs by physical and covalent methods.

  1. It would be better to draw and explain Figure 2 in more detail.

As suggested, we modified Figure 2 and described the legend in more detail.

  1. The "line 104-106" does not seem to need 8 references. It is recommended to reduce the number of references in other sentences.

All references cited in this part are also cited in the table. Anyway, we have reduced the number of references in this part and in others.

  1. It would be better to reduce the text of session 4, empzhasizing section 6 as it is meaningful

We agree that the section 4 is very extended, so we dedicated efforts to reducing it.

Reviewer 4 Report

The review in its current form is not suitable for publication and should be considered for major revision.

 Below are comments which should help to improve the manuscript:

 1) The work contains some grammatical, stylistic and terminological errors. These mistakes – along with a convoluted structure – makes the paper hard to comprehend. A more compact format should be considered.

2) The introduction section should be carefully revised, Authors should give a concise, brief and comparative overview of the topic. A table which contains the overview of the topic and materials discussed should be inserted for easier review.

3) My main concern is that there are a number of reviews on this topic (e.g., Yu, Y., Xu, Q., He, S., Xiong, H., Zhang, Q., Xu, W., ... & Zhou, D. (2019). Recent advances in delivery of photosensitive metal-based drugs. Coordination Chemistry Reviews, 387, 154-179.; Su, S., & M Kang, P. (2020). Recent advances in nanocarrier-assisted therapeutics delivery systems. Pharmaceutics, 12(9), 837.), which papers should also be cited and discussed. What is the novelty of this paper?

4) Although the investigation is quite thorough, it is too long. Generally speaking, the manuscript feels like it is not finished yet. There is a huge room for improvement and a lot more could have been done to improve this work. Authors should keep in mind the review paper style, namely, the available data should be adequately demonstrated and discussed (critics, evaluation, not just writing the literature results) briefly. At least Author should insert a brief summary after each subchapter.

Author Response

The review in its current form is not suitable for publication and should be considered for major revision. Below are comments which should help to improve the manuscript:

Thank you for raising some relevant issues for improving the manuscript

1) The work contains some grammatical, stylistic and terminological errors. These mistakes – along with a convoluted structure – makes the paper hard to comprehend. A more compact format should be considered.

We put effort into carefully revising the whole manuscript to improve the English throughout the manuscript as required. To facilitate re-evaluation, all the changes were highlighted. We compacted Table 1 and made efforts to reduce the text, especially in section 4.

2) The introduction section should be carefully revised, Authors should give a concise, brief and comparative overview of the topic. A table which contains the overview of the topic and materials discussed should be inserted for easier review.

We agree that the introduction was so big, so we made changes. Now the introduction is more concise and gives a brief overview of the topic. All the changes are highlighted for easier review.

3) My main concern is that there are a number of reviews on this topic (e.g., Yu, Y., Xu, Q., He, S., Xiong, H., Zhang, Q., Xu, W., ... & Zhou, D. (2019). Recent advances in delivery of photosensitive metal-based drugs. Coordination Chemistry Reviews, 387, 154-179.; Su, S., & M Kang, P. (2020). Recent advances in nanocarrier-assisted therapeutics delivery systems. Pharmaceutics, 12(9), 837.), which papers should also be cited and discussed. What is the novelty of this paper?

The review “Recent advances in nanocarrier-assisted therapeutics delivery systems” by S. Su and P. Kang is a notable work describing the recent advances in the development and applications of various nanocarriers, emphasizing their different compositions as well as different functionalization techniques. Although this article addresses relevant issues concerned with nanocarriers' properties, it does not specifically address metal-based therapeutic agents as in our manuscript.

The review entitled “Recent advances in delivery of photosensitive metal-based drugs” by Zhou and collaborators addresses delivery carriers, including polymer nanoparticles, solid lipid, and inorganic nanoparticles, developed for delivering photosensitive metal-based drugs. Some aspect addressed by this review is coincident with our manuscript, but the main objectives of the two review articles are quite different. Our review focus on presenting the state of the art of polymeric nanocarriers for metal-based therapeutic agents’ delivery, including the non-photosensitive ones, also highlighting the employment of these nanocarriers for diseases beyond cancer. We explore in our review some biological and physicochemical aspects of the polymer-metallodrug assembly and we also summarize some recent advances in polymeric nanocarriers coupled with metal-based compounds.

Both review articles mentioned by the reviewer were cited in our manuscript.

4) Although the investigation is quite thorough, it is too long. Generally speaking, the manuscript feels like it is not finished yet. There is a huge room for improvement and a lot more could have been done to improve this work. Authors should keep in mind the review paper style, namely, the available data should be adequately demonstrated and discussed (critics, evaluation, not just writing the literature results) briefly. At least Author should insert a brief summary after each subchapter.

We thank you for the advice. We dedicated efforts to improving the quality of the manuscript. Your suggestions added to the recommendations of the other reviewers helped us in this task. Through the survey that we made to build Table 1, we could discuss the most common polymers/co-polymers used for developing nanocarriers for metal-based drug delivery. We could analyze which method (physical encapsulation or covalent conjugation) is more common for each metallic center (Fig. 4). We expand the examples of metal-based agents encapsulated into polymeric nanocarriers applied for other diseases beyond cancer and added a conclusion for this subchapter. We provided a discussion and a schematic illustration representing targeting delivery and controlled release, considering the new findings of the EPR effect, which isn’t the predominant entry pathway of NPs into tumor cells.

Round 2

Reviewer 2 Report

The authors have invested great effort to revise this manuscript and it is worth publishing in this journal. I am only a bit concerned about some language mistakes, which if corrected, would increase the quality of the work. In addition, a lot of spacings are missing in the text. Examples of language and terminology mistakes are:

-Line 9 and generally in text "targeting" should be "targeted"

-Line 10 I believe that it should be "revealing great potential"

-Line 18 should be "non-platinum"

-Line 22 should be "metal-based therapeutic agent delivery systems". I believe that several such mistakes exist in the manuscript and in such cases, plural goes only to the last noun, eg, polymeric nanocarrier(s)-assisted drug delivery systems.

-Line 25 maybe should be "assemblies". Is it one?

These are all minor. I would suggest that the authors go one more time carefully through the whole manuscript and perfect it.

Reviewer 3 Report

The authors revised the paper well. However, there are many errors such as mismatch of tense and spacing in the English expression in the text. If these problems are solved, it would be okay to publish.

Reviewer 4 Report

I think that the manuscript has been sufficiently improved and can be published in the journal.